

# On the shape of forward transit time distributions in low-order catchments

Ingo Heidbüchel[1], Jie Yang[1], Andreas Musolff[1], Peter Troch[2], Ty Ferré[2], Jan H. Fleckenstein[1]

[1]Department of Hydrogeology, Helmholtz Centre for Environmental Research – UFZ, Leipzig, 04318, Germany
[2]Department of Hydrology and Atmospheric Sciences, University of Arizona, Tucson, 85721, USA

*Correspondence to*: Ingo Heidbüchel (ingo.heidbuechel@ufz.de)

**Abstract.** Transit time distributions (TTDs) integrate information on timing, amount, storage, mixing and flow paths of water and thus characterize hydrologic and hydrochemical catchment response unlike any other descriptor. Here, we simulate the shape of TTDs in an idealized low-order catchment investigating whether it changes systematically with certain catchment and climate properties. To this end, we used a physically-based, spatially-explicit 3-D model, injected tracer with a precipitation event and recorded the resulting TTDs at the outlet of a small (~6000 m$^2$) catchment for different scenarios. We found that the TTDs can be subdivided into four parts: 1) early part – controlled by soil hydraulic conductivity and antecedent soil moisture content, 2) middle part – transition zone with no clear pattern or control, 3) later part – influenced by soil hydraulic conductivity and subsequent precipitation amount and 4) very late tail of the breakthrough curve – governed by bedrock hydraulic conductivity. The modeled TTD shapes can be predicted using a dimensionless number: higher initial peaks are observed if the inflow of water to a catchment is not equal to its capacity to discharge water via subsurface flow paths, lower initial peaks are connected to increasing available storage. In most cases the modeled TTDs were humped with non-zero initial values and varying weights of the tails. Therefore, none of the best-fit theoretical probability functions could exactly describe the entire TTD shape. Still, we found that generally the Gamma and the Advection-Dispersion distribution work better for scenarios of low and high hydraulic conductivity, respectively.

## 1. Introduction

Transit time distributions (TTDs) characterize hydrologic catchment behavior unlike any other function or descriptor. They integrate information on timing, amount, storage, mixing and flow paths of water and can be modified to predict reactive solute transport (van der Velde et al., 2010; Harman et al., 2011; Musolff et al., 2017; Lutz et al., 2017). If observed in a time series, TTDs bridge the gap between hydrologic response (celerity) and hydrologic transport (velocity) in catchments by linking them via the change in water storage and the varying contributions of old (pre-event) and young (event) water to streamflow (Heidbüchel et al., 2012). TTDs are time and space-variant and hence no TTD of any individual precipitation event completely resembles another one. Therefore, in order to effectively utilize TTDs for the prediction of, e.g., the effects of pollution events or water availability, it is necessary to find ways to understand and systematically describe the shape and





scale of TTDs so that they are applicable in different locations and at different times. In this paper we look for first order

principles that describe how the shape and scale of TTDs change, both spatially and temporally. This way we hope to

improve our understanding of the dominant factors affecting hydrologic transport and response behavior at the catchment

scale.

### 1.1.     Initial use of theoretical probability distributions

Since the concept of TTDs was introduced, many studies have reported on their potential shapes and sought ways to describe

them with different mathematical models like, e.g., the piston-flow and exponential models (Begemann and Libby, 1957;

Eriksson, 1958), the advection-dispersion model (Nir, 1964; Małoszewski and Zuber, 1982) and the two parallel linear

reservoirs model (Małoszewski et al., 1983; Stockinger et al., 2014). Dinçer et al. (1970) were the first to combine TTDs for

individual precipitation events via the now commonly used convolution integral.

Early studies reported that the outflow from entire catchments is characterized best with the exponential model (Rodhe et al.,

1996; McGuire et al., 2005). However, neither the advection-dispersion nor the exponential model is able to capture the

observed heavy tails of the solute signals in the streamflow (Kirchner et al., 2000). Instead, the more heavy-tailed TTDs

created by advection and dispersion of spatially distributed rainfall inputs traveling toward the stream can be modeled with

TTDs resembling Gamma distributions (Kirchner et al., 2001). Likewise, tracer time series from many catchments exhibit

fractal 1/f scaling, which is consistent with Gamma TTDs with shape parameter $\alpha \approx 0.5$ (Kirchner, 2016). Gamma

distributions are quite flexible and can take on very different shapes when $\alpha$ is changed: $\alpha < 1$, highly skewed distributions

with initial maximum and heavier (i.e. sub-exponential) tails; $\alpha = 1$, exponential distribution; $\alpha > 1$, less skewed, "humped"

distributions with initial value of 0, a mode and lighter tails (see Fig. S9 in the supplement for examples). Gamma

distributions can be stretched or compressed with a scale parameter ($\beta$) and their mean is the product of $\alpha$ and $\beta$. Thus when

using Gamma distributions for the determination of mean transit times (mTTs), it is necessary to choose the correct shape

parameter $\alpha$ to avoid problems of equifinality.

### 1.2.     General observations on the shape of TTDs

General observations on TTD shapes from the application of conceptual and physically-based models include that TTDs for

individual precipitation events are highly irregular and rapidly changing in time (van der Velde et al., 2010; Rinaldo et al.,

2011; Heidbüchel et al., 2012). For radial flow to a well Pedretti et al. (2013) simulated that given strong contrasts of

hydraulic conductivity between aquifer layers, TTDs tend to have power law tails with unit slope that breaks down at very

late times. If disregarded, these heavy tails constitute a significant problem for using TTDs to predict solute transport

because the legacy of contamination can be greatly underestimated. Hence, a truncation of heavy TTD tails should be

avoided. Also, when using transfer function models the computed mTT is highly sensitive to the shape of the chosen transfer

function (Seeger and Weiler, 2014) with the poorly identifiable tails greatly influencing the mTT estimates. Further

complicating matters are special cases of bimodal TTDs that can be caused by varying contributions from fast and slow





storages (McMillan et al., 2012) or from urban and rural areas (Soulsby et al., 2015). Apart from individual catchment and event properties, mixing assumptions also affect TTD modeling since certain TTD shapes are inherently linked to specific mixing assumptions (e.g. a well-mixed system is best represented by an exponential distribution, partial mixing can be approximated with Gamma distributions and no mixing with the piston-flow model) (van der Velde et al., 2015).

### 1.3.    Controls on shape variations

A number of studies reported on the best-fit shape of Gamma distributions generally ranging from $\alpha$ 0.01 to 0.90 (Hrachowitz et al., 2009; Godsey et al., 2010; Berghuijs and Kirchner, 2017; Birkel et al., 2016) which indicates L-shaped distributions with high initial values and heavier tails. Several studies found that $\alpha$ values decrease with increasing wetness conditions (e.g., Birkel et al., 2012; Tetzlaff et al., 2014) causing higher initial values and heavier tails. However, the opposite was observed in a boreal headwater catchment (Peralta-Tapia et al., 2016) where $\alpha$ ranged between 0.43 and 0.76 for all years except the wettest year ($\alpha = 0.98$). In the Scottish highlands $\alpha$ showed little temporal variability (and therefore no relation to precipitation intensity) but was closely related to catchment landscape organization – especially soil parameters and drainage density – where a high percentage of responsive soils and a high drainage density resulted in small values of $\alpha$ (Hrachowitz et al., 2010).

Conceptual and physically-based models have also been used to investigate the (temporally variable) shapes of TTDs. Haitjema (1995) found that the TTD of groundwater can resemble an exponential distribution while Kollet and Maxwell (2008) and Cardenas and Jiang (2010) derived a power-law form and fractal behavior adding macrodispersion and systematic heterogeneity to the domain in the form of depth-decreasing poromechanical properties. Increasing the vertical gradient of conductivity decay in the soil decreased the shape parameter $\alpha$ (from 0.95 for homogeneous conditions down to a value of 0.5 for extreme gradients) in a study by Ameli et al. (2016). Somewhat surprisingly, the level of "unstructured" heterogeneity within the soil and the bedrock was found to only have a weak influence on the shape of TTDs (Fiori and Russo, 2008) since the dispersion is predominantly ruled by the distribution of flow path lengths within a catchment. Antecedent moisture conditions and event characteristics influenced catchment TTDs at short timescales while land use affected both short and long timescales (Weiler et al., 2003; Roa-Garcia and Weiler, 2010). TTD shapes appeared highly sensitive to catchment wetness history and available storage, mixing mechanisms and flow path connectivity (Hrachowitz et al., 2013).

Kim et al. (2016) recorded actual TTDs in a sloping lysimeter and reported that their shapes varied both with storage state and the history of inflows and outflows. They argued that "the observed time variability […] can be decomposed into two parts: [1] 'internal' […] – associated with changes in the arrangement of, and partitioning between, flow pathways; and [2] 'external' […] – driven by fluctuations in the flow rate along all flow pathways". Replacing transit time with flow-weighted time or cumulative outflow (Niemi, 1977; Nyström, 1985) erased a substantial amount of the TTD shape variation associated with the external variability. However, since a change in the inflow often causes both fluctuations along and also a



rearrangement between the flow pathways (i.e. internal variability), flow-weighted time approaches are not able to
completely erase the influence of changes in the inflow rate.

From these partly contradictory findings, it is clear that relating best-fit values for the shape parameter $\alpha$ of the Gamma distribution to catchment or precipitation event properties does not yield a consistent picture yet. Moreover, the shape of TTDs is also dependent on the resolution of time series data (sampling frequency). While $\alpha$ can decrease with longer sampling intervals (since the nonlinearity of the flow system is overestimated when sampling becomes more infrequent
(Hrachowitz et al., 2011)), higher $\alpha$ values can also result from lowering the sampling frequency in both input (precipitation) and output (streamflow) (Timbe et al., 2015).

## 1.4.    TTD theory

To summarize, soil hydraulic conductivity, antecedent moisture conditions (storage state), soil thickness and precipitation amount and intensity are amongst the most frequently cited factors that influence the shape of TTDs. Obviously, there is not
one single property that controls the TTD shape. Instead, the interplay of several catchment and event characteristics results in the unique shape of every single TTD. One approach to deal with this problem of multicausality is the use of dimensionless numbers. Heidbüchel et al. (2013) introduced the flow path number $F$ which combines several catchment, climate and event properties into one index relating flows in and out of the catchment to the available subsurface storage. It was originally designed to monitor the exceedance of certain storage thresholds for the activation of different dominant flow
paths (groundwater flow, interflow, overland flow) at the catchment scale but can also help to categorize and predict TTD shapes.

Since McGuire and McDonnell (2006) stated a lack of theoretical work on the actual shapes of TTDs, quite a diverse range of research has been conducted to approach this problem from different angles and has yielded fragments of important knowledge. However, what is still missing is a coherent framework that enables us to structure our understanding of the
nature of TTDs so that it eventually becomes applicable to real world hydrologic problems. Already in 2010, McDonnell et al. had asked how the shape of TTDs could be generalized and how it would vary with ambient conditions, from time to time and from place to place. This study sets out to provide such a coherent framework which – although not exhaustive (or entirely correct for that matter) – will provide us with testable hypotheses on how shape and scale of TTDs change spatially and temporally. As Hrachowitz et al. (2016) put it: "an explicit formulation of transport processes, based on the concept of
transit times has the potential to improve the understanding of the integrated system dynamics […] and to provide a stronger link between […] hydrological and water quality models".

Moreover, from continuous time series of TTDs one can mathematically derive residence time distributions (describing the age distribution of water stored in the catchment), storage selection functions (describing the selection preference of the catchment discharge for younger or older stored water) (Botter et al., 2010; van der Velde et al., 2012; Benettin et al., 2015;
Harman, 2015; Pangle et al., 2017; Danesh-Yazdi et al., 2018; Yang et al., 2018) and master transit time distributions (MTTDs) (representing the flow-weighted average of all TTDs of a catchment) (Heidbüchel et al., 2012; Sprenger et al.,





2016; Benettin et al., 2017) which all can take on different shapes depending on climate and catchment properties, just like the individual TTDs. Hence the results presented in this paper can also provide insights into the use of these descriptors of catchment hydrologic processes.

**1.5.    Our approach**

In this study we will make use of a physically-based, spatially-explicit, 3-D model to systematically simulate how different catchment properties and climate characteristics and also their interplay control the shape of forward TTDs. We test which TTD shapes are most appropriate for capturing hydrologic and hydrochemical catchment response at different locations and for specific points in time. Furthermore we will try to interpret the results in the most general way possible, so that the theory

can be extended to other potential controls of the TTD shape in the future. Our modeling does not explicitly include preferential flow within the soil and bedrock (like, e.g., macropores or fractures), therefore our TTDs mostly represent systems where water is transported via overland flow coupled with subsurface matrix flow. Still, on the smaller scale the hydrologic effect of evenly distributed macropores can be represented by and reproduced with the concept of effective hydraulic conductivity. Hence, we consider our results the base for further investigations approaching ever more realistic

representations of the many hydrological processes taking place at the catchment scale.

**2.    Methods**

We used HydroGeoSphere (HGS), a 3-D numerical model describing fully coupled surface-subsurface, variably saturated flow and advective-dispersive solute transport (Therrien et al., 2010). Groundwater flow in the 3-D subsurface is simulated with Richards' equation and Darcy's law, surface runoff in the 2-D surface domain with Manning's equation and the

diffusive-wave approximation of the Saint-Venant equations. The classical advection-dispersion equation for solute transport is solved in all domains. The surface and subsurface domains are numerically coupled using a dual node approach, allowing for the interaction of water and solutes between the surface and subsurface. The general functionality of HGS and its adequacy for solving analytical benchmark tests has been proven in several model intercomparison studies (Maxwell et al., 2014; Kollet et al., 2017) and its solute transport routines have been verified against laboratory (Chapman et al., 2012) and

field measurements (Sudicky et al., 2010; Liggett et al., 2015; Gilfedder et al., 2019). Since our modeling approach entails only subsurface flow in porous media (no explicit fractures or macropores are included), the resulting TTDs have to be considered a special subset of distributions lacking some of the dynamics we can expect in real-world catchments while still providing a sound basis for further investigations (like, e.g., adding more complex interaction dynamics along the flow pathways).

## 2.1.      Model setup

A small zero-order catchment was set up, 100 m long, 75 m wide (~6000 m$^2$) with an average slope of 20 % towards the outlet and elliptical in shape (Fig. 1). The catchment converges slightly towards the center creating a gradient that concentrates flow. The bedrock is 10 m thick and has a saturated hydraulic conductivity of $K_{Br,x} = K_{Br,y} = 10^{-5}$ m day$^{-1}$ (horizontal) and $K_{Br,z} = 10^{-6}$ m day$^{-1}$ (vertical). The soil layer is isotropic, of uniform thickness and has a higher hydraulic conductivity. All other parameters are uniform across the entire model domain (based on values typically found in many catchments in Central Europe): porosity $n = 0.39$ m$^3$ m$^{-3}$, van Genuchten parameters alpha $\alpha_{vG} = 0.5$ m$^{-1}$, beta $\beta_{vG} = 1.6$, saturated water content $\theta_s = 0.39$ m$^3$ m$^{-3}$, residual water content $\theta_r = 0.05$ m$^3$ m$^{-3}$ and pore-connectivity parameter $l_p = 0.5$, longitudinal and transverse dispersivity $\alpha_L = 5$ m and $\alpha_T = 0.5$ m, respectively, free-solution diffusion coefficient $D_{free} = 8.64 \cdot 10^{-5}$ m$^2$ day$^{-1}$. Both bedrock and soil are exclusively porous media without any potential preferential flow paths like macropores or rock fractures.

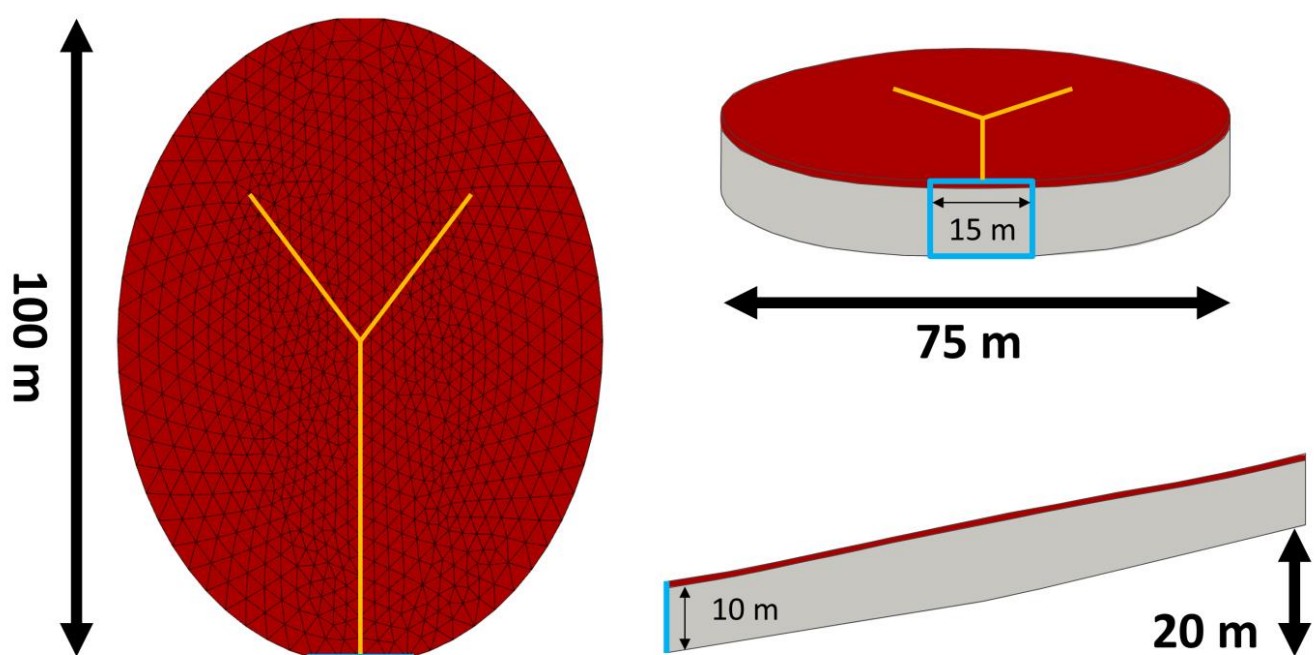

**Figure 1: 3-D model domain and shape of the virtual catchment from top (left), front (upper right) and side (lower right). The blue square indicates the outflow boundary with constant head condition. The red layer represents the soil which has a much higher hydraulic conductivity than the underlying bedrock (grey). The orange lines indicate the zone of convergence (but no explicit channel).**





### 2.1.1. Boundary conditions

Both the bottom and the sides of the domain were impermeable boundaries. A constant head boundary condition was assigned to the lower front edge of the subsurface domain (nodes in the blue square in Fig. 1), allowing outflow from both the bedrock and the soil. A critical depth boundary was assigned to the lower edge of the surface domain (above the constant head boundary) to allow for overland flow out of the catchment. The surface of the catchment received spatially uniform precipitation. Neither evaporation nor transpiration was considered during the simulations. This means that all precipitation we applied was effective precipitation that would eventually discharge at the catchment outlet. The addition of the process of evapotranspiration is planned in a follow-up modeling study to investigate what influence it exerts on catchment TTDs. The tracer was applied uniformly over the entire catchment during a precipitation event that lasted one hour, had an intensity of 0.1 mm h$^{-1}$ and a tracer concentration of 1 kg m$^3$. This resulted in a total applied tracer mass of 0.589 kg over the entire catchment.

### 2.1.2. Initial conditions

The model runs were initialized with three different antecedent soil moisture conditions $\theta_{ant}$ – a dry one ($\theta_{ant} = 22.0\,\%$; correspondent to an average effective saturation of the soil layer $S_{eff} \approx 50\,\%$), an intermediate one ($\theta_{ant} = 28.8\,\%$; $S_{eff} \approx 70\,\%$) and a wet one ($\theta_{ant} = 35.6\,\%$; $S_{eff} \approx 90\,\%$). To obtain realistic distributions of soil moisture, we first ran the model starting with full saturation and without any precipitation input and let the soils drain until the average effective saturation reached the states for our initial conditions. We recorded these conditions and used them as initial conditions of the virtual experiment runs. In general, the soil remained wetter close to the outlet in the lower part of the catchment and became drier in the upper part of the catchment. Note that the process of evapotranspiration was excluded from the modeling so that the lowest achievable saturation was essentially defined by the field capacity. An average effective saturation $S_{eff}$ of approximately 50 % was the lowest that could be achieved by draining the soil layer since the lower part stayed highly saturated due to the constant head boundary condition being equal to the surface elevation at the outlet. The upper parts of the catchment, however, were initiated with much lower $S_{eff}$ values ($\approx 30\,\%$ in the dry scenarios). That means that although an $S_{eff}$ value of 50 % seems to be quite high, it actually represents an overall dry state of the catchment soil. Throughout the modeling runs the dry initial condition did not occur again as that would have taken 13 years of drainage without any precipitation for the scenarios with high soil hydraulic conductivity $K_S$ and almost 1500 years for the scenarios with low $K_S$. The inclusion of evapotranspiration would, however, speed up the drying process of the soil and hence make these initial conditions realistic.

## 2.2. Model scenarios

To investigate how different catchment and climate properties influence the shape of forward TTDs we systematically varied four characteristics from high to low values and looked at the resulting TTD shapes of all the possible combinations (for a





total number of 36 scenarios). The properties we focused on were soil depth ($D_{soil}$), saturated soil hydraulic conductivity ($K_S$), antecedent soil moisture content ($\theta_{ant}$) and subsequent precipitation amount ($P_{sub}$) (Fig. 2).


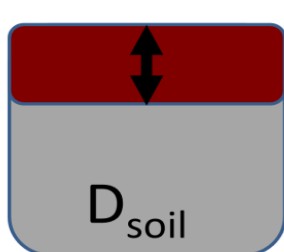 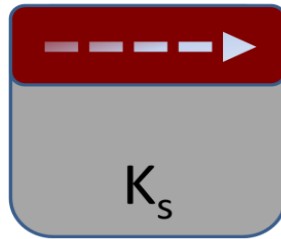 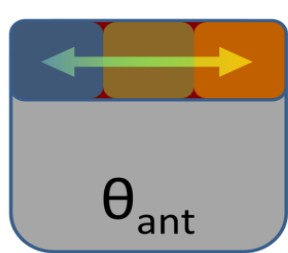 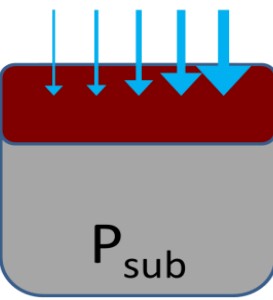

**Figure 2: The four properties that were varied to explore their influence on the shape and scale of TTDs: soil depth $D_{soil}$, saturated soil hydraulic conductivity $K_S$, antecedent soil moisture $\theta_{ant}$ and subsequent precipitation amount $P_{sub}$. The bedrock hydraulic conductivity $K_{Br}$ was kept constant for all of these base-case scenarios.**

We tested two soil depths $D_{soil}$, namely depths of 0.5 m and 1.0 m, evenly distributed across the entire catchment. Similarly, we chose two saturated soil hydraulic conductivities $K_S$, a high one with 2.0 m day$^{-1}$ (similar to fine sand) and a low one with 0.02 m day$^{-1}$ (similar to silt). Three states of antecedent moisture content $\theta_{ant}$ were selected to represent initial conditions – 50, 70 and 90 % of effective saturation. Finally the subsequent precipitation amount $P_{sub}$ was varied in three steps from 345 over 690 up to 1380 mm a$^{-1}$. We used a recorded time series of precipitation from the north-east of Germany

(the original one amounted to 690 mm a$^{-1}$) and rescaled it to obtain time series with smaller and larger amounts (Fig. 3a). The time series was 1 year long and we repeated it 32 more times to cover the entire modeling period which lasted a total of 33 years. With two soil depths, two soil hydraulic conductivities, three antecedent moisture conditions and three subsequent precipitation amounts this resulted in 36 model scenarios. Based on these 36 runs we evaluated the differences in the shape of the TTDs. The abbreviated names of the 36 model runs consist of four letters, each representing one of the properties that

we varied: the first one is $D_{soil}$ (T = thick; F = flat), the second one is $K_S$ (H = high; L = low), the third one is $\theta_{ant}$ (W = wet; I = intermediate; D = dry) and the fourth one is $P_{sub}$ (S = small; M = medium; B = big). For example the name FHIB would indicate a run with a "F"lat (shallow) soil, a "H"igh $K_S$, an "I"ntermediate $\theta_{ant}$ and a "B"ig (large) amount of subsequent precipitation (see Table 1 for an overview of the names of all 36 scenarios). We are well aware that "thick" and "flat" are technically incorrect descriptions of soil depth. However, in order to have unique identifiers (i.e. individual letters) for all 10

property states we decided to use T and F for describing deep and shallow soils, respectively.

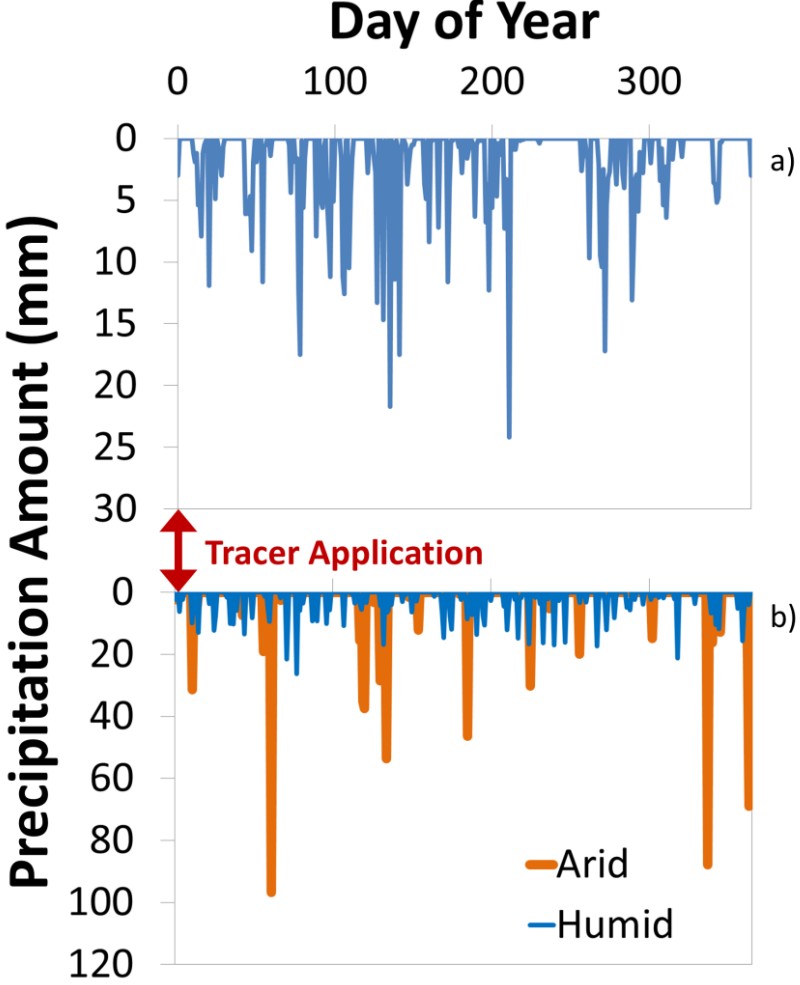

**Figure 3: a) One-year time series of subsequent precipitation (looped 33 times for the entire modeling period and rescaled for smaller or larger subsequent precipitation amounts). Tracer application took place during the first hour of the model runs. b) Time series of subsequent precipitation for a high-frequency scenario (humid) and a low-frequency scenario (arid). The total precipitation amount is the same for both scenarios.**

To complement the results obtained from the systematic variation of catchment and climate characteristics we tested the influence of six other factors: 1) soil porosity, 2) bedrock hydraulic conductivity, 3) exponential decay in hydraulic conductivity with depth in the soil, 4) frequency of precipitation events, 5) soil water retention curve and 6) effect of extreme precipitation after full saturation – conditions during which direct surface runoff may occur. These additional runs with altered soil properties, boundary and initial conditions were performed on the basis of some of the 36 initial runs (in the following sections we always indicate which runs form the basis of the specific scenarios).

We did not test the role of catchment topography and kept size, shape, slope and curvature constant. Apart from investigating the effect of an exponential decay in soil hydraulic conductivity with depth we did not add heterogeneity to the subsurface





hydraulic properties. Therefore we cannot make statements about how multiple soil layers or different spatial patterns of hydraulic conductivity would influence TTDs.

### 2.2.1. Soil porosity

The influence of larger and smaller soil porosity was investigated with six additional runs based on the three scenarios THDM, THIM and THWM (see Table S1 in the supplement for an overview on how the additional scenarios are related to

the 36 basic model scenarios). Three of the additional runs had larger (0.54 $m^3$ $m^{-3}$) and three had smaller soil porosity (0.24 $m^3$ $m^{-3}$) than the base-case scenarios (0.39 $m^3$ $m^{-3}$).

### 2.2.2. Bedrock hydraulic conductivity

Six runs were performed on the basis of the THDB scenario (which had a bedrock hydraulic conductivity $K_{Br}$ of $10^{-5}$ m day$^{-1}$). In the first run $K_{Br}$ was decreased to $10^{-7}$ m day$^{-1}$, in the following runs it was successively increased to $10^{-3}$, $10^{-2}$, $10^{-1}$,

$10^0$, $2 \cdot 10^0$ m day$^{-1}$, matching $K_S$ of the soil layer in the final run.

### 2.2.3. Decay in saturated hydraulic conductivity with depth

Because all other model scenarios had a constant hydraulic conductivity throughout the soil layer, we wanted to test whether the introduction of an exponential decay in hydraulic conductivity with depth (from high conductivity at the surface to low conductivity at the soil–bedrock interface; see Bishop et al., 2004; Jiang et al., 2009) would have a large influence on the

TTD shapes. We based the conductivity decay test on four scenarios (THDB, THWB, TLDB and TLWB) adding relationships of soil depth $z$ and saturated hydraulic conductivity $K_S$ with a shape parameter $f = 0.29$ m and saturated hydraulic conductivity at the surface $K_{S0} = 7$ m day$^{-1}$ (for the high conductivity scenarios) or $K_{S0} = 0.07$ m day$^{-1}$ (for the low conductivity scenarios), respectively (Eq. (1) and left panel on Fig. 4):

$$K_S(z) = K_{S0}e^{-\frac{z}{f}}. \tag{1}$$

This preserved the mean $K_S$ values of $2 \cdot 10^{-0}$ (high) and $2 \cdot 10^{-2}$ m day$^{-1}$ (low) (from the base-case scenarios), respectively.





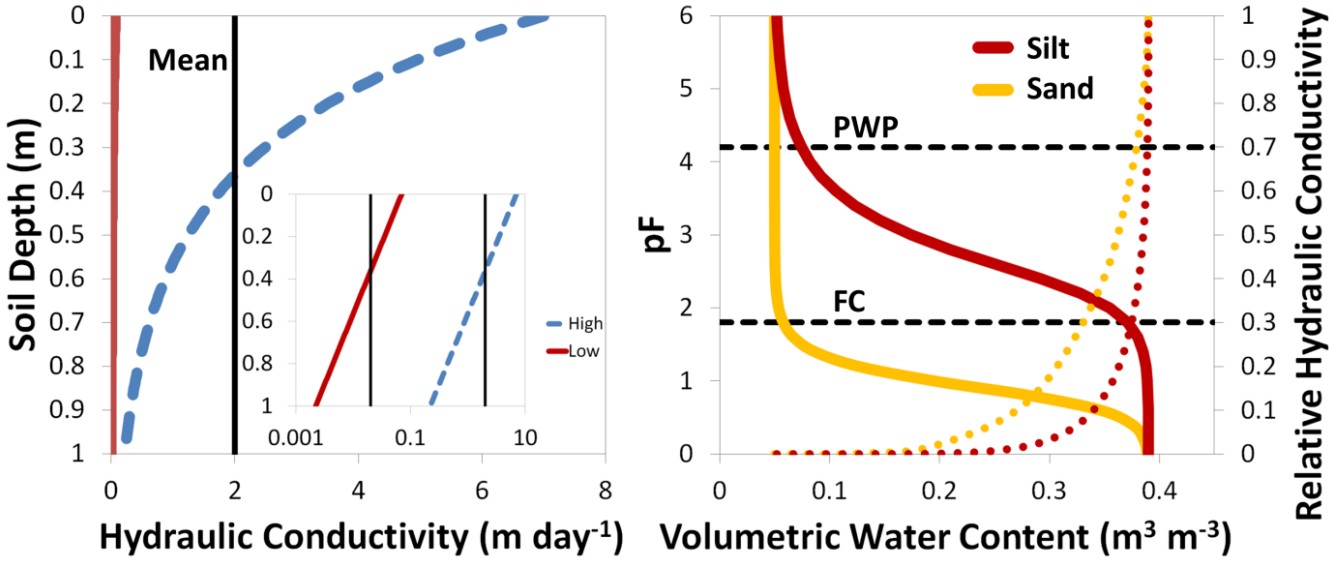

**Figure 4: Left panel: Exponential decay in saturated soil hydraulic conductivity with depth for the high (blue) and the low (red) $K_S$ scenario. The x-axis in the inset has a log scale. The spatial mean $K_S$ is indicated by the vertical black lines. Right panel: Water retention curves (solid) and relative hydraulic conductivities (dotted) for sandy and silty soils. The permanent wilting point (PWP) and the field capacity (FC) are marked as references (dashed).**

### 2.2.4. Precipitation frequency

Five time series with high precipitation event frequency and five time series with low precipitation event frequency were created using the rainfall generator used by Musolff et al. (2017) (Fig. 3b). It generates Poisson effective rainfall which is characterized by exponentially distributed rainfall event amounts and interarrival times. The mean interarrival time was set to three days and 15 days for the high frequency scenarios (comparable to a humid precipitation distribution and intensity pattern with lower intensities and more frequent events) and low frequency scenarios (comparable to an arid precipitation distribution and intensity pattern with higher intensities and less frequent events), respectively. The total precipitation for all scenarios (both humid and arid type) was 690 mm so that it matched our medium $P_{sub}$ scenarios.

### 2.2.5. Water retention curve

All the base-case model scenarios were conducted with water retention curves (WRC) resembling silty soils (Eq. 2):

$$\theta = \theta_r + \frac{\theta_s - \theta_r}{[1+(\alpha_{vG}|\psi|)^{\beta_{vG}}]^\nu},  \tag{2}$$

with van Genuchten parameters $\alpha_{vG}$ (m$^{-1}$) and $\beta_{vG}$ (dimensionless), saturated water content $\theta_s$, residual water content $\theta_r$ (both m$^3$ m$^{-3}$), pressure head $\psi$ (m) and $\nu = 1-1/\beta_{vG}$ (see Section 2.1 for van Genuchten parameter values). However, we also wanted to investigate how a different WRC in the soil layer (see right panel on Fig. 4) would influence the shape of TTDs. We chose to test a sand-type WRC since it can, in some aspects and to a certain extent, also indicate how a system with the





threshold-like initiation of rapid preferential flow behaves. The sand-type WRC causes an increase in hydraulic conductivity already at relatively lower soil water contents compared to the silt-type WRC. Hence, for the same precipitation event lateral flow is initiated faster (at lower saturations) in sandy soils since water reaches the soil–bedrock interface more quickly where it is diverted from vertical to lateral flow. The relative hydraulic conductivity $k_r$ was derived with Eq. 3:

$$k_r = S_{eff}{}^{l_p} \left[ 1 - \left( 1 - S_{eff}{}^{v^{-1}} \right)^v \right]^2,$$ (3)

with effective saturation $S_{eff}$ and pore-connectivity parameter $l_p$ (both dimensionless). Other aspects of preferential flow – like bypass flow through macropores in deeper soil layers – are, however, not captured by sand-type WRCs. The van Genuchten parameters for the sand-type WRC were defined as follows: $\alpha_{vG} = 14.5$ m$^{-1}$ and $\beta_{vG} = 2.68$. We based the additional eight runs on the scenarios THDB, THWB, THDS, THWS, TLDB, TLWB, TLDS and TLWS.

### 2.2.6. Full saturation and extreme precipitation intensity

We tested these effects for two scenarios (THWB and TLWB) out of the 36 systematic model runs since both of these scenarios were already close to creating overland flow. Full saturation in this case means that the initial condition for these model runs consisted of a fully saturated domain (both in the bedrock and in the soil), i.e. $S_{eff}$ was 100 % ($\theta_{ant} = 39$ %). Additionally, we increased the intensity of the input precipitation event (delivering the tracer) from 0.1 mm h$^{-1}$ (normal) over 10 mm h$^{-1}$ (very large, +) to 100 mm h$^{-1}$ (extreme, +++), in an attempt to create infiltration excess overland flow and record its influence on the shape of TTDs.

### 2.3. Influence of the sequence of precipitation events

We also tested to what extent sequences of precipitation events with different magnitude, intensity and interarrival time influence TTD shapes. This was necessary to assure that our resulting TTD shapes were not primarily a product of the point in time – within the sequence of precipitation events – at which the tracer was applied to the catchment. To this end 15 precipitation event time series were created using the rainfall generator used by Musolff et al. (2017). The mean interarrival time was set to three days (comparable to a precipitation distribution and intensity pattern found in humid environments with low intensities and more frequent events) and the total precipitation amount for all scenarios was 690 mm matching our medium $P_{sub}$ scenarios (Fig. S1). The generated precipitation time series resembled our original time series of precipitation which also had an interarrival time close to three days. All other parameters and properties of the 15 model runs were based on the THDM scenario.

### 2.4. Processing of the output data

The output data from HydroGeoSphere was mainly processed with Microsoft Excel. We summed surface and subsurface flows, computed total tracer outflow from the catchment, created the probability density and cumulative probability density



distribution for tracer outflow, calculated the shape parameters of the forward TTDs, fitted theoretical distributions to our data and smoothed the original TTDs for better visual comparability of the shapes. HydroGeoSphere keeps track of the mass balance of inflow, outflow and storage and calculates the discrepancy (mass balance error) between the three terms (Fig. S2). The absolute mean mass balance error for the 36 runs was negligible ($6.8 \cdot 10^{-2} \pm 7.2 \cdot 10^{-2}$ %).

### 2.4.1. Creation of TTDs

The probability density distributions of transit time (the forward TTDs) were created by normalizing the mass outflux $J_{out}$ (kg d$^{-1}$) for each time step by the total inflow mass $M_{in}$ (kg) (Eq. 4).

$$TTD(t) = J_{out}^{norm}(t) = \frac{J_{out}(t)}{M_{in}}. \tag{4}$$

The cumulative TTDs (dimensionless) were created by multiplying the normalized mass outflux (d$^{-1}$) of each time step by the associated time step length $\Delta t$ (d) before cumulating it (Eq. 5):

$$TTD_{cml}(t) = \sum_{t=0}^{t}(J_{out}^{norm}(t) * \Delta t). \tag{5}$$

### 2.4.2. Calculation of TTD metrics

For each TTD we calculated seven parameters to characterize its shape: the first quartile ($Q_1$), the median ($Q_2$), the mean (mTT), the third quartile ($Q_3$), the standard deviation ($\sigma$), the skewness ($\nu$) and the excess kurtosis ($\gamma$) (see Text S1 and Fig. S3 in the supplement for details on the calculation and for visual comparison of the metrics). Furthermore we determined the young water fraction $F_{yw}$ as the fraction of water leaving the catchment after 2.3 months (Jasechko et al., 2016; Kirchner, 2016; Wilusz et al., 2017). For more details on how $F_{yw}$ changes with catchment and climate properties, see Text S2, Fig. S4 and Table S2 in the supplement.

### 2.4.3. Fitting

We fitted predefined mathematical probability density functions to the modeled data since condensing the main characteristics of an observed probability distribution into just one to three parameters of a mathematical function is appealing and eases the potential of transferability of the findings. Massoudieh et al. (2014) explored the use of freeform histograms as groundwater age distributions and concluded that mathematical distributions performed better in terms of their ability to capture the observed tracer data relative to their complexity. In order to determine which theoretical probability density function best captures the shape of our modeled TTDs, we chose two probability density functions that are commonly used to describe the transit of water through catchments (the Advection-Dispersion and the Gamma model), as well as the Beta distribution because its shape is extremely flexible:

1) The Advection-Dispersion distribution (AD) with dispersion parameter $D$ (dimensionless) and mean mTT (d) is of the form of an inverse Gaussian distribution (Eq. 6):



$AD(t) = (\frac{4\pi Dt}{mTT})^{-0.5} \frac{1}{t} exp\{-[(1-\frac{t}{mTT})^2 * \frac{mTT}{4Dt}]\},$                                   (6)

2) The three parameter Beta distribution with shape parameters $\alpha$ and $\beta$ (dimensionless) and upper limit $c$ (d) (with mean mTT=$\alpha c/(\alpha+\beta)$) (Eq. 7):

$Beta(t) = \frac{t^{\alpha-1}(c-t)^{\beta-1}}{c^{\alpha+\beta-1}B(\alpha,\beta)}.$                                   (7)

The fourth parameter of the Beta distribution is the lower limit $a$. It is not included in the above definition since in our case it

is zero.

3) The Gamma distribution with shape parameter $\alpha$ (dimensionless) and scale parameter $\beta$ (d) (with mean mTT=$\alpha\beta$) (Eq. 8):

$Gamma(t) = t^{\alpha-1} \frac{e^{-t/\beta}}{\beta^\alpha \Gamma(\alpha)}.$                                   (8)

The method of least squares was used to find the best fit between the modeled TTDs and the theoretical distribution functions (i.e. minimizing the sum of the squared residuals with the Solver function in Excel).

The fitting was performed on the cumulative probability distributions since their shape is not subject to the more extreme internal variability that the probability distributions can experience.

### 2.4.4.   Smoothing

Smoothing was only applied to enhance the visual comparability of the TTDs. All calculations were performed on the unsmoothed TTDs. For details on the smoothing method see Text S3 and Fig. S5 in the supplement.

### 2.5.   Flow path number

The flow path number $F$ is a dimensionless number proposed by Heidbüchel et al. (2013) that relates catchment inflow to outflow (in the numerator) while simultaneously assessing available storage space (in the denominator) for each point in time and at the catchment scale. It was introduced to define thresholds for the activation and deactivation of different flow paths that transport water more slowly (e.g. groundwater flow), faster (interflow) or very fast (macropore flow, overland

flow). For this paper we modified $F$ slightly so that both numerator and denominator have the dimensions (m$^3$) (Eq. 9):

$F(t) = \frac{P_{dr}(t)-K_{rem}}{D_{soil}(n-\theta_{ant}(t))A_{in}},$                                   (9)

where soil depth $D_{soil}$ (m), catchment surface area $A_{in}$ (m$^2$), porosity $n$ (m$^3$ m$^{-3}$) and antecedent moisture content $\theta_{ant}$ (m$^3$ m$^{-3}$) are paired with the driving precipitation amount $P_{dr}$ (m$^3$) which is calculated as the average subsequent precipitation amount $P_{sub}$ (m a$^{-1}$) over the average event duration $t_e$ (d) (Eq. 10):

$P_{dr}(t) = \frac{t_e P_{sub}(t)A_{in}}{365.25}.$                                   (10)


The subsequent precipitation amount $P_{sub}$ (m a$^{-1}$) is calculated for every time step as the amount of precipitation falling within the year that follows this time step using a moving window. Note that differing from Heidbüchel et al. (2013) we used the event duration $t_e$ instead of the interevent duration $t_i$ to compute $P_{dr}$ since it better represents the amount of precipitation falling during an average event filling up the available storage. Furthermore, there is the subsurface discharge capacity of the soil $K_{rem}$ (m$^3$) consisting of the effective saturated soil hydraulic conductivity $K_S$ (m day$^{-1}$), the sum of the average interevent and event duration $t_i+t_e$ (d), the porosity $n$ (m$^3$ m$^{-3}$) and the cross-sectional area of the soil layer at the outlet of the catchment $A_{out}$ (m$^2$) (Eq. 11):

$$K_{rem} = (t_i + t_e)K_S nA_{out}. \tag{11}$$

The cross-sectional area of the soil layer at the outlet of the catchment $A_{out}$ can be regarded to represent the connection of the catchment to either a river channel or to the alluvial valley fill where medium to rapid subsurface outflow from the catchment can occur. Note that differing from Heidbüchel et al. (2013) we used the sum of the interevent and event duration $t_i+t_e$ instead of just the event duration $t_e$ to compute $K_{rem}$ since it better represents the amount of water that can be removed from the catchment during an average precipitation cycle.

The flow path number $F$ varies in time mainly due to the changes in antecedent moisture content $\theta_{ant}$ since variations in the amount of driving precipitation $P_{dr}$ are damped due to the moving window approach that is used to compute it. A positive flow path number $F$ indicates that there is a surplus of water entering the catchment that cannot be removed by subsurface transport at the same rate. Hence, the storage fills up. Conversely, a negative $F$ indicates that the drainage capacity of the catchment exceeds the water inputs and the amount of stored water decreases. Furthermore, values between 0 and 1 signal that the available soil storage space is able to accommodate the net inflow of water, while values larger than 1 mean that the catchment receives more water than it can discharge or store in the subsurface. In turn, the larger the storage capacity in the subsoil, the more $F$ converges towards 0. There is only one notable important exception to this last rule: In highly conductive soils the increase in discharge capacity (caused by the increase in the cross-sectional area of the soil layer at the outlet $A_{out}$) can be larger than the increase in storage capacity itself – leading to $F$ becoming even more negative with increasing storage capacity.

## 3.      Results

Output from the model runs comprised subsurface discharge, overland discharge and tracer concentration in the discharge from which we derived TTDs (for an example see Fig. S6 in the supplement). Additionally, the model provided spatially and temporally resolved tracer concentrations throughout the entire domain. The differences emerging between the individual TTDs can be tracked by looking at the spatio-temporal evolution of the applied tracer impulse throughout the entire catchment. For a detailed example please refer to Text S4 and Fig. S7 in the supplement.





### 3.1. Influence of the sequence of precipitation events

Changing the sequence of precipitation events affects the shape of the TTDs to a certain degree. Especially the timing and magnitude of the first precipitation event determines how strong the early response turns out. This can be observed in Fig. 5 where the different TTDs split up into different branches according to the arrival and magnitude of the first event after tracer application. However, following this initial split – with more and more precipitation events taking place – all TTDs tend to converge towards a single line. Examining the cumulative TTDs in Fig. 5 it is obvious that the variability in the TTD shape introduced by different precipitation event sequences is much smaller than the variability introduced by the other catchment and climate properties. While the range of $Q_1$ observed for the 15 scenarios with different event sequences is still 14 % of the total range observed for the 36 base-case scenarios, this percentage decreases down to 2 % for $Q_3$. The other distribution metrics describing the shape of the TTDs also vary a lot less between the scenarios with different event sequences compared to the scenarios with different catchment and climate properties (the range of all event sequences is only 1.1 % of the range of all base-case scenarios for the standard deviation, 1.6 % for the skewness and 1.0 % for the excess kurtosis). A table with the distribution metrics for all 15 scenarios can be found in the supplement (Table S3). Therefore we can assume that the shape of TTDs is not significantly influenced by the precipitation event sequence – at least in environments with a naturally short interarrival time resembling humid climate conditions and an event amount distribution that is exponential.

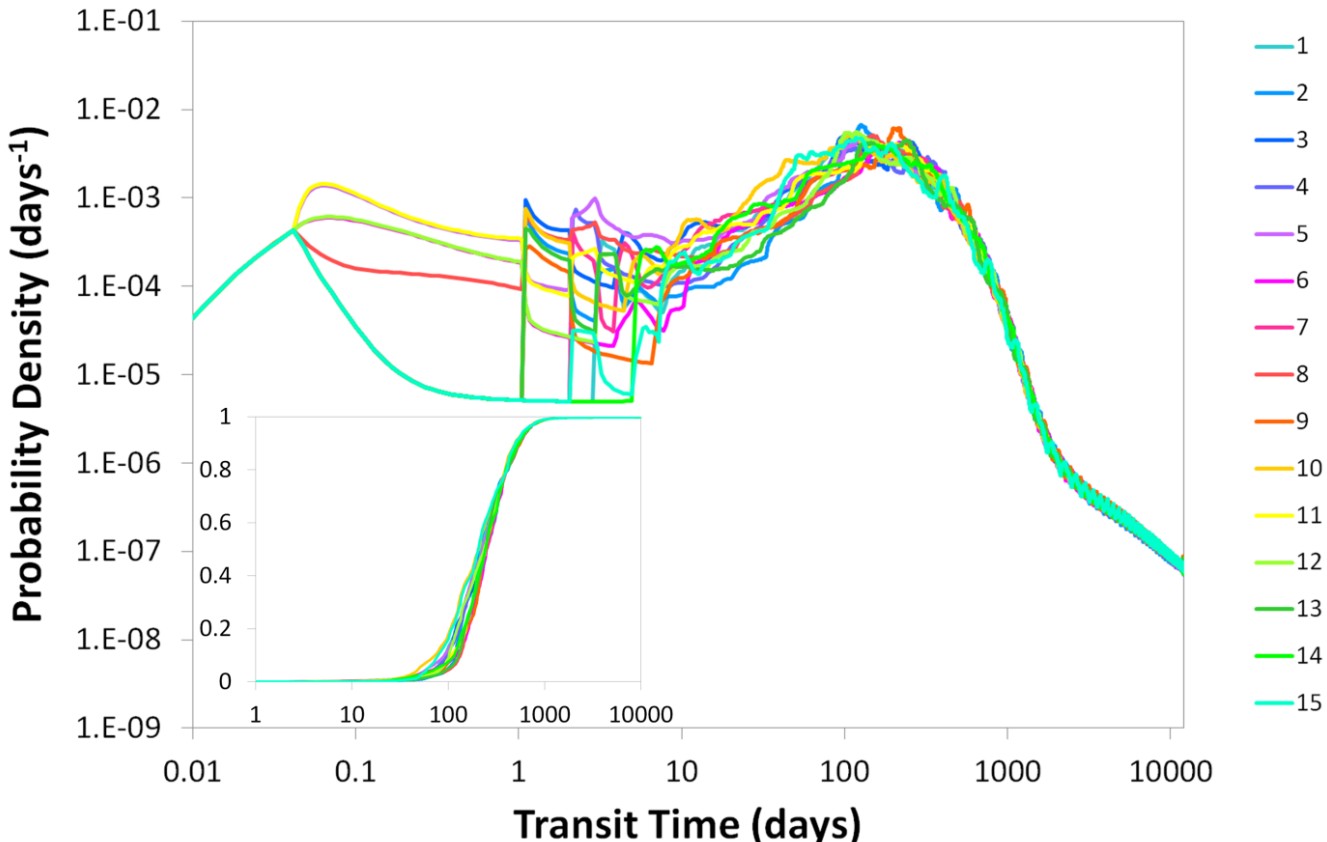

**Figure 5: 15 TTDs resulting from 15 different precipitation time series with all other catchment and climate properties being equal. The first few events have the largest influence on the TTD shapes, while subsequent events gradually even out the differences. Inset shows cumulative distributions.**

### 3.2. Effects on TTD metrics

We found that $\theta_{ant}$ affects the young parts of TTDs (the first 10 days) a lot more than the older parts (its influence is hardly discernible after approximately 100 days). By contrast, $K_S$ affects the older parts more than the young parts. $D_{soil}$ and $P_{sub}$ influence all parts of the TTDs equally strong and hence have the smallest influence on the actual shape of the distributions (center panel in Fig. 6). As can be observed in the upper left panel, the influence of $K_S$ is a lot stronger in scenarios with high $\theta_{ant}$ while the influence of $P_{sub}$ decreases with increasing $\theta_{ant}$. The upper right panel shows that both $\theta_{ant}$ and $P_{sub}$ have a larger influence when $K_S$ is high, but for $P_{sub}$ this increase in influence is only seen for the longer transit times. The lower left panel confirms the impression that $D_{soil}$ only has a minor influence on the shape of TTDs – all parts of the TTDs are equally affected and it does not make a significant difference for the influence of the other factors whether the soils are deeper or shallower. Finally in the lower right panel it is demonstrated that $P_{sub}$ has opposite effects on the influence of $\theta_{ant}$ and $K_S$: Larger $P_{sub}$ causes the influence of $K_S$ to increase for the longer transit times while the influence of $\theta_{ant}$ decreases when $P_{sub}$ becomes larger. The fact that different catchment and climate properties have varying degrees of control on transit times





depending on current conditions and the interplay of dominant hydrologic processes has already been observed in the field (Heidbüchel et al., 2013). Table 1 lists all metrics of the 36 TTDs resulting from the base-case scenarios.

**Figure 6: Influence of different properties on different parts of the TTDs. Shown is the average percent decrease in transit time for each quartile (Q₁, Q₂, Q₃) and the mean (μ) of the TTDs caused by a decrease in $D_{soil}$ from 1 to 0.5 m, an increase in $K_S$ from 0.02 to 2 m day$^{-1}$, an increase in $\theta_{ant}$ from 50 % to 90 % effective saturation $S_{eff}$ and an increase of $P_{sub}$ from 0.3 to 1.4 m a$^{-1}$. The panel in the center in the foreground shows the decrease in transit time for changing each of the four properties, the four panels in the background show the decrease in transit time conditional on the variation of one of the four properties ($\theta_{ant}$, $K_S$, $D_{soil}$, and $P_{sub}$), respectively.**

### 3.2.1. Antecedent moisture content

High $\theta_{ant}$ results in higher initial peaks for TTDs (Fig. 7). When increasing $\theta_{ant}$ by 14 % (from $S_{eff}$ 50 % to 90 %), on average

Q₁ is shortened by 44 %, Q₂ decreases by 27 %, the mTT by 19 % and Q₃ by 15 % (Fig. 6 center, Table 1). The median $F_{yw}$

increases by 16 %. Neither the standard deviation (and hence the width) nor the skewness nor the kurtosis values of the





TTDs are affected much by $\theta_{ant}$ though. Higher $\theta_{ant}$ initially promotes faster lateral transport (both on the surface and in the subsurface) while impeding percolation of tracer towards the bedrock, therefore more tracer is transported fast towards the outlet and less tracer is entering the deeper soil layers and the bedrock.

**3.2.2.    Saturated hydraulic conductivity**

A decrease in $K_S$ causes more pronounced ups and downs in the TTD with the effect of individual rainfall events being better discernible even in the later parts of the TTD (Fig. 8). Increasing $K_S$ by 2 orders of magnitude on average shortens $Q_1$ by 44 %, $Q_2$ by 58 %, the mTT by 59 % and $Q_3$ by 62 % (Fig. 6 center, Table 1). The median $F_{yw}$ increases by 13 %. The standard deviation increases with decreasing $K_S$, while the skewness and kurtosis both decrease significantly – TTDs become

less skewed and more platykurtic (flatter). The interplay between $K_S$ and $\theta_{ant}$ is obvious in that the influence of $\theta_{ant}$ decreases over time while the influence of $K_S$ increases. Initially $\theta_{ant}$ controls the soil hydraulic conductivity, the partitioning of the tracer into surface and subsurface flow and also the spreading within the soil. Later on, as moisture conditions become more similar for scenarios with identical $P_{sub}$ and $D_{soil}$, $K_S$ gains in importance while $\theta_{ant}$ becomes less relevant.

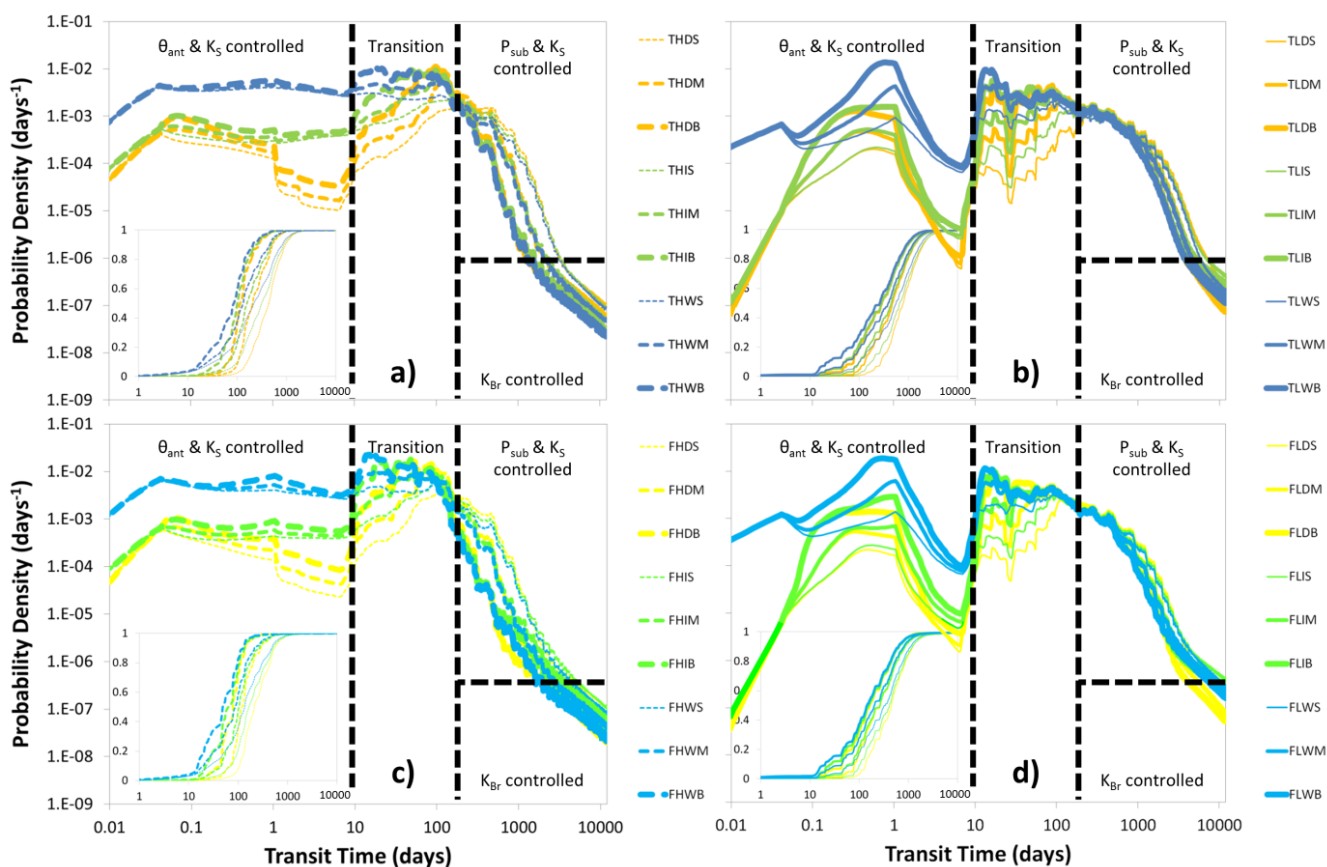



**Figure 7: Results of the 36 model runs. TTDs are grouped by soil depth (upper panels a and b = deep (thick); lower panels c and d = shallow (flat)) and hydraulic conductivity (left panels a and c = high; right panels b and d = low). Yellow colors indicate dry, green intermediate and blue wet antecedent moisture conditions; thick lines indicate large, mid-sized lines medium and thin lines small amounts of subsequent precipitation amounts. Insets show cumulative TTDs. Dashed black lines divide TTDs into four parts, each part controlled by different properties. Note the log-log axes.**

### 3.2.3. Subsequent precipitation amount

Large $P_{sub}$ compresses the TTDs (Fig. 7). Doubling $P_{sub}$, on average shortens $Q_1$ by 63 %, $Q_2$ decreases by 61 %, the mTT by 57 % and $Q_3$ by 58 % (Fig. 6 center, Table 1). The median $F_{yw}$ increases by 22 %. The standard deviation (and hence the width) decreases by 42 %, while the skewness of the TTDs more than doubles. Larger $P_{sub}$ causes more leptokurtic (peaked) TTDs. Big amounts of $P_{sub}$ increase the total flow through the catchment (both in the soil and bedrock) and hence control how effectively tracer is flushed out of the system.

### 3.2.4. Soil depth

Decreasing $D_{soil}$ causes a larger fraction of tracer to arrive at the outlet faster (Fig. 8). Halving $D_{soil}$ shortens all the quartiles and the mTT of the TTDs on average by approximately 40 % (Fig. 6 center, Table 1), while the median $F_{yw}$ increases by 10 %. The standard deviation (the width of the TTD) is decreased by 19 % and the skewness is increased by about 56 %. Shallower soils cause more leptokurtic (peaked) TTDs almost doubling the excess kurtosis. Shallower soils saturate faster than deeper soils, they also redirect tracer more quickly from vertical to lateral flow, and therefore the early response in shallower soils is slightly stronger.

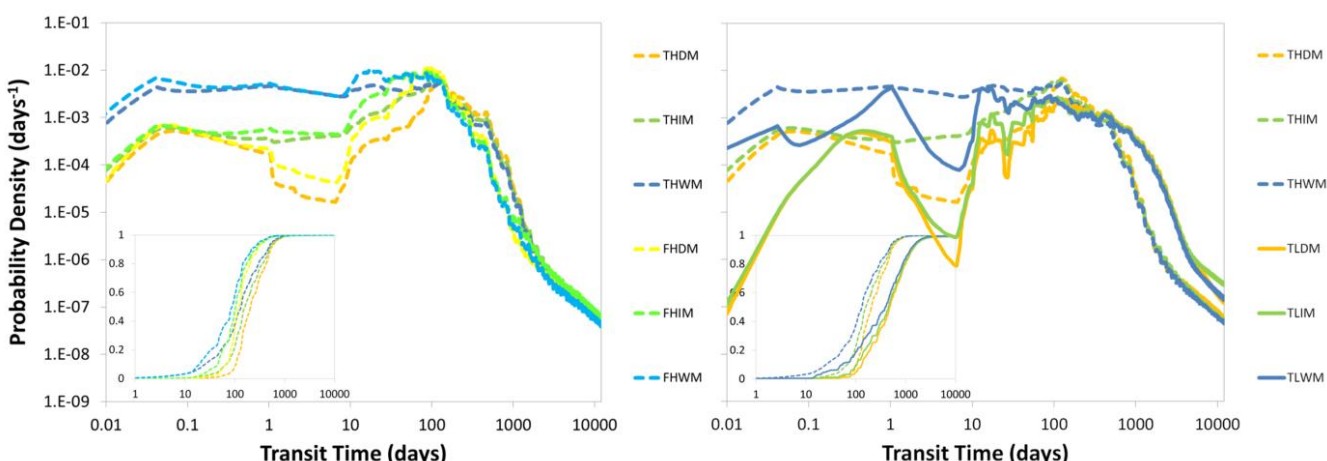

**Figure 8: Influence of soil depth (left) and saturated soil hydraulic conductivity (right) on the shape of TTDs. Lighter shades of one color indicate shallower soils; dashed lines indicate higher hydraulic conductivity. Insets show cumulative TTDs.**



### 3.3. General observations on the shape of TTDs

The simulation results suggest that the TTDs can be visually divided into four distinct parts (Fig. 7), where the shape of three
parts is clearly controlled by the catchment and climate properties and the fourth is a transition zone. The shape of the initial
part of the TTD (up to ~10 days) depends strongly on $\theta_{ant}$ and $K_S$ and less strongly on $D_{soil}$. For example, TTDs in soils with
high $\theta_{ant}$ or $K_S$ exhibit higher initial peaks with a larger probability for short transit times. Starting approximately after 10
days a transition period follows where no individual parameter dominates. During this period precipitation drives the
emptying of the uppermost soil layers with the presence of faster and/or larger flows (in catchments with higher $K_S$ / bigger
$P_{sub}$) being gradually compensated by higher remaining concentrations of tracer (in catchments with lower $K_S$ / smaller $P_{sub}$)
so that the tracer mass outflux at the catchment outlet converges towards a very similar value at around 120 days before
diverging again. After the transition period, the shape of the TTDs is governed by $P_{sub}$ (i.e. essentially the climate) and $K_S$,
with larger $P_{sub}$ and higher $K_S$ causing a more rapid decline of outflow and hence a compression of the TTDs. Finally, the
shape of the tails of the TTDs is controlled by the hydraulic conductivity of the bedrock $K_{Br}$ (not the soil $K_S$). These tails
follow power law functions in many (but not in all) cases. Furthermore, all modeled TTDs share one common feature – for
every subsequent precipitation event there is a more or less discernible spike. Generally, larger subsequent events cause
higher spikes (i.e., a higher proportion of outflow during those events) while the size of the spikes decreases at later times.
And although this multitude of local maxima in the probability density curve does invoke a sense of irregularity, the general
pattern of shapes of the TTDs is not influenced by the individual subsequent events, which is why we decided to smooth the
TTDs for visual comparison so that the underlying systematic changes in shapes are more clearly visible and understood (see
Fig. S5 in the supplement).

We also plotted the probability density replacing the actual transit time with the cumulative outflow to check whether this
would eradicate the differences between the different distributions (see Fig. S8 in the supplement). We made two interesting
observations: 1. For the scenarios with high $K_S$, the differences between the distributions were reduced considerably.
Especially for the cumulative probability distributions there were hardly any discernible differences left. The largest
discrepancies could still be found in the early part of the distributions where the distributions with high $\theta_{ant}$ continued to have
larger outflow probabilities. 2. For the scenarios with low $K_S$, the individual distributions did not collapse into a single
cumulative probability distribution. They rather split up into three distributions according to their $P_{sub}$ values. That means
that for the scenarios with larger $P_{sub}$ a larger amount of cumulative outflow was necessary to flush out the same amount of
tracer compared to the scenarios with smaller $P_{sub}$.

### 3.4. Distribution fitting

Shape parameters of the best-fit Gamma ($\alpha$) and Advection-Dispersion ($D$) distributions as well as flow path numbers ($F$) for
the 36 different scenarios are listed in Table 2. The parameters $\alpha$ and $D$ range from 0.78 to 3.66 and from 0.15 to 0.98,
respectively. $F$ ranges from –0.22 to 0.63.



Out of the 36 model scenarios, the Advection-Dispersion model (AD) yielded the best fit 19 times, the Beta model 5 times and the Gamma model 12 times (however, 14 times there was no significant difference in the performance of the Beta and Gamma models). In general, the AD model works better for high $K_S$ and dry $\theta_{ant}$, the Beta and Gamma models for low $K_S$ and wet $\theta_{ant}$ (Table 3). The Gamma model represents the mean transit time (mTT) less correctly than the other two models (Table 3). On average, the mTT of the fitted Gamma models deviates from the observed mean by 30 % (88 days) with a

maximum deviation of 423 days for one scenario, underpredicting in dry and overpredicting for wet $\theta_{ant}$. The AD and Beta models perform much better in this regard with an average deviation from the mTT of only 5 and 4 % (17 and 13 days) with maximum deviations of 102 and 38 days, respectively. The Beta model almost always slightly underpredicts the mTT while the AD model overpredicts the mean when $P_{sub}$ is small. The correct identification of the median transit time works much better for the Gamma model – here the average deviation of the fitted median from the observed median is only 4 % (12

days) with a maximum deviation of 59 days matching the performance of the Beta model. The AD model yields average deviations from the median transit time of 6 % (15 days) with a maximum deviation of 50 days.

### 3.5. Predicting the shape of TTDs

Non-linear regression analysis relating the shape and scale parameters of the fitted AD and Gamma distributions to any single soil, precipitation or storage property ($D_{soil}$, $K_S$, $\theta_{ant}$, $P_{sub}$) did not yield satisfying relations that could be used to predict

TTD shapes. The best relationships we found were between the shape and scale parameters and $K_S$: 1) $\alpha$ is related to $K_S$ via a positive exponential relationship ($R^2 = 0.74$) for dry $\theta_{ant}$, 2) $\beta$ is related to $K_S$ via a negative exponential relationship ($R^2 = 0.73$) for dry $\theta_{ant}$, 3) $D$ is related to $K_S$ via a negative exponential relationship ($R^2 = 0.74$) for dry $\theta_{ant}$ and 4) mTT is related to $K_S$ via a negative exponential relationship ($R^2 = 0.60$) for wet $\theta_{ant}$. Here, we would like to present the significant non-linear relationships we found between the shape parameters of the fitted TTDs and the flow path number $F$ ($R^2 = 0.90$) (Eq. 12 and

13), mainly because we can draw much more general conclusions on TTD shapes using a dimensionless number (Fig. 9):

$$Shape\ parameter\ \alpha(F) = 0.64|F|^{-0.20}, if\ K_S < 0.2\ md^{-1}, \tag{12}$$

$$Shape\ parameter\ D(F) = 1.27|F|^{0.36}, if\ K_S \geq 0.2\ md^{-1}. \tag{13}$$

Generally, for similar catchments with low $K_S$, Gamma distributions are more likely to fit the TTDs. The relatively higher proportion of surface flow within and surface outflow from these catchments seems to favor flow and transport dynamics

that are best represented by the shapes of Gamma distributions because they are able to capture both rapid response (high initial values) as well as the relatively slow outflow from the soils and the bedrock (long tails). In contrast, similar catchments with high $K_S$ and only small proportions of surface flow are more likely to behave according to Advection-Dispersion distributions with less rapid response from surface flow (low initial values) and faster outflow from the more conductive soils (shorter tails). A notable exception are scenarios where catchments with highly conductive soils still

experience larger proportions of surface outflow (> 25 %; $F > 0.05$) due to large amounts of $P_{sub}$ – these dynamics cannot be





predicted by the same relationship since they produce AD distributions with larger contributions of advective transport and hence smaller values of *D* (indicated by the black circle in Fig. 9).



**Figure 9: Relationship between the dimensionless flow path number *F* and the shape parameters α (upper panel, scenarios with low $K_S$) and *D* (lower panel, scenarios with high $K_S$) of the Gamma and the Advection-Dispersion distribution, respectively. Yellow colors indicate dry, green intermediate and blue wet antecedent moisture conditions; thick marker lines indicate large, mid-sized lines medium and thin lines small amounts of subsequent precipitation; solid lines indicate low, dashed lines high saturated hydraulic conductivities; lighter shades of a color indicate shallow, darker shades deep soils. The dotted trend lines are the best-fit**
**regressions for the relationship between the flow path number and the shape parameters α (light blue) and *D* (orange). The points in the black circle are excluded from the regression analysis since they are associated with scenarios of excessive surface outflow.**





Figure 10 gives an overview of the shape and scale of our modeled TTDs. Figure 11 shows how the shape and scale of TTDs change with the individual catchment and climate properties. For increasing $\theta_{ant}$, TTDs converge towards L-shaped distributions with short mTTs (in highly conductive soils the shape is more affected than the scale, in soils with low $K_S$ the scale is more affected than the shape). When $K_S$ is decreasing mTT is decreasing (in case $P_{sub}$ is big then the shapes of the TTDs also changes towards having lighter tails). Quite similar patterns can be observed for increasing $D_{soil}$ and decreasing $P_{sub}$ – with mTTs becoming longer and TTD shapes increasing the tail weight when $K_S$ is high and becoming more humped when $K_S$ is low.

**Figure 10: Gamma shape parameters ($\alpha$) and mean transit times (mTTs) for individual scenarios with different combinations of catchment and climate properties. Yellow colors indicate dry, green intermediate and blue wet antecedent moisture conditions; thick marker lines indicate large, mid-sized lines medium and thin lines small amounts of subsequent precipitation; solid lines indicate low, dashed lines high saturated hydraulic conductivities; lighter shades of a color indicate shallow, darker shades deep soils. The red boxes contain exemplary Gamma distributions with shape and scale corresponding to the red dot location.**





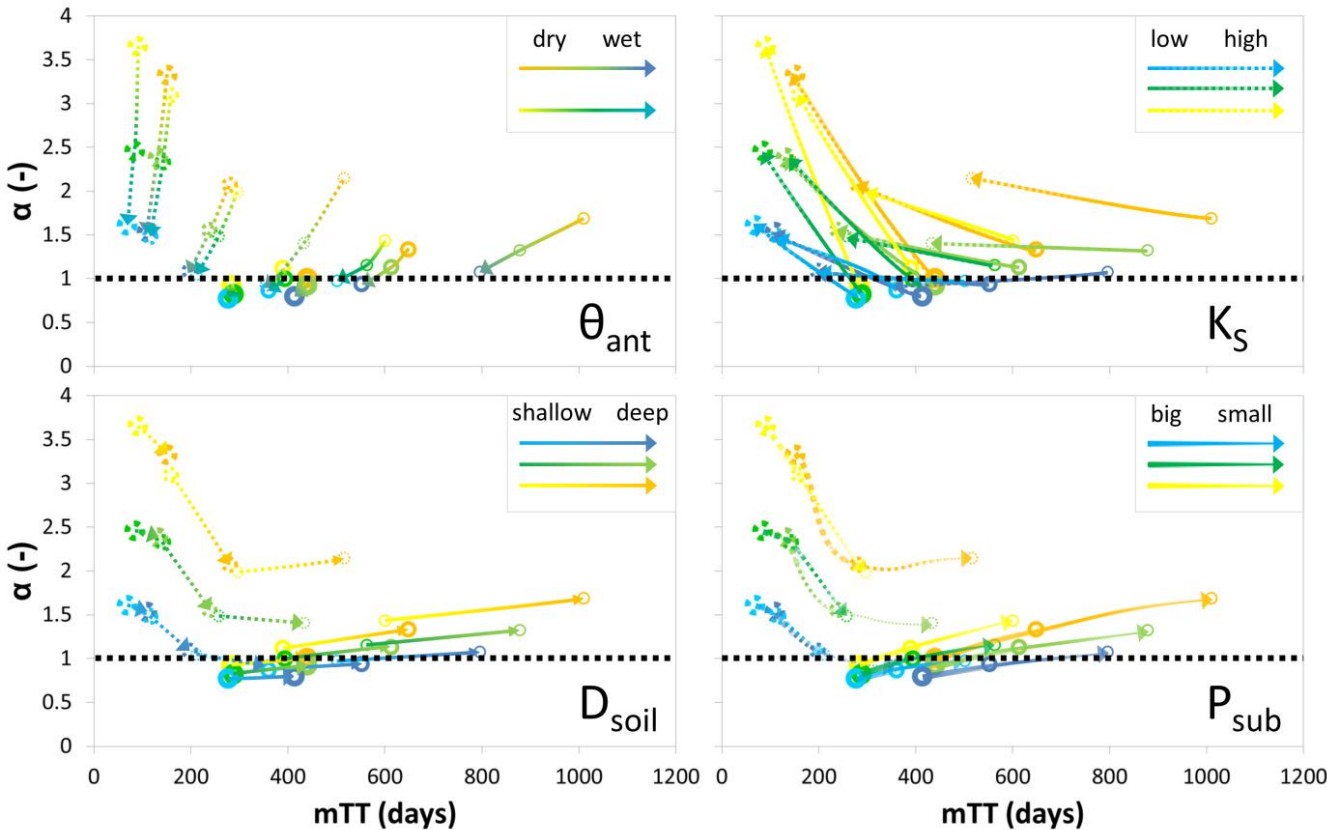

**Figure 11: Change of Gamma shape parameters (α) and mean transit times (mTTs) for four catchment and climate properties. Yellow colors indicate dry, green intermediate and blue wet antecedent moisture conditions; thick marker lines indicate large, mid-sized lines medium and thin lines small amounts of subsequent precipitation; solid lines indicate low, dashed lines high saturated hydraulic conductivities; lighter shades of a color indicate shallow, darker shades deep soils.**

### 3.6.    Effects of other factors on the shape of TTDs





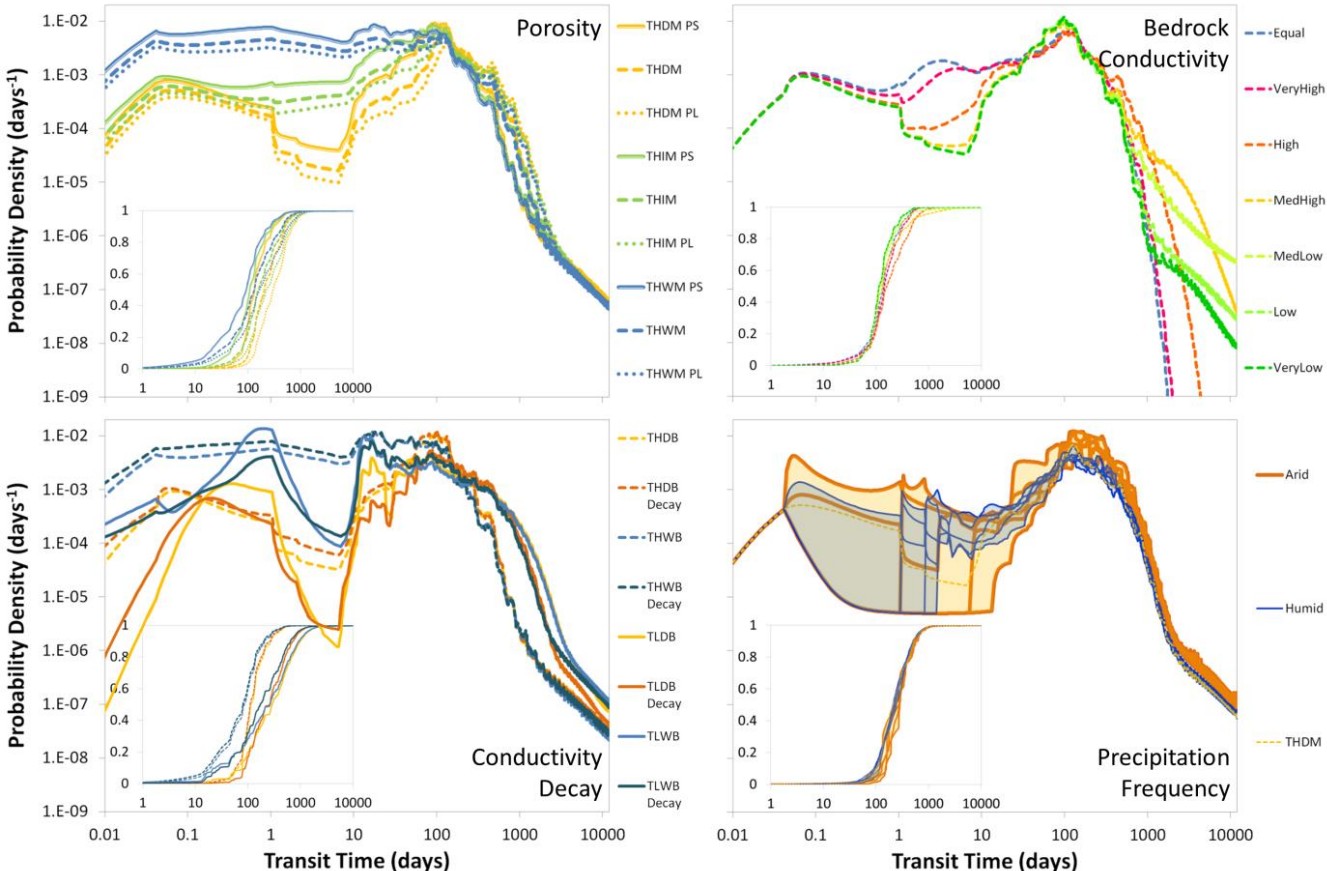

**Figure 12: Overview of how certain catchment and climate characteristics influence the shape of TTDs. 1. Porosity – solid lines indicate small, dotted lines large porosity. 2. Hydraulic conductivity of the bedrock. 3. Decay in saturated soil hydraulic conductivity with depth – darker shades of one color represent scenarios with decay, lighter shades scenarios without decay. 4. Precipitation frequency – orange TTDs are low-frequency ("arid type") scenarios, blue TTDs are high-frequency ("humid type") scenarios. The shaded areas between the lines illustrate the higher shape variability for the low-frequency TTDs. Insets show cumulative TTDs.**

### 3.6.1. Porosity

The influence that soil porosity exerts on the shape of TTDs is quite similar to the influence of $D_{soil}$. Larger soil porosity causes a dampening of the initial response and increasing transit times in all parts of the TTD (just like deeper soils, see Fig. 12 and Table 4). Increasing porosity also causes larger standard deviations, smaller skewness and smaller kurtosis (i.e. less peaked TTDs).

### 3.6.2. Hydraulic conductivity of the bedrock

Variations in the saturated hydraulic conductivity of the bedrock $K_{Br}$ affect the shape of TTDs both in the initial part of the distributions but even more so in the tail (Fig. 12 and Table 5). If $K_{Br}$ is increased so that it equals the $K_S$ of the soil layer, we





basically create one large continuum of homogeneous bedrock (or soil). Hence, the resulting TTD does not contain any
abrupt breaks in slope and basically resembles outflow from a larger homogeneous reservoir. For lower $K_{Br}$ breaks in the
slope of the TTD tails start to appear indicating that the soil layers have already been emptied while the bedrock still
contains water from the input precipitation event. For scenarios where $K_{Br}$ is at least 3 orders of magnitude smaller than the
soil $K_S$, the tails initially resemble power law distributions with constants ($a$) around 0.2 and exponents ($k$) around 1.6 for
longer periods of time (Eq. 14):

$$TTD(t) = at^{-k}.$$                                                                                   (14)

An exponent $k$ smaller than 2 indicates that a mean value of the power law distribution cannot be defined since it is basically
infinite, however, in our simulation results, the power law tails eventually break down when the bedrock domain is almost
empty. Somewhat counterintuitively, the scenario with the lowest $K_{Br}$ exhibits the shortest quartile and mean transit times.
This is clearly an effect of a smaller fraction of water infiltrating into the bedrock and more water being transported laterally
in the relatively conductive soil layer. We observe the longest quartile transit times in the scenario where $K_{Br}$ is one order of
magnitude lower than $K_S$ and the longest mean transit time when it is 2 orders of magnitude lower. This is due to the fact that
for these cases the higher $K_{Br}$ causing faster transport within the bedrock is counterbalanced by the larger fraction of event
water that enters into the bedrock where it is transported more slowly than in the soil. Therefore what seems paradoxical in
the first place – longer mTTs when $K_{Br}$ is higher – can be explained by differences in the runoff partitioning between soil and
bedrock. This also explains the observation that the standard deviation of the TTDs initially increases with increasing $K_{Br}$
while both skewness and excess kurtosis decrease.

### 3.6.3.    Decay in saturated hydraulic conductivity with depth

For catchments that already have highly conducting soils, adding a decay in $K_S$ with higher $K_S$ close to the surface and lower
$K_S$ close to the soil–bedrock interface does not change the shape of TTDs to a great extent – all fitted shape parameters
remain rather similar and transit times across the entire TTD are moderately shortened (Fig. 12 and Table 6). We observe a
larger impact if soil $K_S$ is low. In these cases adding a decay reduces the standard deviation and increases the skewness and
the kurtosis of the resulting TTDs (i.e. they become narrower, more skewed and more peaked). Additionally, the difference
in transit times increases towards the late part of the TTD with mTT and $Q_3$ being considerably shorter when there is a decay
in $K_S$. This difference between the smaller effects of a $K_S$ decay in an already highly conductive soil compared to the larger
effects for a low conductivity soil can be explained by the fact that the additional soil zones of higher conductivity are more
effectively used for scenarios of generally low conductivity – in soils that are already quite conductive, a larger fraction of
the incoming event water will still infiltrate to deeper soil layers before moving laterally whereas in low conductivity soils
the faster lateral transport possible due to the $K_S$ decay will be triggered much sooner and for a larger fraction of the
incoming event water.



### 3.6.4. Precipitation frequency

The shape of TTDs is not influenced significantly by precipitation frequency since the mean values of all distribution metrics for the low-frequency (arid type) and the high-frequency (humid type) scenarios are quite similar to each other (Fig. 12 and Table 7). However, transit times in the high-frequency (humid) environment are shorter ($Q_1 = -17$ %, $Q_2 = -11$ %, mTT = $-9$ %, $Q_3 = -3$ %). Additionally, the higher the precipitation frequency the lower is the variation between individual TTDs. This is mainly due to two facts: When the precipitation frequency is high 1) the interarrival times are shorter which will more often mobilize event water and avoid longer periods of relative inactivity when the water "just sits" in the soil, 2) the amounts of precipitation events are on average smaller so that there is a smaller chance of a very big event "flushing" the entire system creating very short transit times for a preceding event followed by a long period of no or only small precipitation events. These transit time dynamics with regard to different patterns of precipitation have already been observed in the field (Heidbüchel et al., 2013).

### 3.6.5. Water retention curve

The TTDs from the scenarios with sand-type WRCs have higher initial peaks and lighter tails compared to the ones with silt-type WRCs (Fig. 13). Their transit times are consistently shorter over the entire distributions and the influence of other parameters (like $K_S$ and $\theta_{ant}$) on their shape is reduced. Sand-type TTDs are more skewed and more peaked than silt-type TTDs (Table 8). They more closely resemble TTDs that we would expect in environments where preferential flow is present. Generally, the differences in TTDs between the different WRCs are more pronounced in the scenarios with low $K_S$ because the wetting of the upper soil layers and hence the increase in the hydraulic conductivity takes relatively more time such that the differences between the two WRC scenarios are amplified. In the scenarios with silt-type WRCs the saturation process causes a slower increase in hydraulic conductivity since soil water potential decreases more gently with increasing soil water content.

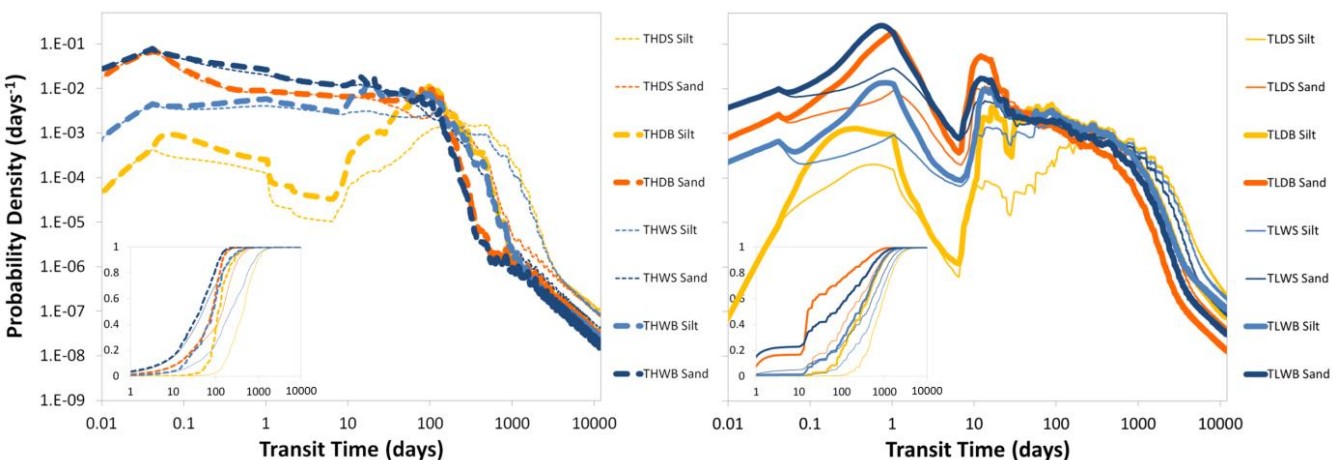





**Figure 13: Changes in TTDs due to differences in water retention curves (WRCs). Left panel: scenarios with high $K_S$, right panel: scenarios with low $K_S$. Light blue and yellow lines indicate silt-type soil WRCs, dark blue and orange lines indicate sand-type soil**
**WRCs. Insets show the cumulative TTDs.**

### 3.6.6.    Full saturation and extreme precipitation

Starting runs with fully saturated soils increased the fractions of overland flow for both the high and the low $K_S$ scenario (THSB and TLSB). For THSB the fraction of outflow during the first 10 days that was overland outflow ($SOF_{10}$) increased from 1 to 9 %. For TLSB the increase was even higher from 76 to 91 %. The increase had clear effects on the resulting
transit times. Especially the short transit times decreased while the longer transit times were less affected. That means the changes we observed in the shape of the TTDs followed the pattern of increasing $\theta_{ant}$ (i.e. a higher percentage of transit time decrease in the young fraction of the TTD, smaller impact at later times and in the shape metrics). Increasing the precipitation amount and intensity of the input event by a factor of 100 (+; from 0.1 to 10 mm h$^{-1}$) affected only the low $K_S$ scenario (TLSB+) further decreasing the short transit times while the high $K_S$ scenario was unaffected (THSB+). We had to
increase the precipitation intensity of the input event by a factor of 1000 (to 100 mm h$^{-1}$) to eventually create substantial amounts of initial overland flow for both scenarios. Once this was triggered, the shape of the TTDs changed considerably. For these scenario (THSB+++ and TLSB+++), all quartiles of the TTDs decreased to less than one day and the whole distribution became extremely leptokurtic (Fig. 14 and Table 9).





**Figure 14: Full saturation and extreme precipitation – black lines indicate fully saturated initial conditions, pink lines fully saturated initial conditions and very large event precipitation (+), red lines fully saturated initial conditions and extreme event precipitation (+++). The horizontal lines in the box above the diagram indicate periods where actual overland flow was recorded during the respective runs. The inset shows the cumulative TTDs.**

## 4. Discussion

### 4.1. Use of theoretical distributions

None of the theoretical distribution functions we tested captures the shape of the observed TTDs adequately over the entire age range. On the one hand, this is due to the missing power law tails, on the other hand – and this is more relevant from a mass balance perspective – it results from a misrepresentation of the initial response. Looking at Fig. 7, 8 and 12 to 14, it becomes clear that all TTDs are humped distributions, with none of them having an initial maximum (with a monotonically decreasing limb afterwards) and none of them having a value of 0 after 1.5 minutes (the first time step reported). Since all AD distributions start with a value of 0 and all Beta and Gamma distributions are either monotonically decreasing or start with a value of 0 they are not perfect representations of the modelled TTDs for porous media. A set of theoretical probability





distributions – with initial values larger than 0, a rising limb to a maximum probability density and a falling limb with lighter
or heavier tails – would be the best option to represent variable TTDs. Potential candidates for these theoretical distributions
are truncated Gamma or lognormal distributions. The fact that TTDs in highly conductive soils and in dry antecedent
conditions are better represented by the AD model can be explained by the fact that the catchment storage has to be filled at
least a little bit before faster flow paths are activated and substantial flow out of the system can occur. This means that the
early response is much better captured by a distribution that starts with an initial value of 0. Furthermore, the high $K_S$
produces tails that are lighter than the ones of Gamma distributions and more closely related to the AD model. Contrary to
that, low $K_S$ values and wet antecedent conditions favor Gamma distributions because initial values are generally higher
when the soil is closer to saturation while tails are heavier in soils that are less conductive.

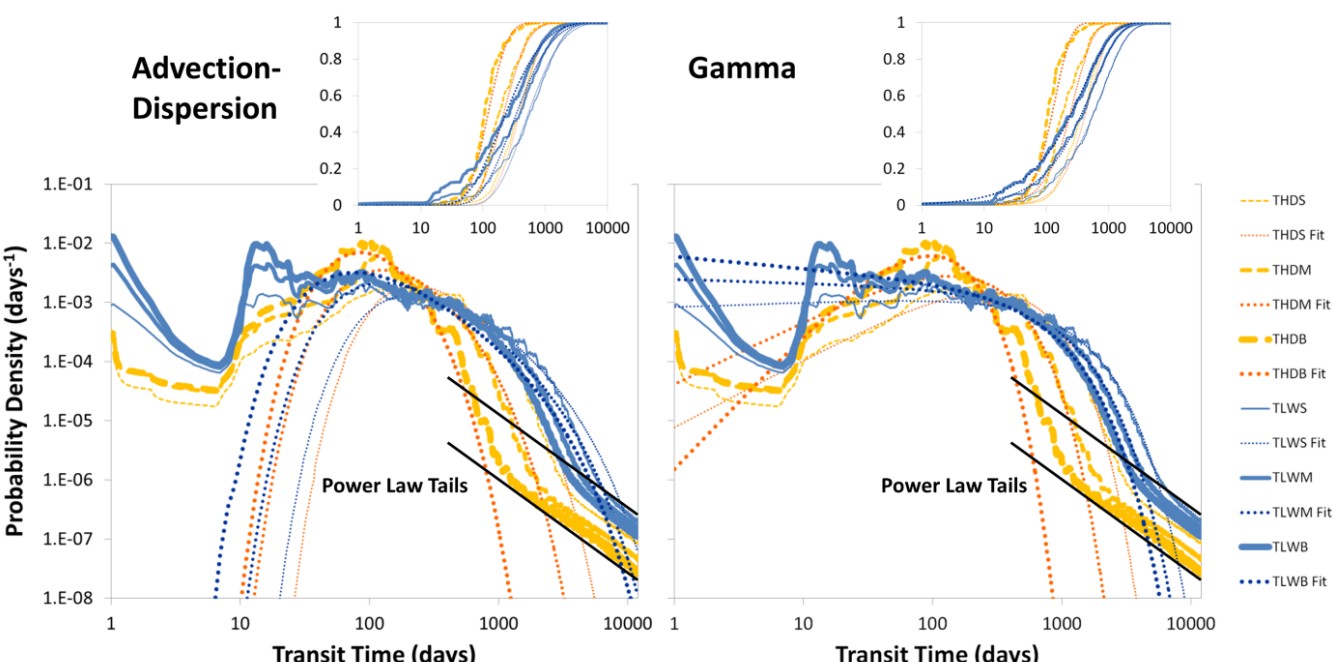

**Figure 15: Modeled TTDs for low $K_S$, high $\theta_{ant}$ (blue) and high $K_S$, low $\theta_{ant}$ (yellow). Best-fit theoretical distributions (dotted lines) for the individual scenarios for the Advection-Dispersion model (left panels) and the Gamma model (right panels). Power law tails of the modeled TTDs are marked by the solid black lines. Small panels show cumulative TTDs.**

None of the theoretical distributions was able to reproduce the TTD tails we observed in the model output that initially
follow a power law. This, however, did not constitute a substantial problem with regard to the correct mass balance since
these power law tails only comprise a very small fraction of the mass that was added to the system as a tracer. Still, if the
tailing of the TTDs is relevant to a problem (e.g. when dealing with legacy contamination) one can add the observed power
law tails to the distributions (for a description see Text S5 and Fig. S9 in the supplement). Although the Beta model
performed best overall (by a slim margin), we do not recommend using it due to its additional fitting parameter (the upper





limit c) which increases equifinality problems (that we set out to eliminate). Therefore we recommend utilizing the AD
model for high $K_S$ and the Gamma model for low $K_S$ scenarios – but only taking the median (and not the mean) as a reliable
transit time estimate.

### 4.2. Connection between the shape of TTDs and the flow path number $F$

We can pretty accurately predict the shape of a TTD within the parameter range of our model scenarios using $F$ alone (Fig.
9). Linked to that, some interesting conclusions can be drawn from the identified relationships between $F$ and the shape
parameters $\alpha$ and $D$. 1. A flow path number between $-1$ and $+1$ characterizes catchments where the available storage is larger
than the change in storage caused by the incoming and outgoing flows. 2. If the system receives more water than it can
remove (inflow-dominated), $F$ is positive and the shape of TTDs is generally better represented by Gamma distributions. 3.
With increasing $F$, $\alpha$ decreases to values below 1. This decrease in the shape parameter $\alpha$ is mainly caused by the initial
peaks of the TTDs becoming higher while the tails remain rather similar. 4. If the system has the capacity to remove more
water in the subsurface than it receives (outflow-dominated), $F$ becomes negative and the shape of the TTDs is generally
better represented by AD distributions. 5. When $F$ becomes more negative, $D$ increases from values below 0.5 to values
above 0.5 indicating higher peaks. 6. $F$ converges towards 0 for systems with increasing available storage (because the
denominator keeps increasing) or if inflow and outflow capacity are evenly balanced. For these cases both Gamma and AD
distributions become more and more dominated by advection and less by dispersion (while their initial peaks decrease).

Our simulation results suggest that the tails of the Gamma TTDs become lighter with increasing $F$ values. Therefore $\alpha$
should increase with increasing $F$ values. The circumstance that we find a better relationship between increasing $F$ and
decreasing $\alpha$ values is due to the fact that the change in the initial response (higher initial values and peaks) outweighs the
tails becoming lighter in the total mass balance. The same logic applies to the AD distributions where $D$ becomes larger with
more negative $F$ values.

Instead of using TTDs with constant shapes for determining variable transit times with transfer function-convolution models,
one can use these relationships to pre-define the TTD shapes – reducing the problem of equifinality that stems from the
simultaneous determination of shape and scale parameters (Fig. 16).



**Figure 16: Predicted TTD shapes based on their relationship to the flow path number *F*, resulting from different antecedent moisture conditions $\theta_{ant}$ (yellow – dry, blue – wet) and subsequent precipitation amounts $P_{sub}$. TTDs for low $K_S$ are Gamma distributions (middle panel), for high $K_S$ they are AD distributions (lower panel). Individual TTDs start with time shifts so that they do not overlap (individual start times correspond to the $P_{sub}$ markers in the upper panel).**

Again, we would like to point out that our results can be considered valid for systems that do not experience a large fraction

of preferential flow in the soil and bedrock since we only model flow taking place in the porous matrix of the subsurface

domain. This is the likely reason that we also encounter $\alpha$ values that are larger than 1 – although such high $\alpha$ values were





not found in previous studies (Hrachowitz et al., 2009; Godsey et al., 2010; Berghuijs and Kirchner, 2017; Birkel et al.,
2016). Therefore, in terms of expanding the modeling effort, it would be very beneficial to include both evapotranspiration
and macropore flow into the simulations. An inclusion of these processes will shift the flow path number $F$ to more negative
values. On the one hand, evapotranspiration will provide an additional way to remove water from the subsurface
(representing another sink term similar to $K_{rem}$) and macropore flow will enhance the subsurface outflow capacity of the
catchment. This could result in a shift towards TTDs with higher initial peaks. On the other hand, evapotranspiration also has
the potential of reducing $\theta_{ant}$ below moisture levels obtainable with free drainage alone. This more extreme dryness could
lead to even more humped TTDs with initial values closer to 0. The inclusion of additional heterogeneity in soil properties
(layering, small-scale variations) would also be a worthwhile exercise that is, however, out of the scope of our study.

The theoretical framework around the flow path number $F$ could also be used to assess the impact that other catchment and
climate properties have on TTD shapes. For example catchment size would only have an impact on TTD shape if the cross-
sectional area of the outflow boundary $A_{out}$ changed disproportionately. If, e.g., the catchment area $A_{in}$ increased but the
cross-sectional area $A_{out}$ remained the same, then the subsurface outflow capacity $K_{rem}$ would decrease and hence $F$ would
change.

### 4.3.  Replacing transit time with cumulative outflow

For certain scenarios we still see differences in the probability distributions if we replace transit time with cumulative
outflow (see Fig. S8 in the supplement). This observation can be explained by the fact that for the high $K_S$ scenarios (where
differences are reduced) we only generate external flow variability while for the low $K_S$ scenarios (where differences remain)
we also cause internal flow variability (Kim et al, 2016). That means that in the high $K_S$ scenarios an increase in $P_{sub}$
increases the flow in all of the available flow paths proportionally (without changing the flow path partitioning or activating
previously unused flow paths) while for the low $K_S$ scenarios an increase in $P_{sub}$ causes pronounced shifts in the flow path
partitioning where the additional amount of precipitation can bypass the subsurface flow paths by predominantly utilizing
overland flow paths (leading to the observation that a larger amount of $P_{sub}$ is necessary to flush out an equal amount of
tracer). This can serve as direct proof that replacing transit time with cumulative outflow does not erase all differences
between TTDs.

### 5.  Conclusion

In our simulations for a virtual low-order catchment we observed that the shape of TTDs changes systematically with the
four investigated catchment and climate properties ($D_{soil}$, $K_S$, $\theta_{ant}$ and $P_{sub}$) so that it is possible to predict the change using
the dimensionless flow path number $F$. The results can be summarized in three main conclusions (see also Fig. 9): 1) The
shape of TTDs converges towards L-shaped distributions with high initial values if a catchment's capacity to store inflow
decreases or if the actual inflow to a catchment does not equal its subsurface outflow capacity. 2) For catchments with low





$K_S$ values, Gamma distributions are generally better representations of the TTDs (due to the heavier tails associated with lower $K_S$); for catchments with high $K_S$ values, AD distributions work better (due to the lighter tails). 3) Heavier tails are observed when the system is in a more "relaxed" state where all potential flow paths (deep and shallow, slower and faster) are equally used for transport. This is generally the case if $P_{sub}$ is relatively small. Lighter tails appear when the system is in a more "stressed" state where the shallow and faster flow paths are disproportionally used for transport. This can be associated with larger $P_{sub}$ values. Moreover, power law tails emerge if there is a sufficiently large difference in hydraulic conductivity between the bedrock $K_{Br}$ and the soil $K_S$.

According to our findings, $D_{soil}$ has only little influence on TTD shape and is linearly related to the mTT. That means that in catchments with deeper soils we should expect longer transport times but the same relation of solute advection to solute dispersion as in catchments with shallower soils. High $K_S$ values are associated with TTDs that have higher initial values and lighter tails while $K_S$ and mTT are related via a negative power-law relationship. The influence of $K_S$ increases for wet $\theta_{ant}$ (especially for short transit times) and for large $P_{sub}$ (especially for long transit times) since both maximize the differences in hydraulic conductivity between catchments – the drier the conditions the more similar are the unsaturated hydraulic conductivities generally. In locations with higher precipitation amounts TTDs will have lighter tails and shorter mTTs (there is a power law relationship between $P_{sub}$ and mTT) mainly due to the fact that a larger $P_{sub}$ flushes the soils faster and only allows a smaller fraction of the precipitation events to infiltrate into the bedrock. The influence of $P_{sub}$ is larger for dry $\theta_{ant}$ and high $K_S$ (especially for the longer transit times). Long-term trends or interannual changes in $P_{sub}$ can cause temporal variations in TTDs but substantial short-term temporal variations in TTDs are derived mainly from differences in $\theta_{ant}$: While under dry $\theta_{ant}$ there is a lower probability for shorter transit times, wet $\theta_{ant}$ triggers faster responses and hence higher initial peaks. Also, there is a negative linear relationship between mTT and $\theta_{ant}$. The influence of $\theta_{ant}$ is stronger for catchments with higher $K_S$ and for climates with smaller $P_{sub}$. Due to the changes in $\theta_{ant}$, variations in TTD shape and scale can be high even in relatively small catchments. The influence of precipitation frequency on the shape of TTDs is detectable but relatively minor, however changes in the sequence of subsequent precipitation events can be relevant in regions with a low precipitation frequency. The fraction of water entering the bedrock depends strongly on the contact time of that water with the soil–bedrock interface. That means that in regions with small $P_{sub}$ a larger fraction of precipitation has the chance to infiltrate into the bedrock before it is flushed out of the soil layer by subsequent precipitation. Therefore the tails of TTDs in more arid regions tend to be heavier than the TTD tails in humid regions.

Gamma functions were able to capture the time-variance of TTDs in an appropriate way, especially for low $K_S$ scenarios and wet antecedent soil moisture conditions, while AD distributions worked well for high $K_S$ scenarios and dry antecedent conditions. However, none of the theoretical distributions described the early part of the distributions with non-zero initial values combined with a mode shortly after (i.e. the humped form) that is observed in most cases. Moreover, we observed the general pattern that TTDs with high initial values tend to have lighter tails than TTDs with low initial values. Gamma distributions, unfortunately, exhibit the opposite behavior (with high initial values being associated with heavier tails than low initial values; see Fig. 17). Based on the results from our modelling efforts, we therefore encourage the search for a set





of better fitting theoretical distributions. These distributions should be able to a) represent high initial values paired with lighter tails as well as low initial values paired with heavier tails and b) take on a "humped" form with non-zero initial values. Concerning the TTD metrics, in most cases the median transit time was much better predicted by the theoretical distributions than the mean.




**Figure 17:** Gamma distributions (solid lines) capture the middle part of the modeled TTDs (dashed lines; thickness corresponds to $P_{sub}$ amount) quite well but do not correctly represent their initial parts and power law tails. Inset: Gamma distributions (thick and thin black lines) combine either high initial values with heavier tails or zero initial values with lighter tails while modeled TTDs often are best described by high initial values and lighter tails (blue dashed line) or low (albeit non-zero) initial values with heavier tails (yellow dashed line).

### 5.1. Outlook

It is quite unlikely that we can predict the shape of real-world TTDs with the relationship between $F$ and $\alpha$ that we found in our virtual experiments because we did not consider some (probably) very important processes – like evapotranspiration and





macropore flow. The TTDs we derived are based on surface flow coupled with subsurface flow in a porous matrix. Therefore certain transport and mixing processes related to preferential flow are not included in this analysis. However, the relationships we find can illuminate essential dynamics in catchment hydrology and help forming the basis for further investigations that include additional hydrologic processes. It will be very interesting to see how, e.g., the introduction of
evapotranspiration will modify the relationship between $F$ and $\alpha$. Moreover, these experiments can be repeated with other potentially more appropriate theoretical probability distributions in the future.

An interesting question that remains is whether backward TTDs can be linked to catchment and climate properties in a similar fashion to the one we used here, since backward TTDs are comprised of many individual water inputs that entered the catchment over a very long period of time with potentially greatly varying initial conditions. That leads to the question of
whether it is more important to know the conditions at the time of entry to the catchment or the conditions at the time of exit from the catchment (or both) in order to make predictions about TTD shapes and mTTs. Remondi et al. (2018) were the first to tackle this problem by water flux tracking with a distributed model. They found that mainly soil saturation and groundwater storage affected backward TTDs.

Practical implications can be drawn from these results concerning, e.g., pollution events. Some catchments are more
vulnerable to pollution in the sense that they tend to store pollutants for a longer period of time and hence exhibit long legacy effects. Especially catchments with TTDs with heavy tails belong in that category (i.e. catchments with deeper soils and a moderate hydraulic conductivity difference between soil and bedrock). Also, certain points in time are worse for pollution events to happen – a spill occurring during dry conditions will stay in the catchment longer because it is more likely to reach the bedrock and stay in contact with it before it is flushed out of the soils than a spill during wet conditions. Accordingly,
locations and situations that lead to a longer storage of decaying pollutants will eventually release less of the solutes to the downstream rivers. Further theoretical developments could include the use of TTDs for non-conservative solute transport. This could be achieved by considering the TTD shape a basic function to which different reaction terms can be added (like "cutting the tail" of solutes that decay after a certain time in the catchment or shifting, damping and extending the TTD for solutes that experience retardation). An example is provided for an exponential decay reaction in Text S6 and Fig. S10 in the
supplement.

Finally, this research can also contribute to the field of catchment evolution. One could argue that positive flow path numbers are not sustainable over longer periods of time because that would mean that the subsurface outflow capacity of the (zero-order) catchment is permanently insufficient and the catchment is not capable of efficiently discharging all of the incoming precipitation in the subsurface. Consequently, the catchment storage would be filled up completely and overland
flow would be occurring on a regular basis. Since widespread overland flow is rarely observed in most catchments it could be argued that most catchments have already evolved towards negative flow path numbers (e.g. by increasing $K_S$ or $D_{soil}$). That, in turn, could also mean that L-shaped (or initially slightly humped) TTDs with heavier tails and Gamma shape parameters $\alpha$ around 0.5 are the natural endpoint of catchment evolution.



Ideally, this work will help to generate new or to expand existing hypotheses on hydrologic and hydrochemical catchment
response that can be tested in future field experiments.

**Data availability**

All data used in this study is presented either in the main manuscript or in the supplement.

**Author contribution**

Conceptualization, I.H., P.T., and T.F.; Formal Analysis, I.H.; Funding Acquisition, J.F.; Investigation, I.H., A.M., J.Y., and
J.F.; Software, J.Y.; Writing – Original Draft, I.H.; Writing – Review & Editing, I.H., A.M., J.F., J.Y., P.T., and T.F.

**Competing interests**

The authors declare that they have no conflict of interest.

**Acknowledgments**

This research was supported by the Helmholtz Research Programme "Terrestrial Environment", topic 3: "Sustainable Water
Resources Management", with the integrated project: "Water and Matter Flux Dynamics in Catchments". We would like to
thank Carlotta Scudeler for her guidance on hydrologic modeling and her contribution to a previous version of this
manuscript. Thanks also to René Therrien for his help with the HGS modeling. Finally, we would like to acknowledge
Stefanie Lutz for an excellent discussion of the manuscript.

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




**1030  Tables**

**Table 1: Metrics of the TTDs derived from the modeling of 36 scenarios with different combinations of catchment and climate properties. All times are given in days.**

| D_soil | DEEP (THICK) | | | | | | | | | | | | | | | | | |
|---|---|---|---|---|---|---|---|---|---|---|---|---|---|---|---|---|---|
| K_S | HIGH | | | | | | | | | LOW | | | | | | | | |
| θ_ant | DRY | | | INT | | | WET | | | DRY | | | INT | | | WET | | |
| P_sub | SMALL | MED | BIG | SMALL | MED | BIG | SMALL | MED | BIG | SMALL | MED | BIG | SMALL | MED | BIG | SMALL | MED | BIG |
| Name | THDS | THDM | THDB | THIS | THIM | THIB | THWS | THWM | THWB | TLDS | TLDM | TLDB | TLIS | TLIM | TLIB | TLWS | TLWM | TLWB |
| 1st Quartile | 244 | 137 | 89 | 159 | 105 | 66 | 101 | 67 | 45 | 458 | 214 | 126 | 312 | 191 | 111 | 232 | 135 | 94 |
| Median | 441 | 207 | 115 | 315 | 159 | 101 | 218 | 132 | 85 | 785 | 475 | 291 | 640 | 456 | 289 | 565 | 394 | 269 |
| Mean | 515 | 280 | 151 | 433 | 238 | 132 | 354 | 197 | 110 | 1009 | 648 | 439 | 878 | 613 | 439 | 796 | 552 | 413 |
| 3rd Quartile | 656 | 366 | 167 | 569 | 299 | 143 | 501 | 258 | 136 | 1308 | 862 | 576 | 1191 | 832 | 576 | 1116 | 778 | 561 |
| Stand Dev | 455 | 298 | 189 | 454 | 285 | 190 | 443 | 275 | 173 | 880 | 646 | 505 | 881 | 700 | 587 | 816 | 635 | 530 |
| Skewness | 7 | 15 | 28 | 7 | 14 | 28 | 7 | 15 | 29 | 3 | 4 | 5 | 4 | 5 | 7 | 3 | 4 | 6 |
| Exc Kurtosis | 125 | 407 | 1233 | 117 | 404 | 1214 | 123 | 437 | 1426 | 20 | 41 | 70 | 27 | 56 | 94 | 22 | 46 | 80 |
| D_soil | SHALLOW (FLAT) | | | | | | | | | | | | | | | | | |
| Name | FHDS | FHDM | FHDB | FHIS | FHIM | FHIB | FHWS | FHWM | FHWB | FLDS | FLDM | FLDB | FLIS | FLIM | FLIB | FLWS | FLWM | FLWB |
| 1st Quartile | 139 | 91 | 49 | 107 | 70 | 44 | 72 | 46 | 22 | 211 | 127 | 80 | 173 | 109 | 77 | 135 | 94 | 62 |
| Median | 212 | 120 | 79 | 165 | 104 | 63 | 136 | 88 | 49 | 458 | 269 | 163 | 413 | 266 | 158 | 342 | 204 | 146 |
| Mean | 296 | 159 | 90 | 257 | 142 | 84 | 211 | 116 | 68 | 600 | 389 | 284 | 563 | 394 | 288 | 501 | 360 | 277 |
| 3rd Quartile | 389 | 174 | 106 | 312 | 147 | 97 | 272 | 136 | 90 | 796 | 504 | 389 | 750 | 504 | 385 | 656 | 474 | 378 |
| Stand Dev | 357 | 231 | 154 | 372 | 258 | 208 | 338 | 219 | 157 | 619 | 461 | 377 | 713 | 588 | 505 | 660 | 557 | 492 |
| Skewness | 14 | 25 | 41 | 14 | 23 | 31 | 14 | 26 | 41 | 5 | 7 | 9 | 7 | 9 | 11 | 6 | 9 | 10 |
| Exc Kurtosis | 332 | 903 | 2245 | 297 | 742 | 1274 | 345 | 998 | 2199 | 59 | 109 | 169 | 70 | 119 | 174 | 73 | 121 | 170 |

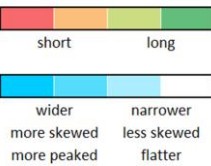

short — long

wider / more skewed / more peaked — narrower / less skewed / flatter

**Table 2: Shape parameters of the best-fit Gamma (α) and Advection-Dispersion (D) distributions and associated flow path 1035 numbers (F) for the 36 different scenarios.**

| D_soil | DEEP (THICK) | | | | | | | | | | | | | | | | | |
|---|---|---|---|---|---|---|---|---|---|---|---|---|---|---|---|---|---|---|
| K_S | HIGH | | | | | | | | | LOW | | | | | | | | |
| θ_ant | DRY | | | INT | | | WET | | | DRY | | | INT | | | WET | | |
| P_sub | SMALL | MED | BIG | SMALL | MED | BIG | SMALL | MED | BIG | SMALL | MED | BIG | SMALL | MED | BIG | SMALL | MED | BIG |
| Name | THDS | THDM | THDB | THIS | THIM | THIB | THWS | THWM | THWB | TLDS | TLDM | TLDB | TLIS | TLIM | TLIB | TLWS | TLWM | TLWB |
| α | 2.15 | 2.07 | 3.33 | 1.41 | 1.55 | 2.38 | 0.92 | 1.09 | 1.53 | 1.69 | 1.33 | 1.01 | 1.32 | 1.13 | 0.92 | 1.08 | 0.94 | 0.80 |
| D | 0.28 | 0.29 | 0.18 | 0.46 | 0.40 | 0.25 | 0.75 | 0.60 | 0.41 | 0.37 | 0.49 | 0.69 | 0.51 | 0.62 | 0.79 | 0.62 | 0.74 | 0.90 |
| F | -0.04 | -0.02 | 0.01 | -0.07 | -0.04 | 0.01 | -0.22 | -0.13 | 0.01 | 0.01 | 0.03 | 0.06 | 0.02 | 0.05 | 0.10 | 0.08 | 0.16 | 0.32 |
| D_soil | SHALLOW (FLAT) | | | | | | | | | | | | | | | | | |
| Name | FHDS | FHDM | FHDB | FHIS | FHIM | FHIB | FHWS | FHWM | FHWB | FLDS | FLDM | FLDB | FLIS | FLIM | FLIB | FLWS | FLWM | FLWB |
| α | 1.99 | 3.09 | 3.66 | 1.49 | 2.33 | 2.46 | 1.05 | 1.48 | 1.61 | 1.43 | 1.12 | 0.92 | 1.16 | 1.00 | 0.82 | 0.97 | 0.87 | 0.78 |
| D | 0.30 | 0.19 | 0.15 | 0.43 | 0.27 | 0.24 | 0.65 | 0.43 | 0.39 | 0.45 | 0.61 | 0.77 | 0.63 | 0.74 | 0.92 | 0.74 | 0.85 | 0.98 |
| F | -0.02 | 0.01 | 0.07 | -0.04 | 0.01 | 0.11 | -0.13 | 0.03 | 0.35 | 0.03 | 0.06 | 0.12 | 0.05 | 0.10 | 0.20 | 0.16 | 0.31 | 0.63 |

TTD shape: I-shaped / heavy-tailed — humped / light-tailed

TTD scale: short — long

F dominated by: outflow / storage / inflow

**Table 3: Deviations of mean (green) and median (blue) transit times between the best-fit theoretical probability distributions and the modeled TTDs. Sum of the squared residuals (yellow) indicating goodness of fit between theoretical probability distribution and modeled TTDs.**





| $D_{soil}$ | | | | | | DEEP (THICK) | | | | | | | | | | | |
| $K_S$ | | | HIGH | | | | | | | | | LOW | | | | | |
| $\theta_{ant}$ | DRY | | | INT | | | WET | | | DRY | | | INT | | | WET | | |
| $P_{sub}$ | SMALL | MED | BIG | SMALL | MED | BIG | SMALL | MED | BIG | SMALL | MED | BIG | SMALL | MED | BIG | SMALL | MED | BIG |
| Name | THDS | THDM | THDB | THIS | THIM | THIB | THWS | THWM | THWB | TLDS | TLDM | TLDB | TLIS | TLIM | TLIB | TLWS | TLWM | TLWB |
| AD Δ Mean | 6 | -4 | -9 | 12 | -2 | -6 | 21 | 4 | -1 | 31 | 25 | 22 | 102 | 44 | 32 | 60 | 35 | 18 |
| Beta Δ Mean | -14 | -14 | -14 | -13 | -14 | -12 | -6 | -10 | -8 | -19 | -13 | -8 | 38 | -2 | -6 | 6 | -5 | -15 |
| Gamma Δ Mean | -282 | -152 | -109 | -132 | -94 | -81 | 26 | -25 | -42 | -423 | -172 | -10 | -186 | -74 | 30 | -52 | 31 | 84 |
| AD Δ Median | -32 | 7 | 6 | -6 | 11 | -1 | 1 | -6 | -8 | -22 | -19 | -13 | 17 | -44 | -21 | -28 | -50 | -37 |
| Beta Δ Median | -15 | 17 | 8 | 13 | 21 | 2 | 17 | 2 | -4 | 18 | 10 | 8 | 59 | -13 | 1 | 6 | -26 | -20 |
| Gamma Δ Median | -15 | 17 | 8 | 12 | 20 | 2 | 17 | 2 | -4 | 18 | 10 | 8 | 59 | -13 | 1 | 6 | -26 | -20 |
| AD Fit | 0.44 | 0.32 | 0.33 | 0.68 | 0.22 | 0.19 | 1.20 | 0.31 | 0.30 | 0.51 | 0.92 | 1.10 | 1.78 | 1.80 | 1.65 | 2.63 | 2.40 | 2.10 |
| Beta Fit | 0.38 | 0.79 | 0.64 | 0.41 | 0.69 | 0.37 | 0.24 | 0.34 | 0.20 | 1.28 | 0.52 | 0.40 | 2.11 | 1.36 | 0.90 | 0.36 | 0.32 | 0.26 |
| Gamma Fit | 0.38 | 0.79 | 0.64 | 0.38 | 0.66 | 0.35 | 0.25 | 0.31 | 0.17 | 1.28 | 0.52 | 0.40 | 2.11 | 1.36 | 0.90 | 0.36 | 0.32 | 0.26 |
| $D_{soil}$ | | | | | | SHALLOW (FLAT) | | | | | | | | | | | |
| Name | FHDS | FHDM | FHDB | FHIS | FHIM | FHIB | FHWS | FHWM | FHWB | FLDS | FLDM | FLDB | FLIS | FLIM | FLIB | FLWS | FLWM | FLWB |
| AD Δ Mean | -7 | -11 | -4 | -7 | -12 | -7 | 1 | -4 | -1 | 13 | 10 | 9 | 34 | 16 | 10 | 29 | 15 | 8 |
| Beta Δ Mean | -18 | -17 | -6 | -21 | -18 | -10 | -14 | -11 | -5 | -20 | -16 | -12 | -12 | -18 | -17 | -12 | -16 | -18 |
| Gamma Δ Mean | -156 | -113 | -67 | -98 | -89 | -54 | -23 | -45 | -29 | -195 | -56 | 11 | -87 | -17 | 40 | 1 | 35 | 57 |
| AD Δ Median | 10 | 3 | -4 | 10 | -2 | -2 | -5 | -10 | -2 | -33 | -18 | 6 | -41 | -27 | 0 | -32 | 3 | 1 |
| Beta Δ Median | 21 | 6 | -2 | 20 | 1 | 0 | 4 | -6 | 1 | -7 | 1 | 20 | -12 | -6 | 14 | -7 | 20 | 13 |
| Gamma Δ Median | 21 | 6 | -2 | 20 | 2 | 0 | 4 | -6 | 1 | -7 | 1 | 20 | -12 | -6 | 14 | -7 | 20 | 13 |
| AD Fit | 0.38 | 0.41 | 0.14 | 0.36 | 0.30 | 0.20 | 0.36 | 0.25 | 0.29 | 0.68 | 0.53 | 0.44 | 2.13 | 1.40 | 0.98 | 1.71 | 1.21 | 0.92 |
| Beta Fit | 0.85 | 0.77 | 0.13 | 0.92 | 0.53 | 0.38 | 0.47 | 0.35 | 0.13 | 0.73 | 0.73 | 0.44 | 2.51 | 1.61 | 0.98 | 1.02 | 0.81 | 0.64 |
| Gamma Fit | 0.85 | 0.77 | 0.14 | 0.92 | 0.54 | 0.38 | 0.47 | 0.35 | 0.13 | 0.73 | 0.73 | 0.44 | 2.51 | 1.61 | 0.98 | 1.02 | 0.81 | 0.64 |

**Table 4: Parameters of the TTDs derived from the simulations with different soil porosities: small = 0.24 m³ m⁻³, normal = 0.39 m³ m⁻³, large = 0.54 m³ m⁻³.**

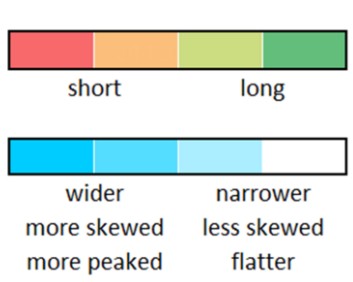

| Name | THDM | | | THIM | | | THWM | | |
| Porosity | Small | Normal | Large | Small | Normal | Large | Small | Normal | Large |
| 1st Quartile | 97 | 137 | 178 | 76 | 105 | 135 | 46 | 67 | 91 |
| Median | 135 | 207 | 301 | 110 | 159 | 226 | 94 | 132 | 168 |
| Mean | 177 | 280 | 385 | 152 | 238 | 326 | 127 | 197 | 269 |
| 3rd Quartile | 202 | 366 | 502 | 169 | 299 | 459 | 143 | 258 | 384 |
| Stand Dev | 248 | 298 | 349 | 239 | 285 | 336 | 239 | 275 | 323 |
| Skewness | 23 | 15 | 10 | 23 | 14 | 9 | 23 | 15 | 9 |
| Exc Kurtosis | 777 | 407 | 223 | 791 | 404 | 211 | 825 | 437 | 220 |

**Table 5. Parameters of the TTDs derived from the simulations with different saturated bedrock hydraulic conductivity $K_{Br}$. Very low = $10^{-7}$, low = $10^{-5}$, medium low = $10^{-3}$, medium high = $10^{-2}$, high = $10^{-1}$, very high = 1, equal = 2 m day⁻¹. The "low" scenario corresponds to THDB.**

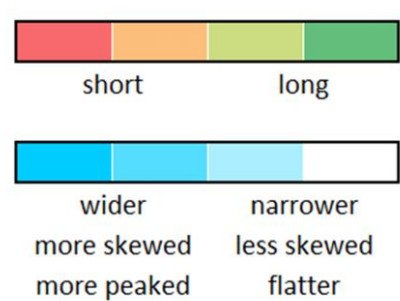

| Name | VLow | Low | MLow | MHigh | High | VHigh | Equal |
| 1st Quartile | 89 | 89 | 90 | 93 | 105 | 102 | 96 |
| Median | 113 | 115 | 122 | 132 | 160 | 144 | 138 |
| Mean | 145 | 151 | 196 | 258 | 239 | 182 | 166 |
| 3rd Quartile | 163 | 167 | 180 | 211 | 308 | 222 | 206 |
| Stand Dev | 138 | 189 | 497 | 520 | 211 | 129 | 116 |
| Skewness | 26 | 28 | 14 | 7 | 2 | 2 | 2 |
| Exc Kurtosis | 1472 | 1233 | 252 | 79 | 11 | 4 | 5 |




**Table 6: Parameters of the TTDs for the simulations with a decay in saturated soil hydraulic conductivity $K_S$. Mean values of scenarios with and without decay are presented in the two columns on the right (μ).**

| Name | THDB | | THWB | | TLDB | | TLWB | | $\mu_{noDecay}$ | $\mu_{Decay}$ |
|---|---|---|---|---|---|---|---|---|---|---|
| Decay | No | Yes | No | Yes | No | Yes | No | Yes | | |
| 1st Quartile | 89 | 84 | 45 | 37 | 126 | 128 | 91 | 81 | 88 | 82 |
| Median | 115 | 111 | 85 | 81 | 291 | 261 | 263 | 173 | 189 | 156 |
| Mean | 151 | 144 | 110 | 103 | 439 | 342 | 400 | 288 | 275 | 219 |
| 3rd Quartile | 167 | 158 | 136 | 132 | 576 | 462 | 546 | 411 | 356 | 291 |
| Stand Dev | 189 | 182 | 173 | 173 | 505 | 354 | 519 | 401 | 347 | 278 |
| Skewness | 28 | 30 | 29 | 31 | 5 | 8 | 6 | 10 | 17 | 20 |
| Exc Kurtosis | 1233 | 1373 | 1426 | 1492 | 70 | 158 | 86 | 201 | 704 | 806 |

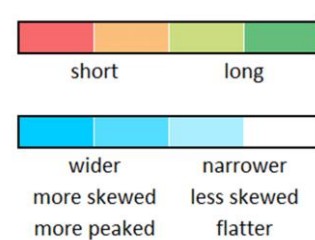

short — long

wider — narrower / more skewed — less skewed / more peaked — flatter

**Table 7: Parameters of the TTDs derived from the model simulations with different precipitation frequencies (arid: low-frequency, 15 days interarrival time; humid: high-frequency, 3 days interarrival time). For comparison, the THDM scenario has a precipitation frequency (derived from a natural precipitation time series) which is quite similar to the humid case. Means (μ) and standard deviations (σ) of the arid and humid scenarios.**

| Name | Arid | | | | | THDM | Humid | | | | | $\mu_{Arid}$ | $\mu_{Humid}$ | $\sigma_{Arid}$ | $\sigma_{Humid}$ |
|---|---|---|---|---|---|---|---|---|---|---|---|---|---|---|---|
| 1st Quartile | 134 | 162 | 173 | 180 | 193 | 137 | 138 | 143 | 136 | 144 | 136 | 168 | 139 | 20 | 3 |
| Median | 222 | 231 | 273 | 282 | 274 | 207 | 220 | 208 | 245 | 241 | 227 | 256 | 228 | 25 | 14 |
| Mean | 290 | 305 | 308 | 324 | 325 | 280 | 277 | 280 | 286 | 291 | 280 | 310 | 283 | 13 | 5 |
| 3rd Quartile | 377 | 352 | 370 | 369 | 368 | 366 | 357 | 339 | 358 | 367 | 360 | 367 | 356 | 8 | 9 |
| Stand Dev | 293 | 281 | 288 | 285 | 286 | 298 | 299 | 294 | 298 | 302 | 302 | 287 | 299 | 4 | 3 |
| Skewness | 14 | 14 | 15 | 14 | 15 | 15 | 16 | 16 | 15 | 15 | 15 | 15 | 15 | 0 | 0 |
| Exc Kurtosis | 382 | 417 | 417 | 407 | 426 | 407 | 433 | 434 | 423 | 416 | 422 | 410 | 426 | 15 | 7 |

short — long

wider — narrower / more skewed — less skewed / more peaked — flatter

**Table 8: Parameters of the TTDs derived from the modeling with silt-type and sand-type soil water retention curves (WRCs). The mean values for the silt $\mu_{Silt}$ and sand $\mu_{Sand}$ scenarios are given on the right side.**

| Name | THDS | | THDB | | THWS | | THWB | | TLDS | | TLDB | | TLWS | | TLWB | | $\mu_{Silt}$ | $\mu_{Sand}$ |
|---|---|---|---|---|---|---|---|---|---|---|---|---|---|---|---|---|---|---|
| WRC | Silt | Sand | Silt | Sand | Silt | Sand | Silt | Sand | Silt | Sand | Silt | Sand | Silt | Sand | Silt | Sand | | |
| 1st Quartile | 244 | 45 | 89 | 38 | 101 | 19 | 45 | 16 | 458 | 54 | 126 | 13 | 232 | 105 | 91 | 13 | 173 | 38 |
| Median | 441 | 142 | 115 | 81 | 218 | 50 | 85 | 42 | 785 | 160 | 291 | 16 | 565 | 393 | 263 | 76 | 345 | 120 |
| Mean | 515 | 175 | 151 | 87 | 354 | 98 | 110 | 58 | 1009 | 341 | 439 | 115 | 796 | 575 | 400 | 225 | 472 | 209 |
| 3rd Quartile | 656 | 223 | 167 | 114 | 501 | 118 | 136 | 82 | 1308 | 491 | 576 | 100 | 1116 | 837 | 546 | 307 | 626 | 284 |
| Stand Dev | 455 | 325 | 189 | 171 | 443 | 245 | 173 | 142 | 880 | 455 | 505 | 250 | 816 | 665 | 519 | 378 | 497 | 329 |
| Skewness | 7 | 18 | 28 | 37 | 7 | 23 | 29 | 44 | 3 | 5 | 5 | 9 | 3 | 3 | 6 | 6 | 11 | 18 |
| Exc Kurtosis | 125 | 453 | 1233 | 1811 | 123 | 791 | 1426 | 2586 | 20 | 62 | 70 | 237 | 22 | 25 | 86 | 98 | 388 | 758 |

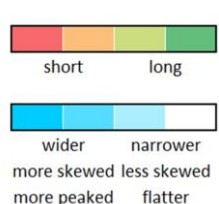

short — long

wider — narrower / more skewed — less skewed / more peaked — flatter

**Table 9: Parameters of the TTDs derived from the modeling with wet (W) or fully saturated (S) antecedent conditions and very large ($^+$; $10\ \text{mm h}^{-1}$) or extreme ($^{+++}$; $100\ \text{mm h}^{-1}$) event precipitation.**




| $K_S$ | HIGH | | | | LOW | | | |
|---|---|---|---|---|---|---|---|---|
| Name | THWB | THSB | THSB$^+$ | THSB$^{+++}$ | TLWB | TLSB | TLSB$^+$ | TLSB$^{+++}$ |
| % SOF$_{10}$ | 0.5 | 8.9 | 9.3 | 64.2 | 75.7 | 91.3 | 92.1 | 99.3 |
| 1st Quartile | 45 | 26 | 26 | 0 | 91 | 12 | 1 | 0 |
| Median | 85 | 77 | 77 | 0 | 263 | 96 | 44 | 0 |
| Mean | 110 | 96 | 96 | 22 | 400 | 258 | 206 | 7 |
| 3rd Quartile | 136 | 124 | 124 | 0 | 546 | 380 | 271 | 0 |
| Stand Dev | 173 | 169 | 169 | 93 | 519 | 413 | 378 | 79 |
| Skewness | 29 | 31 | 31 | 45 | 6 | 5 | 6 | 28 |
| Exc Kurtosis | 1426 | 1526 | 1528 | 4099 | 86 | 81 | 91 | 1930 |

| short | | | long |
|---|---|---|---|

wider — narrower
more skewed — less skewed
more peaked — flatter