# Peer review of "On the shape of forward transit time distributions in low-order catchments"

_Hydrology and Earth System Sciences, 2019_

## Referee Comment (RC1) · Anonymous Referee #1 · 2 Oct 2019

The authors perform a set of numerical experiments to investigate the shape of the transit time distribution for a watershed under different catchment and climate characteristics. They focused mainly the role of soil depth, soil hydraulic conductivity, antecedent soil moisture content and subsequent precipitation amount, but other runs explored also soil porosity, bedrock hydraulic conductivity, depth dependence of the soil hydraulic conductivity and precipitation frequency. The ambitious goal of the article is to relate the shape (i.e., parameters) of common probability density functions (the AD, Gamma, and Beta distributions) to the variability of catchment and climate characteristics.

The paper is well written, with a simple structure that makes it easy to follow. Of course, they authors could not explore the role of all parameters, but the analysis is yet very

inclusive overall. All the details that necessary to reproduce the work are explained in detail, and the presentation and discussion of the results are comprehensive.

However, I have both some major and minor questions that I would ask the authors.

The major question is mostly conceptual. The authors aim at finding general results about the TTD shape variability across locations with different characteristics. I like their systematic approach as an attempt to quantify this variability, e.g. by linking alpha to F. However, I am not surprised that they could only partly achieve their goal.

The issue is that the authors assume a given distribution (e.g., the gamma) for each run. This is analogous to assume that the discharge depends only on the residence time of the water, and not on the water storage. In other words, the authors do not move away from the assumptions behind the instantaneous unit hydrograph approach. From a mathematical standpoint, other authors introduced this assumption by stating that the storage selection function or the loss function (e.g., Botter, 2011; Calabrese and Porporato, 2015) depend on only the residence time (or age). This, however, is the simplest scenario and the farthest from reality. It is very likely, in fact, that if the authors injected the tracer later in the simulation, the TTDs would again differ.

As an example, a more realistic assumption would be to somewhat include a dependence of the TTDs on the overall water storage, or some proxy for it. I think it would be very instructive to explore the dependence of time dependent TTDs parameters on the time dependent water storage. As I mentioned earlier, I still believe that their analysis is very insightful. It is only that, in my opinion, this work could be even more impactful. I wonder whether the authors have comments on this.

I also have some minor questions/comments.

-It seems that boundary conditions, mainly I am referring to the shape of control volume, may have a big effect on TTDs, perhaps that could partly overwhelm the effect of the parameters studied by the authors. Have the authors tested this (e.g., with a

non-elliptical shape)?

-I don't agree with repeating the one year precipitation time series in loop 32 times. First, it is not realistic, and second it might cause some statistical bias. Why not using a Poisson generator throughout the analysis? It would certainly be more consistent. On a different note, there are numerous references that introduced Poisson rainfall/storm. One of the first was Cox and Isham (1988).

-The authors believe that a truncated Gamma or a lognormal distribution may work better over the all range of ages. Why not trying it?

Hoping that these comments may help improve the manuscript, I suggest major revisions.

Botter, Gianluca, Enrico Bertuzzo, and Andrea Rinaldo. "Catchment residence and travel time distributions: The master equation." Geophysical Research Letters 38.11 (2011).

Calabrese, Salvatore, and Amilcare Porporato. "Linking age, survival, and transit time distributions." Water Resources Research 51.10 (2015): 8316-8330.

Cox, David Roxbee, and Valerie Isham. "A simple spatial-temporal model of rainfall." Proceedings of the Royal Society of London. A. Mathematical and Physical Sciences 415.1849 (1988): 317-328.

---

## Referee Comment (RC2) · Anonymous Referee #2 · 10 Oct 2019

The manuscript presents and discusses an interesting analysis based on virtual (numerical) experiments on the TTD in small catchments / hilsllopes. The work is interesting and well done and it touches a relevant topic, namely the identification of the leading components and parameters in the definition of TTDs. The approach is rather "classic" in the sense that the analysis is somewhat based on the concept of time invariant TTD, while recent approaches have shown the importance of other metric, like e.g. the backward TT distributions, for a comprehensive description of water age and contaminant dynamics. Still, the analysis is useful and instructive.

Perhaps the manuscript is too long and involved at times, with plenty of text (with some verbosity) and figures. See for instance the long Conclusion section (and it is the first time I see a subsection there. . .). I think that this might be detrimental to the work as

the reader can easily get lost in the many details and miss the important aspects. Thus, I suggest further distilling the principal results, moving the details that are not important for the storyline in the supplementary material and concentrate on the main results that the Authors want to convey. This would strengthen the message of the work and its diffusion.

With so many fine details, I miss a description of the physical processes, as observed in the model runs, which determine the TTD. What is the impact of subsurface stormflow? Saturated and unsaturated flows? Groundwater? This is important in order to explain the impact of the parameters examined.

In the following a few specific comments.

- Line 38. I would also cite the pioneer works by Niemi (1977) and Nauman (Residence time distribution theory for unsteady stirred tank reactors, Chemical Engineering Science, 1969).

- Line 55-57. Here the introduction moves to the field of groundwater hydrology, where the issue of the BTC tailing (power-law or not) has been the subject of intense discussions in the last 2 decades or so; this short text and citation does not even scratch the surface and it looks quite superficial here.

- Line 57: The sentence of the "great" underestimation of mass is very much debatable, in most cases it's a tiny fraction of the total mass. It may be important for risk assessment of highly toxic compounds, but uncertainty is anyway very large there.

- mTT: please define it (I guess it's mean TT)

- Line 94-95. This sentence is repeated in other parts of the manuscript. By definition such approach cannot "completely" erase differences. The question is whether the approximation is good enough for applications. The study by Ali et al (A comparison of travel-time based catchment transport models, with application to numerical experiments, JoH 2014) shows that in many cases it does the job, also considering the

several sources of uncertainty, including for instance the estimation of ET (not done here).

- Lines 137-139. Unfortunately the effective hydraulic conductivity cannot replace the dispersive effects of the distributed macropores because it only impacts the mean velocity. I would delete this sentence as it is not needed: the exclusion of such component is legitimate and meaningful in my view because of the important role of macrodispersion in the TTD determination.

- Line 159. vertical or hortogonal to the slope? I guess the latter.

- Line 163. 5m of dispersivity is quite a lot, even more so for the vertical one. Why the choice? In this case the inclusion of Dfree looks irrelevant.

- Lines 174-175. What head is provided in the boundary condition? Where is the water table located? This is quite important.

- Line 204. What is the "subsequent precipitation amount"?

- Line 214. I guess that mm/a means mm/y

- Line 214. Please provide more details on the rainfall time series, e.g. regime, climate etc. As a matter of fact TTD depends also on the rainfall regime, not only the total rainfall per year (e.g. Botter et al 2010).

- Line 338. I don't like the definition, I would rather speak of "The Inverse Gaussian distribution, with parameters D, ….., that is a particular solution of the Advection Dispersion Equation". AD is misleading, as ADE can have several different solutions.

- Line 401. This discussion is based on log-log plots, which many times are misleading. The convergence of curves at large time can be an artifact of the plots.

- Line 408-409. Differences seems larger to me. Again, the log-log plot does not help.

- Section 3.3. Some of the (interesting) conclusions here are very similar to those of

Fiori et al (Stochastic analysis of transport in hillslopes: Travel time distribution and source zone dispersion, WRR 2009) which I think is important for this work. There, the different parts of the Gamma distribution pertains to different mechanisms and parameters (soil, bedrock, etc.). The main difference is that they identify the important role of KBr in the behavior of the tail, which is the exponential part of the Gamma, which in turn is related to groundwater discharge. The aquifer volume, which depends on water table, thickness and slope, has an important role here.

- Line 490. I don't see the power law.

- Line 510. How is the fitting done? What inference methods? How one can say that a distribution performs better than another? Any statistical test?

- Line 668. I don't agree with this analysis, the presumed power-law tail covers less than one logscale. Also, identification of power law tails is not simple (see e.g. Pedretti and Bianchi, Reproducing tailing in breakthrough curves: Are statistical models equally representative and predictive? AWR 2018), the emergence of a (short) straight line in a log-log plot may not be enough. At any rate, I would not say that the inadequacy of the distributions in fitting the TTD is because of the tail, that by the way involves a tiny fraction of the mass, which is magnified by the log-log representation. I think that the issue of powerlaw tails is too much emphasized here.

- Section 4.2. This part is not entirely convincing, I can't see the validity of the prediction based on F. By the way the latter does not include other relevant ingredients, like e.g. KBr.

- Line 750. Again, the method cannot erase "all" differences, but perhaps is adequate for many applications.

- Conclusion section. It is too long, one cannot see immediately the main results of the work. It's a pity because there is a lot of interesting material, that however needs to be better distilled and conveyed.

- Line 754-755. "…it is possible to predict the change using the dimensionless flow path number F.". At the third line of the Conclusion section this seems the major conclusion of the work. Is it so? It does not seems like after reading the text.

—————————————————

---

## Referee Comment (RC3) · Anonymous Referee #3 · 19 Nov 2019

This is an interesting paper that describes the relationships between transit time distributions and catchment characteristics. This manuscript is a modeling study for which the authors use a state-of-the-art 3 dimensional saturated unsaturated zone and surface water model. They vary several catchment characteristics and evaluate how this affects the transit time distribution. Moreover they characterize catchment behavior and transittimes using characteristic numbers such as the flowpath number F. The manuscript is well written and mostly easy to read, literature is extensively cited. Maybe the manuscript is long and could be shortened in some sections to gain more impact(17 figures and 9 tables are hard to take in). Having noted this, I must also admit that it is clear that a lot of time, effort and attention has been put into this manuscript. The many variables that have been tested make the results section a bit of a struggle to

read and fully digest. The discussion and conclusions do highlight the most important findings effectively. The conclusion could even be further shortened. I have no major objections to this manuscript and think it could be published with minor revisions. I do wonder why the authors decided to present all their analyses on the transient travel-time distributions instead of the cumulative outflow as mention in section 4.3, which in my opinion would give a results that is less dependent on the precise rainfall sequence? Most interestingly I found that an advection-diffusion based model (mostly darcain) does only under strict conditions yield TTD's that can be described accurately with advection-dispersion TTDs. Therefore a gamma-distribution is not only an effect of preferential flow paths and dual porosity, but also of flowpath-storage relationships as indicated with the flowpath number.

Minor comments Figure 11: why does panel D have curved lines while all the others are straight. Figure 6. I think the order of the legend does not correspond with the panels. But this figure is really hard to understand. For example the center front panel shows "no condition", but still it causes a decrease in traveltime. (y axis). So the decrease is relative to what? All the different colors and linetypes make it hard to understand. Figure 9 and 10: Fig 9 I don't understand why the alpha-plot has no dashed symbols and the D-plot has no solid symbols. This also doesn't seem to match with fig. 10 that has both dashed and solid symbols?

Line 685: not fully sure what you mean to say with "-but only taking". I suggest to replace it with "and use" Line 701. Available storage > storage change. Here I miss the timescale. Do you refer to yearly storage change? Line 701 more water than it can remove (yearly or daily or hourly?) I think you need some kind of characteristic timescale here to define these definition (probably closely related to flowpath number F?) similar in figure 9. Line 760 "where" or "when"?

---

## Author Comment (AC2) · 16 Dec 2019

**Response to Interactive comment by Anonymous Referee #2**

Comments from the referee are printed in black. Authors' responses are printed in red.

The manuscript presents and discusses an interesting analysis based on virtual (numerical) experiments on the TTD in small catchments / hilsllopes. The work is interesting and well done and it touches a relevant topic, namely the identification of the leading components and parameters in the definition of TTDs. The approach is rather "classic" in the sense that the analysis is somewhat based on the concept of time invariant TTD, while recent approaches have shown the importance of other metric, like e.g. the backward TT distributions, for a comprehensive description of water age and contaminant dynamics. Still, the analysis is useful and instructive.

We want to thank referee #2 for the assessment of our manuscript and a detailed and thoughtful review that led to a significant improvement of the study. We would like to point out that in our opinion the concept of 'time variability' is implemented in this study since factors causing time variability of TTDs are either changes in catchment storage (e.g. antecedent soil moisture) or changes in atmospheric forcing (like precipitation amount). Of course, there are other/more factors causing time variability we have not explored yet (e.g. erosion, vegetation, different precipitation patterns).

Perhaps the manuscript is too long and involved at times, with plenty of text (with some verbosity) and figures. See for instance the long Conclusion section (and it is the first time I see a subsection there…). I think that this might be detrimental to the work as the reader can easily get lost in the many details and miss the important aspects. Thus, I suggest further distilling the principal results, moving the details that are not important for the storyline in the supplementary material and concentrate on the main results that the Authors want to convey. This would strengthen the message of the work and its diffusion.

A very valid observation. We have struggled and continue to struggle with exactly the problem the referee describes. In the revised manuscript we are going to condense the conclusion and move more of the details to the supplement.

With so many fine details, I miss a description of the physical processes, as observed in the model runs, which determine the TTD. What is the impact of subsurface stormflow? Saturated and unsaturated flows? Groundwater? This is important in order to explain the impact of the parameters examined.

We have tried to always include explanations of the physical processes that play a role in shaping the TTDs for the different scenarios. Apparently that effort was insufficient in certain places. We are going to add more details on the description of the physical processes where necessary in the revised manuscript.

In the following a few specific comments.

- Line 38. I would also cite the pioneer works by Niemi (1977) and Nauman (Residence time distribution theory for unsteady stirred tank reactors, Chemical Engineering Science, 1969).

Thanks for the additional references. It is very hard to get a comprehensive overview of the pioneering work. Niemi is already cited, we will add Nauman.

- Line 55-57. Here the introduction moves to the field of groundwater hydrology, where the issue of the BTC tailing (power-law or not) has been the subject of intense discussions in the last 2 decades or so; this short text and citation does not even scratch the surface and it looks quite superficial here.

In order to avoid the surface scratching, we are going to do some more research on groundwater breakthrough curves and add some more references. It would be great if you could recommend/point out the most important studies so that we are not going to miss them.

- Line 57: The sentence of the "great" underestimation of mass is very much debatable, in most cases it's a tiny fraction of the total mass. It may be important for risk assessment of highly toxic compounds, but uncertainty is anyway very large there.

Agreed 100%. We will make clear that it might not be relevant from a mass balance perspective (but possibly when conducting a risk assessment).

- mTT: please define it (I guess it's mean TT)

You are correct. We will define it at the first mention (line 48).

- Line 94-95. This sentence is repeated in other parts of the manuscript. By definition such approach cannot "completely" erase differences. The question is whether the approximation is good enough for applications. The study by Ali et al (A comparison of travel-time based catchment transport models, with application to numerical experiments, JoH 2014) shows that in many cases it does the job, also considering the several sources of uncertainty, including for instance the estimation of ET (not done here).

We will add the reference to Ali et al. (2014) and discuss your point.

- Lines 137-139. Unfortunately the effective hydraulic conductivity cannot replace the dispersive effects of the distributed macropores because it only impacts the mean velocity. I would delete this sentence as it is not needed: the exclusion of such component is legitimate and meaningful in my view because of the important role of macrodispersion in the TTD determination.

Thank you for the constructive comment. We will proceed as suggested.

- Line 159. vertical or hortogonal to the slope? I guess the latter.

It is indeed vertical and not orthogonal to the slope (but that makes only a small difference).

- Line 163. 5m of dispersivity is quite a lot, even more so for the vertical one. Why the choice? In this case the inclusion of Dfree looks irrelevant.

The longitudinal dispersivity and lateral dispersivity were estimated with regard to the length scale of the model catchment (100 m). $\alpha L \approx 5$ m were estimated using the relation between the longitudinal dispersivity and length scale described in Gelhar et al., 1992 and Schulze-Makuch, 2005 (regression $\alpha = 0.085*L^{0.81}$). We agree that the free-solution diffusion is significantly smaller than the dispersion and could have been neglected. We will clarified this in the manuscript adding the references [Gelhar et al., 1992] and [Schulze-Makuch, 2005].

References:
Gelhar, L.W., Welty, C., Rehfeldt, K.R., 1992. A critical review of data on field-scale dispersion in aquifers. Water Resources Research 28 (7), 1955–1974.
Schulze-Makuch, D. (2005). Longitudinal dispersivity data and implications for scaling behavior. Groundwater, 43(3), 443-456.

- Lines 174-175. What head is provided in the boundary condition? Where is the water table located? This is quite important.
Thanks for catching that. I thought I would have written it somewhere. We will add information on the location of the head (it is equal to the surface elevation).

- Line 204. What is the "subsequent precipitation amount"?
Will be clarified (essentially a measure of the amount of precipitation after the delivery of the tracer).

- Line 214. I guess that mm/a means mm/y
Yes, HESS officially prefers this abbreviation.

- Line 214. Please provide more details on the rainfall time series, e.g. regime, climate etc. As a matter of fact TTD depends also on the rainfall regime, not only the total rainfall per year (e.g. Botter et al 2010).
We agree it is correct that the TTD also depends on the distribution of rainfall. We investigate the influence of different precipitation event frequencies. The precipitation time series we used has the following properties: Average interarrival time: 2.64 days; Average event duration: 3.17 days. The climate in the north west of Germany can be described as maritime temperate (Cfb in the Köppen classification) Maximum precipitation falls usually in June (65 mm), minimum in February (28 mm). We are going to add this information to the manuscript.

- Line 338. I don't like the definition, I would rather speak of "The Inverse Gaussian distribution, with parameters D, …, that is a particular solution of the Advection Dispersion Equation". AD is misleading, as ADE can have several different solutions.
We would like to follow your suggestion. If we reformulate the description in the following way, would it be correct?
1) The inverse Gaussian distribution with dispersion parameter D (dimensionless) and mean mTT (d) that is a particular solution of the advection dispersion equation (Eq. 6):

- Line 401. This discussion is based on log-log plots, which many times are misleading. The convergence of curves at large time can be an artifact of the plots.
It is correct that log-log plot can make large differences at large times appear smaller. However, they also exaggerate small differences at short times. In this particular case we are interested more in the short time differences because we expect the largest differences at the beginning of the TTDs. At late times, differences are averaged out more and more.

- Line 408-409. Differences seems larger to me. Again, the log-log plot does not help.

We double-checked the numbers and they are correct. The fact that the differences seem larger is probably due to the very high resolution of the log-log plot for short and very short times.

- Section 3.3. Some of the (interesting) conclusions here are very similar to those of Fiori et al (Stochastic analysis of transport in hillslopes: Travel time distribution and source zone dispersion, WRR 2009) which I think is important for this work. There, the different parts of the Gamma distribution pertains to different mechanisms and parameters (soil, bedrock, etc.). The main difference is that they identify the important role of KBr in the behavior of the tail, which is the exponential part of the Gamma, which in turn is related to groundwater discharge. The aquifer volume, which depends on water table, thickness and slope, has an important role here.

Thank you for pointing us to this reference. It is indeed a very interesting study that we were not aware of yet. In the revised manuscript we are going to include it and discuss the similarities/differences we found in our work.

- Line 490. I don't see the power law.

We are aware of the fact that straight lines in log-log plots are necessary for identifying power laws but insufficient as evidence. So you are right, we cannot be sure whether they are actually power laws just from this graphical analysis. Would you have a recommendation on how to call these tails instead?

- Line 510. How is the fitting done? What inference methods? How one can say that a distribution performs better than another? Any statistical test?

In Section 2.4.3 (Fitting) we describe the procedure. It was done by the least squares method on the cumulative distributions.

- Line 668. I don't agree with this analysis, the presumed power-law tail covers less than one logscale. Also, identification of power law tails is not simple (see e.g. Pedretti and Bianchi, Reproducing tailing in breakthrough curves: Are statistical models equally representative and predictive? AWR 2018), the emergence of a (short) straight line in a log-log plot may not be enough. At any rate, I would not say that the inadequacy of the distributions in fitting the TTD is because of the tail, that by the way involves a tiny fraction of the mass, which is magnified by the log-log representation. I think that the issue of powerlaw tails is too much emphasized here.

We agree with your comment. We would like to find a better description of the TTD tail behavior (maybe we should just describe the fact that the tails begin with a sudden break in the slope of the TTD and continue from there on as straight lines on a log-log plot). It's also clear that the tails are not relevant in terms of the total mass balance and will hardly be noticed for most solutes – with the exception of highly toxic pollutants. We will make sure to stress this in the revised manuscript.

- Section 4.2. This part is not entirely convincing, I can't see the validity of the prediction based on F. By the way the latter does not include other relevant ingredients, like e.g. KBr.

We understand your concerns. This section is not meant to represent to full and complete truth about TTD shapes. It is rather an attempt to find some structure in the way TTD shapes change with certain parameters, an attempt to explore overarching

principles. Many of the potential shape-controlling parameters are still excluded from this analysis (like KBr). We will try to put more emphasis on this interpretation of our results in the revised manuscript.

- Line 750. Again, the method cannot erase "all" differences, but perhaps is adequate for many applications.
Agreed. We are going to add this remark to the revised manuscript.

- Conclusion section. It is too long, one cannot see immediately the main results of the work. It's a pity because there is a lot of interesting material, that however needs to be better distilled and conveyed.
There is definitely room for improvement in the conclusion section. We agree with your criticism and we will do our best to further condense the conclusions in the revised manuscript.

- Line 754-755. "…it is possible to predict the change using the dimensionless flow path number F.". At the third line of the Conclusion section this seems the major conclusion of the work. Is it so? It does not seems like after reading the text.
This can indeed be considered the main conclusion of our work. We need to make sure that this outcome is conveyed better in the revised conclusion section.

---

## Author Response (AR1)

**Response to Interactive comment by Anonymous Referee #1**

Comments from the referee are printed in black. Authors' responses are printed in red.

The authors perform a set of numerical experiments to investigate the shape of the transit time distribution for a watershed under different catchment and climate characteristics. They focused mainly the role of soil depth, soil hydraulic conductivity, antecedent soil moisture content and subsequent precipitation amount, but other runs explored also soil porosity, bedrock hydraulic conductivity, depth dependence of the soil hydraulic conductivity and precipitation frequency. The ambitious goal of the article is to relate the shape (i.e., parameters) of common probability density functions (the AD, Gamma, and Beta distributions) to the variability of catchment and climate characteristics.
Exactly.

The paper is well written, with a simple structure that makes it easy to follow. Of course, they authors could not explore the role of all parameters, but the analysis is yet very inclusive overall. All the details that necessary to reproduce the work are explained in detail, and the presentation and discussion of the results are comprehensive.
We want to thank referee #1 for the assessment of our manuscript and a thoughtful review that led to a significant improvement of the study.

However, I have both some major and minor questions that I would ask the authors.

The major question is mostly conceptual. The authors aim at finding general results about the TTD shape variability across locations with different characteristics. I like their systematic approach as an attempt to quantify this variability, e.g. by linking alpha to F. However, I am not surprised that they could only partly achieve their goal.

The issue is that the authors assume a given distribution (e.g., the gamma) for each run. This is analogous to assume that the discharge depends only on the residence time of the water, and not on the water storage. In other words, the authors do not move away from the assumptions behind the instantaneous unit hydrograph approach. From a mathematical standpoint, other authors introduced this assumption by stating that the storage selection function or the loss function (e.g., Botter, 2011; Calabrese and Porporato, 2015) depend on only the residence time (or age). This, however, is the simplest scenario and the farthest from reality. It is very likely, in fact, that if the authors injected the tracer later in the simulation, the TTDs would again differ.
As an example, a more realistic assumption would be to somewhat include a dependence of the TTDs on the overall water storage, or some proxy for it. I think it would be very instructive to explore the dependence of time dependent TTDs parameters on the time dependent water storage. As I mentioned earlier, I still believe that their analysis is very insightful. It is only that, in my opinion, this work could be even more impactful. I wonder whether the authors have comments on this.

This is a very valid point that we hope to address by examining the influence of antecedent moisture content on the shape of TTDs. We believe that the antecedent moisture content of the soil is a proxy for the water storage of the catchment (the bedrock is almost permanently fully saturated). We agree that a tracer injection at a different point in time would cause the TTD shape to differ (depending to a much higher degree on the current antecedent soil moisture content than on the pattern of following precipitation). In section 3.2 (figure 6, panel in the upper left corner) we analyze the dependence of time dependent TTD parameters on the time dependent water storage. You can see that, e.g., for situations when the water storage is high, $K_S$ has a higher influence on TTDs than when water storage is low, while the relative influence of $P_{sub}$ is larger when the initial water storage is low. In the revised manuscript we have clarified Figure 6 and improve its discussion in the text.

I also have some minor questions/comments.

-It seems that boundary conditions, mainly I am referring to the shape of control volume, may have a big effect on TTDs, perhaps that could partly overwhelm the effect of the parameters studied by the authors. Have the authors tested this (e.g., with a non-elliptical shape)?

Again, a valid point that we had not tested yet. Catchment shape was one of the properties we also thought could potentially influence the TTD shape but chose to investigate at a later point in a different study (like, e.g., catchment size or slope). However, after your remarks we decided to try out two additional catchment shapes to get an idea whether it would have a significant impact on the results. So we tested one catchment with the center of gravity located farther away from the outlet and another catchment with the center of gravity located closer to the outlet (catchment size and slope staying the same in all cases). We found that changing the catchment shape had substantially less influence on the TTD shape than we expected. We have added this analysis to the manuscript.

-I don't agree with repeating the one year precipitation time series in loop 32 times. First, it is not realistic, and second it might cause some statistical bias. Why not using a Poisson generator throughout the analysis? It would certainly be more consistent. On a different note, there are numerous references that introduced Poisson rainfall/storm. One of the first was Cox and Isham (1988).

Thanks for the additional reference, we have added it to the manuscript. In order to erase your worries about looping the time series we did what you suggested and created a 33 year time series with a Poisson generator. The resulting TTDs did not differ significantly from the ones we derived from the looped time series. We have added a comment on this to the manuscript and a figure to the supplement.

-The authors believe that a truncated Gamma or a lognormal distribution may work better over the all range of ages. Why not trying it?

Ok, following your suggestion we conducted this analysis. The truncated lognormal distribution did indeed capture almost all of the TTD shapes better. Additionally we also tested the regular (i.e. not truncated) lognormal distribution and found that it is a better representation for the shape of TTDs in catchments with high $K_S$ than the advection-dispersion distribution. To reflect the results of these new analyses we modified our results and discussion sections accordingly in the revised manuscript.

Hoping that these comments may help improve the manuscript, I suggest major revisions. Thanks again for the valuable input that helped to improve our paper.

Botter, Gianluca, Enrico Bertuzzo, and Andrea Rinaldo. "Catchment residence and travel time distributions: The master equation." Geophysical Research Letters 38.11 (2011).

Calabrese, Salvatore, and Amilcare Porporato. "Linking age, survival, and transit time distributions." Water Resources Research 51.10 (2015): 8316-8330.

Cox, David Roxbee, and Valerie Isham. "A simple spatial-temporal model of rainfall." Proceedings of the Royal Society of London. A. Mathematical and Physical Sciences 415.1849 (1988): 317-328.

**Response to Interactive comment by Anonymous Referee #2**

Comments from the referee are printed in black. Authors' responses are printed in red.

The manuscript presents and discusses an interesting analysis based on virtual (numerical) experiments on the TTD in small catchments / hilsllopes. The work is interesting and well done and it touches a relevant topic, namely the identification of the leading components and parameters in the definition of TTDs. The approach is rather "classic" in the sense that the analysis is somewhat based on the concept of time invariant TTD, while recent approaches have shown the importance of other metric, like e.g. the backward TT distributions, for a comprehensive description of water age and contaminant dynamics. Still, the analysis is useful and instructive.

We want to thank referee #2 for the assessment of our manuscript and a detailed and thoughtful review that led to a significant improvement of the study. We would like to point out that in our opinion the concept of 'time variability' is implemented in this study since factors causing time variability of TTDs are either changes in catchment storage (e.g. antecedent soil moisture) or changes in atmospheric forcing (like precipitation amount). Of course, there are also other/more factors causing time variability we have not explored yet (e.g. erosion, vegetation, different precipitation patterns).

Perhaps the manuscript is too long and involved at times, with plenty of text (with some verbosity) and figures. See for instance the long Conclusion section (and it is the first time I see a subsection there…). I think that this might be detrimental to the work as the reader can easily get lost in the many details and miss the important aspects. Thus, I suggest further distilling the principal results, moving the details that are not important for the storyline in the supplementary material and concentrate on the main results that the Authors want to convey. This would strengthen the message of the work and its diffusion.

A very valid observation. We have struggled with exactly the problem the referee describes. In the revised manuscript we have condensed the conclusion, restructured the results and discussion sections and moved more of the details to the supplement.

With so many fine details, I miss a description of the physical processes, as observed in the model runs, which determine the TTD. What is the impact of subsurface stormflow? Saturated and unsaturated flows? Groundwater? This is important in order to explain the impact of the parameters examined.

We have tried to always include explanations of the physical processes that play a role in shaping the TTDs for the different scenarios. Apparently that effort was insufficient in certain places. We have added more details on the description of the physical processes in the results and discussion sections of the revised manuscript.

In the following a few specific comments.

- Line 38. I would also cite the pioneer works by Niemi (1977) and Nauman (Residence time distribution theory for unsteady stirred tank reactors, Chemical Engineering Science, 1969).
Thanks for the additional references. It is very hard to get a comprehensive overview of the pioneering work. Niemi is already cited, we have added Nauman (1969).

- Line 55-57. Here the introduction moves to the field of groundwater hydrology, where the issue of the BTC tailing (power-law or not) has been the subject of intense discussions in the last 2 decades or so; this short text and citation does not even scratch the surface and it looks quite superficial here.
In order to avoid the surface scratching, we have done more research on groundwater breakthrough curves and added more references.

- Line 57: The sentence of the "great" underestimation of mass is very much debatable, in most cases it's a tiny fraction of the total mass. It may be important for risk assessment of highly toxic compounds, but uncertainty is anyway very large there.
Agreed 100%. We have made clear that it might not be relevant from a mass balance perspective (but possibly when conducting a risk assessment).

- mTT: please define it (I guess it's mean TT)
You are correct. We define it at the first mention (line 64).

- Line 94-95. This sentence is repeated in other parts of the manuscript. By definition such approach cannot "completely" erase differences. The question is whether the approximation is good enough for applications. The study by Ali et al (A comparison of travel-time based catchment transport models, with application to numerical experiments, JoH 2014) shows that in many cases it does the job, also considering the several sources of uncertainty, including for instance the estimation of ET (not done here).
We have added the reference to Ali et al. (2014) and discuss your point.

- Lines 137-139. Unfortunately the effective hydraulic conductivity cannot replace the dispersive effects of the distributed macropores because it only impacts the mean velocity. I would delete this sentence as it is not needed: the exclusion of such component is legitimate and meaningful in my view because of the important role of macrodispersion in the TTD determination.
Thank you for the constructive comment. We have proceeded as suggested.

- Line 159. vertical or hortogonal to the slope? I guess the latter.
It is indeed vertical and not orthogonal to the slope (but that makes only a small difference).

- Line 163. 5m of dispersivity is quite a lot, even more so for the vertical one. Why the choice? In this case the inclusion of Dfree looks irrelevant.
The longitudinal dispersivity and lateral dispersivity were estimated with regard to the length scale of the model catchment (100 m). αL ≈ 5 m were estimated using the relation between the longitudinal dispersivity and length scale described in Gelhar et al., 1992 and Schulze-Makuch, 2005 (regression α = 0.085*L$^{0.81}$). We agree that the free-solution diffusion is significantly smaller than the dispersion and could have been neglected. We have clarified this in the manuscript adding the references [Gelhar et al., 1992] and [Schulze-Makuch, 2005].
References:
Gelhar, L.W., Welty, C., Rehfeldt, K.R., 1992. A critical review of data on field-scale dispersion in aquifers. Water Resources Research 28 (7), 1955–1974.
Schulze-Makuch, D. (2005). Longitudinal dispersivity data and implications for scaling behavior. Groundwater, 43(3), 443-456.

- Lines 174-175. What head is provided in the boundary condition? Where is the water table located? This is quite important.
Thanks for catching that. I thought I would have written it somewhere. We have added information on the location of the head (it is equal to the surface elevation).

- Line 204. What is the "subsequent precipitation amount"?
Clarified (essentially a measure of the amount of precipitation after the delivery of the tracer).

- Line 214. I guess that mm/a means mm/y
Yes, HESS officially prefers this abbreviation.

- Line 214. Please provide more details on the rainfall time series, e.g. regime, climate etc. As a matter of fact TTD depends also on the rainfall regime, not only the total rainfall per year (e.g. Botter et al 2010).
We agree it is correct that the TTD also depends on the distribution of rainfall. We investigate the influence of different precipitation event frequencies. The precipitation time series we used has the following properties: Average interarrival time: 2.64 days; Average event duration: 3.17 days. The climate in the north west of Germany can be described as maritime temperate (Cfb in the Köppen classification) Maximum precipitation falls usually in June (65 mm), minimum in February (28 mm). We have added this information to the manuscript.

- Line 338. I don't like the definition, I would rather speak of "The Inverse Gaussian distribution, with parameters D, …, that is a particular solution of the Advection Dispersion Equation". AD is misleading, as ADE can have several different solutions.
We would like to follow your suggestion. We have reformulate the description in the following way:
1) The inverse Gaussian distribution with dispersion parameter D (dimensionless) and mean mTT (d) that is a particular solution of the advection dispersion equation (Eq. 6):

- Line 401. This discussion is based on log-log plots, which many times are misleading.
The convergence of curves at large time can be an artifact of the plots.

It is correct that log-log plot can make large differences at large times appear smaller. However, they also exaggerate small differences at short times. In this particular case we are interested more in the short time differences because we expect the largest differences at the beginning of the TTDs. At late times, differences are averaged out more and more.

- Line 408-409. Differences seems larger to me. Again, the log-log plot does not help.
We double-checked the numbers and they are correct. The fact that the differences seem larger is probably due to the very high resolution of the log-log plot for short and very short times.

- Section 3.3. Some of the (interesting) conclusions here are very similar to those of Fiori et al (Stochastic analysis of transport in hillslopes: Travel time distribution and source zone dispersion, WRR 2009) which I think is important for this work. There, the different parts of the Gamma distribution pertains to different mechanisms and parameters (soil, bedrock, etc.). The main difference is that they identify the important role of KBr in the behavior of the tail, which is the exponential part of the Gamma, which in turn is related to groundwater discharge. The aquifer volume, which depends on water table, thickness and slope, has an important role here.
Thank you for pointing us to this reference. It is indeed a very interesting study that we were not aware of yet. In the revised manuscript have included it.

- Line 490. I don't see the power law.
We are aware of the fact that straight lines in log-log plots are necessary for identifying power laws but insufficient as evidence. So you are right, we cannot be sure whether they are actually power laws just from this graphical analysis. Therefore we have changed our focus away from the power-law towards the characteristic break in the slope where the tail part begins.

- Line 510. How is the fitting done? What inference methods? How one can say that a distribution performs better than another? Any statistical test?
In Section 2.4.3 (Fitting) we describe the procedure. It was done by the least squares method on the cumulative distributions.

- Line 668. I don't agree with this analysis, the presumed power-law tail covers less than one logscale. Also, identification of power law tails is not simple (see e.g. Pedretti and Bianchi, Reproducing tailing in breakthrough curves: Are statistical models equally representative and predictive? AWR 2018), the emergence of a (short) straight line in a log-log plot may not be enough. At any rate, I would not say that the inadequacy of the distributions in fitting the TTD is because of the tail, that by the way involves a tiny fraction of the mass, which is magnified by the log-log representation. I think that the issue of powerlaw tails is too much emphasized here.
We agree with your comment. We have changed our description of the TTD tail behavior (now we just describe the fact that the tails begin with a sudden break in the slope of the TTD and continue from there on as straight lines on a log-log plot). It's also clear that the tails are not relevant in terms of the total mass balance and will hardly be noticed for most solutes – with the exception of highly toxic pollutants. We have made sure to stress this in the revised manuscript.

- Section 4.2. This part is not entirely convincing, I can't see the validity of the prediction based on F. By the way the latter does not include other relevant ingredients, like e.g. KBr.
We understand your concerns. This section is not meant to represent to full and complete truth about TTD shapes. It is rather an attempt to find some structure in the way TTD shapes change with certain parameters, an attempt to explore overarching principles. Many of the potential shape-controlling parameters are still excluded from this analysis (like KBr). We have tried to put more emphasis on this interpretation of our results in the revised manuscript.

- Line 750. Again, the method cannot erase "all" differences, but perhaps is adequate for many applications.
Agreed. We have added this remark to the revised manuscript.

- Conclusion section. It is too long, one cannot see immediately the main results of the work. It's a pity because there is a lot of interesting material, that however needs to be better distilled and conveyed.
There is definitely room for improvement in the conclusion section. We agree with your criticism and we have done our best to condense, restructure and clarify the conclusions in the revised manuscript. To this end we moved a lot of text from the conclusion to the results and discussion sections.

- Line 754-755. "…it is possible to predict the change using the dimensionless flow path number F.". At the third line of the Conclusion section this seems the major conclusion of the work. Is it so? It does not seems like after reading the text.
This can indeed be considered the main conclusion of our work. We have made sure that this outcome is conveyed better in the revised conclusion section.

**Response to Interactive comment by Anonymous Referee #3**

Comments from the referee are printed in black. Authors' responses are printed in red.

This is an interesting paper that describes the relationships between transit time distributions and catchment characteristics. This manuscript is a modeling study for which the authors use a state-of-the-art 3 dimensional saturated unsaturated zone and surface water model. They vary several catchment characteristics and evaluate how this affects the transit time distribution. Moreover they characterize catchment behavior and transittimes using characteristic numbers such as the flowpath number F. The manuscript is well written and mostly easy to read, literature is extensively cited. Maybe the manuscript is long and could be shortened in some sections to gain more impact(17 figures and 9 tables are hard to take in).
Thank you for reading and evaluating our manuscript. We fully agree that it is long and that it would benefit from further condensing certain sections. We have already shortened it considerably in the past and have made another effort to achieve this.

Having noted this, I must also admit that it is clear that a lot of time, effort and attention has been put into this manuscript. The many variables that have been tested make the results section a bit of a struggle to read and fully digest. The discussion and conclusions do highlight the most important findings effectively. The conclusion could even be further shortened.
Thank you also for acknowledging the effort we put into this research. It started small but grew into this large study comprising more and more of the relevant catchment and climate properties. Still, it is far from being complete (there are still more parameters to test and analyze). We have make another effort to streamline the results section better in the revised manuscript and to shorten the conclusion to the most important take-away messages (moving more of the less important findings to the supplement).

I have no major objections to this manuscript and think it could be published with minor revisions. I do wonder why the authors decided to present all their analyses on the transient traveltime distributions instead of the cumulative outflow as mention in section 4.3, which in my opinion would give a results that is less dependent on the precise rainfall sequence?
The decision to plot the TT probabilities against the actual transit time instead of the cumulative outflow is mainly based on the desire to work with TTDs that are 'real' in order to get an impression of how they would look like and change their shape in real-world catchments. Also, we could not have investigated the influence of precipitation frequency or the influence of different precipitation patterns/sequences with the cumulative outflow method.

Most interestingly I found that an advection-diffusion based model (mostly darcain) does only under strict conditions yield TTD's that can be described accurately with advection-dispersion TTDs. Therefore a gamma-distribution is not only an effect of preferential flow paths and dual porosity, but also of flowpath-storage relationships as indicated with the flowpath number.

Thank you for pointing this out. Actually, based on another reviewers comment we additionally tested lognormal and truncated lognormal distributions to fit the modeled TTDs. We found that the lognormal distributions capture the TTD shapes in many cases better than the AD distributions.

Minor comments Figure 11: why does panel D have curved lines while all the others are straight. If you look closely, you can see that the lines in panel A are also slightly curved. This is due to the fact that both $P_{sub}$ and $\theta_{ant}$ have three different modes (large, medium, small and wet, intermediate, dry) while $D_{soil}$ and $K_S$ have both only two modes.

Figure 6. I think the order of the legend does not correspond with the panels. But this figure is really hard to understand. For example the center front panel shows "no condition", but still it causes a decrease in traveltime. (y axis). So the decrease is relative to what? All the different colors and linetypes make it hard to understand.
Agreed. This is a very complex figure that is hard to understand. We have made another effort to make it clearer and simpler (also adding more explanation in the text and in the caption). We double-checked and all the different colors and line types are indeed correct (also the order in the legends).

Figure 9 and 10: Fig 9 I don't understand why the alpha-plot has no dashed symbols and the D-plot has no solid symbols. This also doesn't seem to match with fig. 10 that has both dashed and solid symbols?
This correct observation is due to the fact that we recommend using gamma distributions for catchments with low hydraulic conductivity (solid) and Log-normal distributions for catchments with high hydraulic conductivity (dashed). In figure 10 we show relationships for all (low and high $K_S$) scenarios.

Line 685: not fully sure what you mean to say with "-but only taking". I suggest to replace it with "and use"
Good suggestion. We have modified this section anyways due to the new results we received from the fitting of the lognormal distributions.

Line 701. Available storage > storage change. Here I miss the timescale. Do you refer to yearly storage change?
The timescale is the combined average interevent and event duration (~5 days). A much shorter time scale – compared to the yearly storage change – that makes F more variable/responsive in time. We have added this information to the manuscript.

Line 701 more water than it can remove (yearly or daily or hourly?) I think you need some kind of characteristic timescale here to define these definition (probably closely related to flowpath number F?) similar in figure 9.

Yes, we have added the characteristic time scale (combined average interevent and event duration) to the description.

Line 760 "where" or "when"?
When sounds indeed better. Thanks.

**On the shape of forward transit time distributions in low-order catchments**

Ingo Heidbüchel[1], Jie Yang[1], Andreas Musolff[1], Peter Troch[2], Ty Ferré[2], Jan H. Fleckenstein[1]

[1]Department of Hydrogeology, Helmholtz Centre for Environmental Research – UFZ, Leipzig, 04318, Germany
[2]Department of Hydrology and Atmospheric Sciences, University of Arizona, Tucson, 85721, USA

*Correspondence to*: Ingo Heidbüchel (ingo.heidbuechel@ufz.de)

**Abstract.** Transit time distributions (TTDs) integrate information on timing, amount, storage, mixing and flow paths of water and thus characterize hydrologic and hydrochemical catchment response unlike any other descriptor. Here, we simulate the shape of TTDs in an idealized low-order catchment investigating whether it changes systematically with certain catchment and climate properties. To this end, we used a physically-based, spatially-explicit 3-D model, injected tracer with a precipitation event and recorded the resulting TTDs at the outlet of a small (~6000 $m^2$) catchment for different scenarios. We found that the TTDs can be subdivided into four parts: 1) early part – controlled by soil hydraulic conductivity and antecedent soil moisture content, 2) middle part – transition zone with no clear pattern or control, 3) later part – influenced by soil hydraulic conductivity and subsequent precipitation amount and 4) very late tail of the breakthrough curve – governed by bedrock hydraulic conductivity. The modeled TTD shapes can be predicted using a dimensionless number: higher initial peaks are observed if the inflow of water to a catchment is not equal to its capacity to discharge water via subsurface flow paths, lower initial peaks are connected to increasing available storage. In most cases the modeled TTDs were humped with non-zero initial values and varying weights of the tails. Therefore, none of the best-fit theoretical probability functions could exactly describe the entire TTD shape. Still, we found that generally the Gamma and the Log-normal distribution work better for scenarios of low and high soil hydraulic conductivity, respectively.

**1. Introduction**

Transit time distributions (TTDs) characterize hydrologic catchment behavior unlike any other function or descriptor. They integrate information on timing, amount, storage, mixing and flow paths of water and can be modified to predict reactive solute transport (van der Velde et al., 2010; Harman et al., 2011; Musolff et al., 2017; Lutz et al., 2017). If observed in a time series, TTDs bridge the gap between hydrologic response (celerity) and hydrologic transport (velocity) in catchments by linking them via the change in water storage and the varying contributions of old (pre-event) and young (event) water to streamflow (Heidbüchel et al., 2012). TTDs are time and space-variant and hence no TTD of any individual precipitation event completely resembles another one. Therefore, in order to effectively utilize TTDs for the prediction of, e.g., the effects of pollution events or water availability, it is necessary to find ways to understand and systematically describe the shape and scale of TTDs so that they are applicable in different locations and at different times. In this paper we look for first order principles that describe how the shape and scale of TTDs change, both spatially and temporally. This way we hope to improve our understanding of the dominant factors affecting hydrologic transport and response behavior at the catchment scale.

**1.1. Initial use of theoretical probability distributions**

Since the concept of TTDs was introduced, many studies have reported on their potential shapes and sought ways to describe them with different mathematical models like, e.g., the piston-flow and exponential models (Begemann and Libby, 1957;

Eriksson, 1958; Nauman, 1969), the advection-dispersion model (Nir, 1964; Małoszewski and Zuber, 1982) and the two parallel linear reservoirs model (Małoszewski et al., 1983; Stockinger et al., 2014). Dinçer et al. (1970) were the first to combine TTDs for individual precipitation events via the now commonly used convolution integral.

Early studies reported that the outflow from entire catchments is characterized best with the exponential model (Rodhe et al.,

1996; McGuire et al., 2005). However, neither the advection-dispersion nor the exponential model is able to capture the observed heav tails of  solute signals in  streamflow (Kirchner et al., 2000). Instead, the more heavy-tailed TTDs created by advection and dispersion of spatially distributed rainfall inputs traveling toward the stream can be modeled with

TTDs resembling Gamma distributions (Kirchner et al., 2001). Likewise, tracer time series from many catchments exhibit fractal 1/f scaling, which is consistent with Gamma TTDs with shape parameter $\alpha \approx 0.5$ (Kirchner, 2016).

**1.2. General observations on the shape of TTDs**

From the application of conceptual and physically-based models we know that individual TTDs  are highly irregular and that they can rapidly chang in time for successive precipitation events (van der Velde et al., 2010; Rinaldo et al., 2011; Heidbüchel et al., 2012; Harman and Kim,

2014). If the early part of TTDs (mainly controlled by unsaturated transport in the soil layer) resembles a power law while the subsoil is responsible for the exponential tailing, the combination of those two parts can result in TTD shapes that are similar to Gamma distributions (Fiori et al., 2009). In the field of groundwater hydrology there have been intense discussions on the tailing of break through curves (e.g. on the issue of whether they follow a power-law or not) (Haggerty et al., 2000; Becker and Shapiro, 2003; Zhang et al., 2007; Pedretti et al., 2013; Fiori and Becker, 2015; Pedretti and Bianchi, 2018).

If disregarded,  heavy tails can constitute a significant problem when using TTDs to predict solute transport because the legacy of contamination can be

**Kommentiert [IHh1]:** - Line 38. I would also cite the pioneer works by Niemi (1977) and Nauman (Residence time distribution theory for unsteady stirred tank reactors, Chemical Engineering Science, 1969).
Answer: Thanks for the additional references. It is very hard to get a comprehensive overview of the pioneering work. Niemi is already cited, we have added Nauman.

**Kommentiert [IHh2]:** - Line 57: The sentence of the "great" underestimation of mass is very much debatable, in most cases it's a tiny fraction of the total mass. It may be important for risk assessment of highly toxic compounds, but uncertainty is anyway very large there.
Answer: Agreed 100%. We have made clear that it might not be relevant from a mass balance perspective (but possibly when conducting a risk assessment).

**Kommentiert [IHh3]:** - Line 55-57. Here the introduction moves to the field of groundwater hydrology, where the issue of the BTC tailing (power-law or not) has been the subject of intense discussions in the last 2 decades or so; this short text and citation does not even scratch the surface and it looks quite superficial here.
Answer: In order to avoid the surface scratching, we have done some more research on groundwater breakthrough curves and added more references.

[revised manuscript text omitted]

**Kommentiert [IHh4]:** - Line 94-95. This sentence is repeated in other parts of the manuscript. By definition such approach cannot "completely" erase differences. The question is whether the approximation is good enough for applications. The study by Ali et al (A comparison of travel-time based catchment transport models, with application to numerical experiments, JoH 2014) shows that in many cases it does the job, also considering the several sources of uncertainty, including for instance the estimation of ET (not done here). Answer: We have added the reference to Ali et al. (2014) and discuss your point.

[revised manuscript text omitted]

**Kommentiert [IHh5]:** - Lines 137-139. Unfortunately the effective hydraulic conductivity cannot replace the dispersive effects of the distributed macropores because it only impacts the mean velocity. I would delete this sentence as it is not needed: the exclusion of such component is legitimate and meaningful in my view because of the important role of macrodispersion in the TTD determination.
Answer: Thank you for the constructive comment. We have proceeded as suggested.

**Kommentiert [IHh6]:** - Line 159. vertical or hortogonal to the slope? I guess the latter.
Answer: It is indeed vertical and not orthogonal to the slope (but that makes only a small difference).

**Kommentiert [IHh7]:** - Line 163. 5m of dispersivity is quite a lot, even more so for the vertical one. Why the choice? In this case the inclusion of Dfree looks irrelevant.
Answer: The longitudinal dispersivity and lateral dispersivity were estimated with regard to the length scale of the model catchment (100 m). αL ≈ 5 m were estimated using the relation between the longitudinal dispersivity and length scale described in Gelhar et al., 1992 and Schulze-Makuch, 2005 (regression α = 0.085*L^0.81). We agree that the free-solution diffusion is significantly smaller than the dispersion and could have been neglected. We have clarified this in the manuscript adding the references [Gelhar et al., 1992] and [Schulze-Makuch, 2005].

[Figure]

**Figure 1: 3-D model domain and shape of the virtual catchment from top (left), front (upper right) and side (middle right). The blue square indicates the outflow boundary with constant head condition. The red layer represents the soil which has a much higher hydraulic conductivity than the underlying bedrock (grey). The orange lines indicate the zone of convergence (but no explicit channel). The two additional catchment shapes (top-heavy and bottom-heavy) we tested in section 2.2.1 are shown in the black box.**

**2.1.1. Boundary conditions**

Both the bottom and the sides of the domain were impermeable boundaries. A constant head boundary condition (equal to the surface elevation) was assigned to the lower front edge of the subsurface domain (nodes in the blue square in Fig. 1), allowing outflow from both the bedrock and the soil. A critical depth boundary was assigned to the lower edge of the surface domain (on top of the constant head boundary) to allow for overland flow out of the catchment. The surface of the catchment received spatially uniform precipitation. We used a recorded time series of precipitation from the north-east of Germany (maritime temperate climate: Cfb in the Köppen climate classification) amounting to 690 mm $a^{-1}$ (Fig. 2a). The time series was 1 year long and repeated 32 more times to cover the entire modeling period which lasted a total of 33 years. We made sure that the looping of the precipitation time series would not cause any unwanted artifacts in the resulting TTDs (see Text S1 and Figure S1 in the supplement). Neither evaporation nor transpiration was considered during the simulations. This means that all precipitation we applied was effective precipitation that would eventually discharge at the catchment outlet. The addition of the process of evapotranspiration is planned in a follow-up modeling study to investigate what influence it exerts on catchment TTDs. The tracer was applied uniformly over the entire catchment during a precipitation event that lasted one hour, had an intensity of 0.1 mm $h^{-1}$ and a tracer concentration of 1 kg $m^3$. This resulted in a total applied tracer mass of 0.589 kg .

**Kommentiert [IHh8]:** - Lines 174-175. What head is provided in the boundary condition? Where is the water table located? This is quite important.
Answer: Thanks for catching that. I thought I would have written it somewhere. We have added information on the location of the head (it is equal to the surface elevation).

**Kommentiert [IHh9]:** - Line 214. I guess that mm/a means mm/y
Answer: Yes, HESS officially prefers this abbreviation.

- Line 214. Please provide more details on the rainfall time series, e.g. regime, climate etc. As a matter of fact TTD depends also on the rainfall regime, not only the total rainfall per year (e.g. Botter et al 2010).
Answer: We agree it is correct that the TTD also depends on the distribution of rainfall. We investigate the influence of different precipitation event frequencies. The precipitation time series we used has the following properties: Average interarrival time: 2.64 days; Average event duration: 3.17 days. The climate in the north west of Germany can be described as maritime temperate (Cfb in the Köppen classification) Maximum precipitation falls usually in June (65 mm), minimum in February (28 mm). We have added this information to the manuscript.

[Figure]

Figure 2: a) One-year time series of subsequent precipitation (looped 33 times for the entire modeling period and rescaled for smaller or larger subsequent precipitation amounts). Tracer application took place during the first hour of the model runs. b) Time series of subsequent precipitation for a high-frequency scenario (humid) and a low-frequency scenario (arid). The total precipitation amount is the same for both scenarios.

**2.1.2. Initial conditions**

The model runs were initialized with three different antecedent soil moisture conditions $\theta_{ant}$ – a dry one ($\theta_{ant} = 22.0\ \%$; correspondingent to an average effective saturation of the soil layer $S_{eff} \approx 50\ \%$), an intermediate one ($\theta_{ant} = 28.8\ \%$; $S_{eff} \approx 70\ \%$) and a wet one ($\theta_{ant} = 35.6\ \%$; $S_{eff} \approx 90\ \%$). To obtain realistic distributions of soil moisture, we first ran the model starting with full saturation and without any precipitation input and let the soils drain until the average effective saturation reached the states for our initial conditions. We recorded these conditions and used them as initial conditions of the virtual experiment runs. In general, the soil remained wetter close to the outlet in the lower part of the catchment and became drier in the upper part of the catchment. Note that the process of evapotranspiration was excluded from the modeling so that the lowest achievable saturation was essentially defined by the field capacity. An average effective saturation $S_{eff}$ of approximately 50 % was the lowest that could be achieved by draining the soil layer since the lower part stayed highly saturated due to the constant head boundary condition being equal to the surface elevation at the outlet. The upper parts of the catchment, however, were initiated with much lower $S_{eff}$ values ($\approx$ 30 % in the dry scenarios). That means that although an $S_{eff}$ value of 50 % seems to be quite high, it actually represents an overall dry state of the catchment soil. Throughout the modeling runs the dry initial condition did not occur again as that would have taken 13 years of drainage without any precipitation for the scenarios with high soil hydraulic conductivity $K_S$ and almost 1500 years for the scenarios with low $K_S$. The inclusion of evapotranspiration would, however, speed up the drying process of the soil and hence make these initial conditions more realistic.

**2.2. Model scenarios**

To investigate how different catchment and climate properties influence the shape of forward TTDs we systematically varied four characteristic properties from high to low values and looked at the resulting TTD shapes of all the possible combinations (for a total number of 36 scenarios). The properties we focused on were soil depth ($D_{soil}$), saturated soil hydraulic conductivity ($K_S$), antecedent soil moisture content ($\theta_{ant}$) and subsequent precipitation amount ($P_{sub}$, essentially a measure of the amount of precipitation that falls after the delivery of the traced event) (Fig. 3).

[Figure]

**Figure 3: The four properties that were varied to explore their influence on the shape and scale of TTDs: soil depth $D_{soil}$, saturated
soil hydraulic conductivity $K_S$, antecedent soil moisture $\theta_{ant}$ and subsequent precipitation amount $P_{sub}$. The bedrock hydraulic
conductivity $K_{Br}$ was kept constant for all of these base-case scenarios.**

We tested two soil depths $D_{soil}$, namely depths of 0.5 m and 1.0 m, evenly distributed across the entire catchment. Similarly, we chose two saturated soil hydraulic conductivities $K_S$, a high one with 2.0 m day$^{-1}$ (similar to fine sand) and a low one with

0.02 m day$^{-1}$ (similar to silt). Three states of antecedent moisture content $\theta_{ant}$ were selected to represent initial conditions – 50,

70 and 90 % of effective saturation. Finally the subsequent precipitation amount $P_{sub}$ was varied in three steps from 345 over

690 up to 1380 mm a$^{-1}$. The original  precipitation  time series (690 mm a$^{-1}$, Fig. 2a) was  and rescaled  to obtain time series with smaller and larger amounts

**Kommentiert [IHh10]:** - Line 204. What is the "subsequent precipitation amount"?
Answer: Clarified (essentially a measure of the amount of precipitation after the delivery of the tracer).

**Kommentiert [IHh11]:** - Line 214. I guess that mm/a means mm/y
Answer: Yes, HESS officially prefers this abbreviation.

- Line 214. Please provide more details on the rainfall time series, e.g. regime, climate etc. As a matter of fact TTD depends also on the rainfall regime, not only the total rainfall per year (e.g. Botter et al 2010).
Answer: We agree it is correct that the TTD also depends on the distribution of rainfall. We investigate the influence of different precipitation event frequencies. The precipitation time series we used has the following properties: Average interarrival time: 2.64 days; Average event duration: 3.17 days. The climate in the north west of Germany can be described as maritime temperate (Cfb in the Köppen classification) Maximum precipitation falls usually in June (65 mm), minimum in February (28 mm). We are going to add this information to the manuscript.

[revised manuscript text omitted]

**Kommentiert [IHh12]:** - Line 338. I don't like the definition, I would rather speak of "The Inverse Gaussian distribution, with parameters D, …, that is a particular solution of the Advection Dispersion Equation". AD is misleading, as ADE can have several different solutions.
Answer: We would like to follow your suggestion. If have reformulated the description in the following way:

[revised manuscript text omitted]

**Kommentiert [IHh13]:** - Line 401. This discussion is based on log-log plots, which many times are misleading. The convergence of curves at large time can be an artifact of the plots.
Answer: It is correct that log-log plot can make large differences at large times appear smaller. However, they also exaggerate small differences at short times. In this particular case we are interested more in the short time differences because we expect the largest differences at the beginning of the TTDs. At late times, differences are averaged out more and more.

**Kommentiert [IHh14]:** - Line 408-409. Differences seems larger to me. Again, the log-log plot does not help.
Answer: We double-checked the numbers and they are correct. The fact that the differences seem larger is probably due to the very high resolution of the log-log plot for short and very short times.

[revised manuscript text omitted]

**Kommentiert [IHh15]:** Figure 6. I think the order of the legend does not correspond with the panels. But this figure is really hard to understand. For example the center front panel shows "no condition", but still it causes a decrease in traveltime. (y axis). So the decrease is relative to what? All the different colors and linetypes make it hard to understand. Agreed. This is a very complex figure that is hard to understand. We have made another effort to make it clearer and simpler (also adding more explanation in the text and in the caption). We double-checked and all the different colors and line types are indeed correct (also the order in the legends).

[revised manuscript text omitted]

**Kommentiert [IHh16]:** - Section 3.3. Some of the (interesting) conclusions here are very similar to those of Fiori et al (Stochastic analysis of transport in hillslopes: Travel time distribution and source zone dispersion, WRR 2009) which I think is important for this work. There, the different parts of the Gamma distribution pertains to different mechanisms and parameters (soil, bedrock, etc.). The main difference is that they identify the important role of KBr in the behavior of the tail, which is the exponential part of the Gamma, which in turn is related to groundwater discharge. The aquifer volume, which depends on water table, thickness and slope, has an important role here.
Answer: Thank you for pointing us to this reference. It is indeed a very interesting study that we were not aware of yet. In the revised manuscript we have included it.

120 days before diverging again. After the transition period, the shape of the TTDs is governed by $P_{sub}$ (i.e. essentially the climate) and $K_S$, with larger $P_{sub}$ and higher $K_S$ causing a more rapid decline of outflow and hence a compression of the TTDs.

Finally, the shape of the tails of the TTDs is controlled by the hydraulic conductivity of the bedrock $K_{Br}$ (not the soil $K_S$) (see also Fiori et al., 2009). In many cases tThese tails constitute straight lines in the log-log plots (which is necessary but insufficient for identifying follow power law functions) in many (but not in all) cases. Furthermore, all modeled TTDs share one common feature – for every subsequent precipitation event there is a more or less discernible spike. Generally, larger subsequent events cause higher spikes (i.e., a higher proportion of outflow during those events) while the size of the spikes decreases at later times. And although this multitude of local maxima in the probability density curve does invoke a sense of irregularity, the general pattern of shapes of the TTDs is not influenced by the individual subsequent events (Fig. 5 and Table

S3 in the supplement), which is why we decided to smooth the TTDs for visual comparison so that the underlying systematic changes in shapes are more clearly visible and understood (see Fig. S75 in the supplement).

Practical implications can be drawn from our results concerning, e.g., pollution events. Some catchments are more vulnerable to pollution in the sense that they tend to store pollutants for a longer period of time and hence exhibit long legacy effects.

Especially catchments with TTDs with heavy tails belong in that category (i.e. catchments with deeper soils and a moderate hydraulic conductivity difference between soil and bedrock). Also, certain moments in time are worse for pollution events to happen – a spill occurring during dry conditions will stay in the catchment longer than a spill during wet conditions because it is more likely to reach the bedrock and stay in contact with it before it is flushed out of the soils. Accordingly, locations and situations that lead to a longer storage of decaying pollutants will eventually release less of the solutes downstream.

We also plotted the probability density replacing the actual transit time with the cumulative outflow to check whether this would eradicate the differences between the different distributions (see Fig. S108 in the supplement). We made two interesting observations: 1. For the scenarios with high $K_S$, the differences between the distributions were reduced considerably. Especially for the cumulative probability distributions there were hardly any discernible differences left. The largest discrepancies could still be found in the early part of the distributions where the distributions with high $\theta_{ant}$ continued to have larger outflow probabilities. 2. For the scenarios with low $K_S$, the individual distributions did not collapse into a single cumulative probability distribution. They rather split up into three distributions according to their $P_{sub}$ values. That means that for the scenarios with larger $P_{sub}$ a larger amount of cumulative outflow was necessary to flush out the same amount of tracer compared to the scenarios with smaller $P_{sub}$.

**3.4.    Distribution fitting**

Shape parameters of the best-fit Inverse Gaussian (*D*), Gamma (*α*) and Log-normalAdvection Dispersion (*σ*) (*D*) distributions as well as flow path numbers (*F*) for the 36 different scenarios are listed in Table 2. The parameters *D*, *α* and *σ D* range from

0.15 to 0.98, from 0.78 to 3.66 and from 0.51 to 1.15 0.15 to 0.98, respectively. *F* ranges from –0.22 to 0.63.

First we compared the performances of only these three probability distributions with two parameters. Out of the 36 model scenarios, the Inverse GaussianAdvection Dispersion model (AD) yielded the best fit 519 times, the GammaBeta model 135

**Kommentiert [IHh17]:** - Line 490. I don't see the power law. Answer: We are aware of the fact that straight lines in log-log plots are necessary for identifying power laws but insufficient as evidence. So we cannot be sure whether they are actually power laws just from this graphical analysis. Therefore we have changed our focus away from the power-law towards the characteristic break in the slope where the tail part begins.

times and the Log-normal 12 times

[revised manuscript text omitted]

(13)

Generally, for similar catchments with low $K_S$, Gamma distributions are more likely to fit the TTDs. The relatively higher
proportion of surface flow within and surface outflow from these catchments seems to favor flow and transport dynamics that
are best represented by the shapes of Gamma distributions because they are able to capture both rapid response (high initial
values) as well as the relatively slow outflow from the soils and the bedrock (long tails). In contrast, similar catchments with
high $K_S$ and only small proportions of surface flow are more likely to behave according to Log-normal
distributions with less rapid response from surface flow (low initial values) and faster outflow from the more conductive soils
(higher and narrower modes at intermediate transit times). A notable exception are scenarios where catchments
with highly conductive soils still experience larger proportions of surface outflow (> 25 %; $F$ > 0.05) due to large amounts of
$P_{\text{sub}}$ – these dynamics cannot be predicted by the same relationship since they produce  distributions with larger
contributions of advective transport and lighter tails and hence smaller values of $\sigma$ (indicated by the black circle in Fig. 11).

[Figure]

[Figure]

**Figure 911:** Relationship between the dimensionless flow path number *F* and the shape parameters *α* (upper panel, scenarios with low *K*S) and  (lower panel, scenarios with high *K*S) of the Gamma and the -normal distribution, respectively. Yellow colors indicate dry, green intermediate and blue wet *θ*ant; thick marker lines indicate large, mid-sized lines medium and thin lines small *P*sub; solid lines indicate low, dashed lines high *K*S; lighter shades of a color indicate shallow, darker shades deep *D*soil. The dotted trend lines are the best-fit regressions for the relationship between the flow path number and the shape parameters *α* (light blue) and  (orange). The points in the black circle are excluded from the regression analysis since they are associated with scenarios of excessive surface outflow.

**Kommentiert [IHh20]:** Figure 9 and 10: Fig 9 I don't understand why the alpha-plot has no dashed symbols and the D-plot has no solid symbols. This also doesn't seem to match with fig. 10 that has both dashed and solid symbols?
This correct observation is due to the fact that we recommend using gamma distributions for catchments with low hydraulic conductivity (solid) and Log-normal distributions for catchments with high hydraulic conductivity (dashed). In figure 10 we show relationships for all (low and high *K*S) scenarios.

distributions with short mTTs (in highly conductive soils the shape is more affected than the scale, in soils with low $K_S$ the
scale is more affected than the shape). When $K_S$ is decreasing mTT is decreasing (in case $P_{sub}$ is big then the shapes of the
TTDs also changes towards having lighter tails). Quite similar patterns can be observed for increasing $D_{soil}$ and decreasing $P_{sub}$
— with mTTs becoming longer and TTD shapes increasing the tail weight when $K_S$ is high and becoming more humped when
$K_S$ is low.

[Figure]

Figure 10: Gamma shape parameters ($\alpha$) and mean transit times (mTTs) for individual scenarios with different combinations of
catchment and climate properties. Yellow colors indicate dry, green intermediate and blue wet antecedent moisture conditions; thick
marker lines indicate large, mid-sized lines medium and thin lines small amounts of subsequent precipitation; solid lines indicate
low, dashed lines high saturated hydraulic conductivities; lighter shades of a color indicate shallow, darker shades deep soils. The
red boxes contain exemplary Gamma distributions with shape and scale corresponding to the red dot location.

[Figure]

**3.6.    Effects of other factors on the shape of TTDs**

**Kommentiert [IHh21]:** Figure 11: why does panel D have curved lines while all the others are straight.
If you look closely, you can see that the lines in panel A are also slightly curved. This is due to the fact that both $P_{sub}$ and $θ_{ant}$ have three different modes (large, medium, small and wet, intermediate, dry) while $D_{soil}$ and $K_S$ have both only two modes.

[revised manuscript text omitted]

**Kommentiert [IHh22]:** - Line 668. I don't agree with this analysis, the presumed power-law tail covers less than one logscale. Also, identification of power law tails is not simple (see e.g. Pedretti and Bianchi, Reproducing tailing in breakthrough curves: Are statistical models equally representative and predictive? AWR 2018), the emergence of a (short) straight line in a log-log plot may not be enough. At any rate, I would not say that the inadequacy of the distributions in fitting the TTD is because of the tail, that by the way involves a tiny fraction of the mass, which is magnified by the log-log representation. I think that the issue of powerlaw tails is too much emphasized here.
Answer: We agree with your comment. We have changed our description of the TTD tail behavior (now we just describe the fact that the tails begin with a sudden break in the slope of the TTD and continue from there on as straight lines on a log-log plot). It's also clear that the tails are not relevant in terms of the total mass balance and will hardly be noticed for most solutes – with the exception of highly toxic pollutants. We have made sure to stress this in the revised manuscript.

**Kommentiert [IHh23]:** Line 685: not fully sure what you mean to say with "-but only taking". I suggest to replace it with "and use"
Good suggestion. We have modified this section anyways due to the new results we received from the fitting of the lognormal distributions.

Kommentiert [IHh24]: - Section 4.2. This part is not entirely convincing, I can't see the validity of the prediction based on F. By the way the latter does not include other relevant ingredients, like e.g. KBr.
Answer: We understand your concerns. This section is not meant to represent to full and complete truth about TTD shapes. It is rather an attempt to find some structure in the way TTD shapes change with certain parameters, an attempt to explore overarching principles. Many of the potential shape-controlling parameters are still excluded from this analysis (like KBr). We have tried to put more emphasis on this interpretation of our results in the revised manuscript.

Kommentiert [IHh25]: Line 701. Available storage > storage change. Here I miss the timescale. Do you refer to yearly storage change?
The timescale is the combined average interevent and event duration (~5 days). A much shorter time scale – compared to the yearly storage change – that makes F more variable/responsive in time. We have added this information to the manuscript.

Kommentiert [IHh26]: Line 701 more water than it can remove (yearly or daily or hourly?) I think you need some kind of characteristic timescale here to define these definition (probably closely related to flowpath number F?) similar in figure 9.
Yes, we have added the characteristic time scale (combined average interevent and event duration) to the description.

[revised manuscript text omitted]

Kommentiert [IHh27]: - Line 750. Again, the method cannot erase "all" differences, but perhaps is adequate for many applications.
Answer: Agreed. We have added this remark to the revised manuscript.

**4.4. Limitations and Outlook**

[revised manuscript text omitted]

**Supplement**

**Contents of this file**

**Introduction**

The supplement consists of 7 text files, 11 figures and 11 tables. The individual sections contain a comparison of TTDs resulting from a looped and a continuous precipitation time series (Text S1, Fig. S1), an overview of the different modeling scenarios (Table S1), the precipitation time series created for testing the influence of the sequence of events (Fig. S2) and the table containing all distributions metrics for those 15 scenarios (Table S3), the tracer mass in storage, the cumulative tracer mass of the outflux and the cumulative mass balance errors for the 36 scenarios (Fig. S3), methods for the computation of TTD metrics (Text S2, Fig. S4), methods for and results from the determination of young water fractions (Text S3, Fig. S5, Table S2), a comparison of different theoretical probability density functions (Fig. S6), TTD smoothing (Text S4, Fig. S7), the derivation of TTDs from tracer breakthrough curves (Fig. S8), the analysis of spatial tracer distribution over the catchment and in its profile (Text S5, Fig. S9), outflow probability distributions plotted against cumulative outflow (Fig. S10), measures of how well the different theoretical probability distributions fit the modeled TTDs (Table S4), metrics of the TTDs derived from scenarios with other catchment and climate properties (Tables S5 to S11), a method to add power law tails to  Gamma probability distributions (Text S6, Fig. S6) as well as an example of using TTDs for reactive solute transport applications (Text S7, Fig. S11).

**Text S1.**

We looped a one-year-long time series of precipitation from the north-east of Germany and used it as a boundary condition throughout the 33-year-long model period in all of the scenarios. In order to check whether the looping would cause any unwanted artifacts in the resulting TTDs we additionally created a 32-year-long synthetic continuous precipitation time series with similar attributes: average yearly precipitation amount of 690 mm a$^{-1}$, average event interarrival time of 2.64 days and Poisson distributed precipitation event amounts. This continuous (non-looped) time series was attached to the one-year-long recorded time series to create a second 33-year-long time series. The comparison of the two resulting TTDs shows that the looping does not introduce any artifactual irregularities into the TTD shape (Fig. S1).

**Text S2.**

[revised manuscript text omitted]

short — long
wider — narrower — more skewed — less skewed — more peaked — flatter

**Table S10: Parameters of the TTDs derived from the modeling with different catchment shapes (top-heavy, bottom-heavy). 'Mid' refers to the basic oval shape.**

| $K_S$ | HIGH | | | | LOW | | | |
|---|---|---|---|---|---|---|---|---|
| Name | THWB | THSB | THSB$^+$ | THSB$^{+++}$ | TLWB | TLSB | TLSB$^+$ | TLSB$^{+++}$ |
| % SOF$_{10}$ | 0.5 | 8.9 | 9.3 | 64.2 | 75.7 | 91.3 | 92.1 | 99.3 |
| 1st Quartile | 45 | 26 | 26 | 0 | 91 | 12 | 1 | 0 |
| Median | 85 | 77 | 77 | 0 | 263 | 96 | 44 | 0 |
| Mean | 110 | 96 | 96 | 22 | 400 | 258 | 206 | 7 |
| 3rd Quartile | 136 | 124 | 124 | 0 | 546 | 380 | 271 | 0 |
| Stand Dev | 173 | 169 | 169 | 93 | 519 | 413 | 378 | 79 |
| Skewness | 29 | 31 | 31 | 45 | 6 | 5 | 6 | 28 |
| Exc Kurtosis | 1426 | 1526 | 1528 | 4099 | 86 | 81 | 91 | 1930 |

short — long wider — narrower
more skewed — less skewed
more peaked — flatter

**Table S11: Parameters of the TTDs derived from the modeling with wet (W) or fully saturated (S) antecedent conditions and very large ($^+$; 10 mm h$^{-1}$) or extreme ($^{+++}$; 100 mm h$^{-1}$) event precipitation.**

---

## Author Response (AR2)

**Response to Comment by Anonymous Referee #1**

Comments from the referee are printed in black. Authors' responses are printed in red.

As I stated in the earlier review, this is an interesting paper that explores systematically the role of different catchment characteristics on the shape of the transit time distribution. The authors have provided a thoughtful and solid response to my comments by broadening their analysis and, where necessary, by making substantial changes. This and the overall restructuring definitely improved the manuscript.

We would like to acknowledge the thoughtful review we were provided with (in the first round of reviews) that led to a substantial improvement of this research.

As a point of discussion, it would be important to test these results with observations, especially for understanding their generality. In many circumstances, for examples in areas where soils are characterized by macropores and preferential pathways, traditional hydrological modeling (i.e., Richards equation) may not be suitable.

This is 100% correct. We need to substantiate these model results and the ensuing theories with actual observations. Thanks for catching that this is not discussed anymore in the manuscript. It must have been gotten lost during the extensive restructuring of the discussion and conclusion sections. We have added this discussion point to the 'Limitations and Outlook' section 4.4.

**Response to Comment by Anonymous Referee #2**

Comments from the referee are printed in black. Authors' responses are printed in red.

The Authors have done a commendable job in their revision. I believe that the manuscript has considerably improved and that it is almost ready to be published.

Thank you for this assessment and the acknowledgement of our efforts to improve the quality of the paper. The improvement of the manuscript was in many regards also due to the excellent reviews we received from the referees (in the first round of reviews).

There is just an important issue that is left and that requires to be handled, i.e. the dispersivity employed in the simulations. The value for alphaL is not small, and its justification based on the dataset by Gelhar et al 1992, as well as analyses based on it, is not appropriate. First, Gelhar himself warned against uses of his data to infer macrodispersivity, which unfortunately was done several times in the past. Then, in a recent work (Zech et al, Is unique scaling of aquifer macrodispersivity supported by field data? Water Resources Research, 2015) the whole dataset of Gelhar et al (including data from Schulze-Makuch) was reconsidered and it turned out that most of the data could not be employed for the analysis of macrodispersion, resulting in a lack of trend. Hence, the formula employed by the Authors, although still used and (unfortunately) even recommended by some national agencies, is definitely not correct and cannot be used.

We were not aware of the fact that the data and conclusions of Gelhar were reevaluated recently, thank you for pointing this out. We have now included the citation of Zech et al. (2015). In order to test the impact of a smaller longitudinal dispersivity alphaL on our simulation results we ran additional scenarios (with alphaL = 0.5 m instead of 5.0 m). Fortunately, the differences between the runs with large and small alphaL values where quite small. The differences were small but noticeable (especially in the very early part of the TTDs) for runs with low hydraulic conductivity in the soils and almost non-existent for runs with high hydraulic conductivity. We added this analysis to the supplement.

Frankly speaking I don't think it is appropriate at this stage to run again the simulations with a more realistic value of dispersion, and I would never suggest it. May I suggest instead a simpler way out, i.e. a brief justification saying that the value for "local" dispersivity is not expected to have a significant impact on the TTD, as demonstrated by Fiori and Russo (2008, cited). The reason is that local dispersion is usually negligible compared to the dispersion determined by the spatial distribution of rainfall (coined as "source zone dispersion" in Fiori et al, 2009, cited). This should solve such small issue and let this nice contribution proceed further toward publication.

Thank you for this suggestion. We have added this argument to the description of the model setup. In the previous version of the manuscript we only referred to Fiori et al. (2009) with regard

to macrodispersion - although we were aware of the fact that the source zone dispersion also plays an important role.

[revised manuscript text omitted]

**Kommentiert [IHh1]:** Ref 2: There is just an important issue that is left and that requires to be handled, i.e. the dispersivity employed in the simulations. The value for alphaL is not small, and its justification based on the dataset by Gelhar et al 1992, as well as analyses based on it, is not appropriate. First, Gelhar himself warned against uses of his data to infer macrodispersivity, which unfortunately was done several times in the past. Then, in a recent work (Zech et al, Is unique scaling of aquifer macrodispersivity supported by field data? Water Resources Research, 2015) the whole dataset of Gelhar et al (including data from Schulze-Makuch) was reconsidered and it turned out that most of the data could not be employed for the analysis of macrodispersion, resulting in a lack of trend. Hence, the formula employed by the Authors, although still used and (unfortunately) even recommended by some national agencies, is definitely not correct and cannot be used.

Reply: We were not aware of the fact that the data and conclusions of Gelhar were reevaluated recently, thank you for pointing this out. We have now included the citation of Zech et al. (2015). In order to test the impact of a smaller longitudinal dispersivity alphaL on our simulation results we ran additional scenarios (with alphaL = 0.5 m instead of 5.0 m). Fortunately, the differences between the runs with large and small alphaL values where quite small. The differences were small but noticeable (especially in the very early part of the TTDs) for runs with low hydraulic conductivity in the soils and almost non-existent for runs with high hydraulic conductivity. We added this analysis to the supplement.

**Kommentiert [IHh2]:** Ref 2: Frankly speaking I don't think it is appropriate at this stage to run again the simulations with a more realistic value of dispersion, and I would never suggest it. May I suggest instead a simpler way out, i.e. a brief justification saying that the value for "local" dispersivity is not expected to have a significant impact on the TTD, as demonstrated by Fiori and Russo (2008, cited). The reason is that local dispersion is usually negligible compared to the dispersion determined by the spatial distribution of rainfall (coined as "source zone dispersion" in Fiori et al, 2009, cited). This should solve such small issue and let this nice contribution proceed further toward publication.

Reply: Thank you for this suggestion. We have added this argument to the description of the model setup. In the previous version of the manuscript we only referred to Fiori et al. (2009) with regard to macrodispersion - although we were aware of the fact that the source zone dispersion also plays an important role.

[revised manuscript text omitted]

**Kommentiert [IHh3]:** Ref 1: As a point of discussion, it would be important to test these results with observations, especially for understanding their generality. In many circumstances, for examples in areas where soils are characterized by macropores and preferential pathways, traditional hydrological modeling (i.e., Richards equation) may not be suitable.

Reply: This is 100% correct. We need to substantiate these model results and the ensuing theories with actual observations. Thanks for catching that this is not discussed anymore in the manuscript. It must have been gotten lost during the extensive restructuring of the discussion and conclusion sections. We have added this discussion point to the 'Limitations and Outlook' section 4.4.

[revised manuscript text omitted]

Legend: small error — large error

**Contents of this file**

5  **Introduction**

The supplement consists of 7 text files, 12 figures and 12 tables. The individual sections contain a comparison of TTDs resulting from different dispersivity values (Text S1, Fig. S1, Table S1), a comparison of TTDs resulting from a looped and a continuous precipitation time series (Text S1, Fig. S1), an overview of the different modeling scenarios (Table S2), the precipitation time series created for testing the influence of the sequence of events (Fig. S2) and the table containing all

10  distributions metrics for those 15 scenarios (Table S3), the tracer mass in storage, the cumulative tracer mass of the outflux and the cumulative mass balance errors for the 36 scenarios (Fig. S3), methods for the computation of TTD metrics (Text S2, Fig. S4), methods for and results from the determination of young water fractions (Text S3, Fig. S5, Table S2), a comparison of different theoretical probability density functions (Fig. S6), TTD smoothing (Text S4, Fig. S7), the derivation of TTDs from tracer breakthrough curves (Fig. S8), the analysis of spatial tracer distribution over the catchment

15  and in its profile (Text S5, Fig. S9), outflow probability distributions plotted against cumulative outflow (Fig. S10), measures of how well the different theoretical probability distributions fit the modeled TTDs (Table S5), metrics of the TTDs derived from scenarios with other catchment and climate properties (Tables S5 to S11), a method to add power law tails to Gamma probability distributions (Text S6, Fig. S6) as well as an example of using TTDs for reactive solute transport applications (Text S7, Fig. S11).

 **Text S1.**

In order to rule out that a smaller model value for the longitudinal dispersivity $\alpha_L$ would influence our results significantly, we set up two additional runs. In these runs we reduced $\alpha_L$ by one order of magnitude from 5 m to 0.5 m. We chose to test the two scenarios THWB and TLDS since they result in the longest and shortest transit times of all model scenarios, respectively. We found only small deviations for TLDS in the early part of the TTD (with none of the transit time quartiles being more than five percent longer than in the reference case with larger $\alpha_L$) and virtually no difference for THWB (Fig. S1 and Table S1).

**Text S12.**

We looped a one-year-long time series of precipitation from the north-east of Germany and used it as a boundary condition throughout the 33-year-long model period in all of the scenarios. In order to check whether the looping would cause any unwanted artifacts in the resulting TTDs we additionally created a 32-year-long synthetic continuous precipitation time series with similar attributes: average yearly precipitation amount of 690 mm a$^{-1}$, average event interarrival time of 2.64 days and Poisson distributed precipitation event amounts. This continuous (non-looped) time series was attached to the one-year-long recorded time series to create a second 33-year-long time series. The comparison of the two resulting TTDs shows that the looping does not introduce any artifactual irregularities into the TTD shape (Fig. S12).

**Text S23.**

[revised manuscript text omitted]

145     **Figure S23: 15 different precipitation time series with similar exponential distributions of precipitation event amounts and interarrival times. The y-axes all range from 0 to 40 mm. The time series were created to test the influence of event sequence on the shape of TTDs.**

[Figure]

 Figure S34: a) Total tracer mass in storage, b) cumulative tracer mass outflux, c) cumulative mass balance error for all 36 scenarios. Note that most scenarios plot on top of each other in panel c).

[Figure]

**Figure S45: Distribution metrics of three different Gamma distributions with varying shape parameter *α* and equal mean (300 h).**
a) Black dashed line: mean (300 h), dotted black line and filled areas under the curves: standard deviation. b) Black dashed line:
mean (300 h), colored dashed lines: medians, filled areas under the curves range from the first to the third quartile (Q₁–Q₃).

155

[Figure]

**Figure S6: Change of young water fractions ($F_{yw}$) with the flow path number ($F$) for four different catchment and climate properties. Yellow colors indicate dry, green intermediate and blue wet $\theta_{ant}$. Thick marker lines indicate big, mid-sized lines medium and thin lines small amounts of $P_{sub}$. Solid lines indicate low, dashed lines high $K_S$, lighter shades of a color indicate shallow, darker shades deep $D_{soil}$.**

[Figure]

165      **Figure S67: A set of ten different common theoretical probability distributions (all but the power law having a mean value of 300 h, grey line). The black dotted line is a distribution that is a combination of a Gamma distribution with the tail of a power law distribution. The inset has a log-log scale.**

[Figure]

170     **Figure S8: Unsmoothed (orange) and smoothed (black) version of the same TTD.**

[Figure]

**Figure S89**: Precipitation input (cyan), total outflow (blue) and tracer concentration in the outflow (red) for the first three years of the model run for scenario THDM. The tracer breakthrough curve (when normalized) constitutes the TTD of the injected tracer impulse.

[Figure]

**Figure S910**: Time series of tracer concentration distribution in the subsurface across the entire catchment, in a depth profile in the center of the catchment for two scenarios (top: FHWB; bottom: TLDS) with very different resulting TTDs shapes. The dotted black line in the profiles represents the soil–bedrock interface; the white dashed line is the water table.

180

[Figure]

**Figure S101:** Similar to Fig. 7 except for the fact that outflow probability is plotted against cumulative outflow instead of transit time. Distributions are grouped by soil depth (upper panels a and b = deep (thick); lower panels c and d = shallow (flat)) and saturated hydraulic conductivity (left panels a and c = high; right panels b and d = low). Yellow colors indicate dry, green intermediate and blue wet $\theta_{ant}$. Thick lines indicate big, mid-sized lines medium and thin lines small $P_{sub}$.  Dashed black lines divide TTDs into four parts, each part controlled by different properties. Note the log-log axes. Insets show cumulative outflow probability distributions.

[Figure]

190

Figure S12: Two TTDs from the FHWB (blue) and TLDS (yellow) scenarios. Each one modified by three functions of exponential decay (with half-lives $t_{1/2}$ of 10, 100 and 1000 days). The fraction of mass eventually leaving the system ($\%_M$) can differ greatly: for a half-life of 100 days, the FHWB TTD still delivers 59 % of the original input to discharge while the TLDS TTD only delivers 2 %.

[Figure]

| Name | THWB | | TLDS | |
|---|---|---|---|---|
| Dispersivity | *Large* | *Small* | *Large* | *Small* |
| 1st Quartile | 45 | 45 | 458 | 462 |
| Median | 85 | 86 | 785 | 810 |
| Mean | 110 | 109 | 1009 | 1037 |
| 3rd Quartile | 136 | 136 | 1308 | 1369 |
| Stand Dev | 173 | 172 | 880 | 905 |
| Skewness | 29 | 30 | 3 | 3 |
| Exc Kurtosis | 1426 | 1449 | 20 | 19 |

short    long

wider    narrower
more skewed    less skewed
more peaked    flatter

**Table S1. Metrics of the TTDs for the simulations with larger (5 m) and smaller (0.5 m) values of the longitudinal dispersivity $\alpha_L$. All times are given in days.**

**D_soil / K_s / θ_ant / P_sub — DEEP (THICK)**

| | | HIGH | | | | | | | | LOW | | | | | | | |
|---|---|---|---|---|---|---|---|---|---|---|---|---|---|---|---|---|---|
| | DRY | | INT | | | WET | | | DRY | | | INT | | | WET | | |
| SMALL | MED | BIG | SMALL | MED | BIG | SMALL | MED | BIG | SMALL | MED | BIG | SMALL | MED | BIG | SMALL | MED | BIG |
| THDS | THDM | THDB | THIS | THIM | THIB | THWS | THWM | THWB | TLDS | TLDM | TLDB | TLIS | TLIM | TLIB | TLWS | TLWM | TLWB |

Dispersivity:

| Name | THWB | | TLDS | |
|---|---|---|---|---|
| Dispersivity | Large | Small | Large | Small |

Porosity:

| Name | THDM | | | THIM | | | THWM | | |
|---|---|---|---|---|---|---|---|---|---|
| Porosity | Small | Normal | Large | Small | Normal | Large | Small | Normal | Large |

Bedrock Conductivity:

| | VLow | Low | MLow | MHigh | High | VHigh |
|---|---|---|---|---|---|---|

Decay of Hydraulic Conductivity:

| Name | THDB | | THWB | | TLDB | | TLWB | |
|---|---|---|---|---|---|---|---|---|
| Decay | No | Yes | No | Yes | No | Yes | No | Yes |

Precipitation Frequency:

| Arid | THDM | Humid |
|---|---|---|

Catchment Shape:

| Name | THDM | | | THWM | | |
|---|---|---|---|---|---|---|
| Shape | Top | Mid | Bot | Top | Mid | Bot |

Water Retention Curve:

| Name | THDS | | THDB | | THWS | | THWB | | TLDS | | TLDB | | TLWS | | TLWB | |
|---|---|---|---|---|---|---|---|---|---|---|---|---|---|---|---|---|
| WRC | Silt | Sand | Silt | Sand | Silt | Sand | Silt | Sand | Silt | Sand | Silt | Sand | Silt | Sand | Silt | Sand |

Extreme Precipitation after Full Saturation:

| K_s | HIGH | | | | LOW | | | |
|---|---|---|---|---|---|---|---|---|
| Name | THWB | THSB | THSB+ | THSB++ | TLWB | TLSB | TLSB+ | TLSB++ |

**Table S2:** Information on which of the base-case scenarios (upper table) the other scenarios (dispersivity – italic; porosity – blue; bedrock conductivity – orange; decay in hydraulic conductivity – red; precipitation frequency – green; catchment shape – bold; soil water retention curve – purple; extreme precipitation after full saturation – yellow) are based upon.

**DEEP (THICK)**

| | | HIGH | | | | | | | | LOW | | | | | | | |
|---|---|---|---|---|---|---|---|---|---|---|---|---|---|---|---|---|---|
| | DRY | | INT | | | WET | | | DRY | | | INT | | | WET | | |
| SMALL | MED | BIG | SMALL | MED | BIG | SMALL | MED | BIG | SMALL | MED | BIG | SMALL | MED | BIG | SMALL | MED | BIG |
| THDS | THDM | THDB | THIS | THIM | THIB | THWS | THWM | THWB | TLDS | TLDM | TLDB | TLIS | TLIM | TLIB | TLWS | TLWM | TLWB |

| | THDS | THDM | THDB | THIS | THIM | THIB | THWS | THWM | THWB | TLDS | TLDM | TLDB | TLIS | TLIM | TLIB | TLWS | TLWM | TLWB |
|---|---|---|---|---|---|---|---|---|---|---|---|---|---|---|---|---|---|---|
| $F_{yw\,Gam}$ | 0.11 | 0.29 | 0.89 | 0.14 | 0.30 | 0.77 | 0.19 | 0.32 | 0.63 | 0.04 | 0.09 | 0.15 | 0.05 | 0.10 | 0.15 | 0.08 | 0.13 | 0.18 |
| $F_{yw\,Mod}$ | 0.01 | 0.03 | 0.11 | 0.05 | 0.11 | 0.26 | 0.18 | 0.25 | 0.40 | 0.00 | 0.01 | 0.08 | 0.01 | 0.03 | 0.12 | 0.05 | 0.12 | 0.20 |

**SHALLOW (FLAT)**

| | FHDS | FHDM | FHDB | FHIS | FHIM | FHIB | FHWS | FHWM | FHWB | FLDS | FLDM | FLDB | FLIS | FLIM | FLIB | FLWS | FLWM | FLWB |
|---|---|---|---|---|---|---|---|---|---|---|---|---|---|---|---|---|---|---|
| $F_{yw\,Gam}$ | 0.27 | 0.84 | 1.00 | 0.28 | 0.74 | 0.96 | 0.30 | 0.60 | 0.86 | 0.09 | 0.17 | 0.23 | 0.11 | 0.17 | 0.24 | 0.14 | 0.19 | 0.25 |
| $F_{yw\,Mod}$ | 0.03 | 0.11 | 0.40 | 0.10 | 0.25 | 0.51 | 0.25 | 0.39 | 0.61 | 0.01 | 0.05 | 0.20 | 0.02 | 0.10 | 0.23 | 0.09 | 0.17 | 0.25 |

Young Water Threshold: short – long
Young Water Fraction: small – large

**Table S23.** Young water fractions ($F_{yw}$) for the 36 different base-case scenarios. The young water fractions are determined from the best-fit Gamma distributions ($F_{yw\,Gam}$) and from the modeled TTDs themselves ($F_{yw\,Mod}$).

| Name | THDM | 1 | 2 | 3 | 4 | 5 | 6 | 7 | 8 | 9 | 10 | 11 | 12 | 13 | 14 | 15 | μ | σ |
|---|---|---|---|---|---|---|---|---|---|---|---|---|---|---|---|---|---|---|
| 1st Quartile | 137 | 138 | 143 | 136 | 144 | 136 | 179 | 166 | 163 | 181 | 120 | 162 | 136 | 165 | 159 | 123 | 150 | 19 |
| Median | 207 | 220 | 208 | 245 | 241 | 227 | 250 | 251 | 239 | 246 | 207 | 244 | 236 | 242 | 244 | 204 | 234 | 16 |
| Mean | 280 | 277 | 280 | 286 | 291 | 280 | 306 | 300 | 300 | 302 | 262 | 296 | 285 | 298 | 296 | 265 | 288 | 13 |
| 3rd Quartile | 366 | 357 | 339 | 358 | 367 | 360 | 368 | 363 | 361 | 366 | 349 | 362 | 358 | 355 | 365 | 351 | 359 | 8 |
| Stand Dev | 298 | 299 | 294 | 298 | 302 | 302 | 295 | 298 | 295 | 297 | 300 | 296 | 302 | 299 | 297 | 299 | 298 | 2.5 |
| Skewness | 14.8 | 15.7 | 15.6 | 15.4 | 15.3 | 15.5 | 15.6 | 15.6 | 15.7 | 15.6 | 15.4 | 15.6 | 15.5 | 15.9 | 15.5 | 15.4 | 15.5 | 0.16 |
| Exc Kurtosis | 407 | 433 | 434 | 423 | 416 | 422 | 432 | 432 | 436 | 433 | 421 | 433 | 424 | 439 | 429 | 422 | 429 | 6.5 |

short – long
wider / more skewed / more peaked – narrower / less skewed / flatter

**Table S34.** Distribution metrics for the 15 TTDs resulting from different precipitation event sequences. For comparison we also show the metrics for the THDM scenario which uses an actually measured time series of precipitation and has a slightly different

distribution of precipitation event amounts and interarrival times but otherwise similar catchment and climate properties. The means (μ) and standard deviations (σ) of the metrics of the 15 scenarios are also shown. All times are given in days.

**DEEP (THICK)**

| $D_{soil}$ | | | | HIGH | | | | | | | | | LOW | | | | | |
|---|---|---|---|---|---|---|---|---|---|---|---|---|---|---|---|---|---|---|
| $K_s$ | | DRY | | | INT | | | WET | | | DRY | | | INT | | | WET | |
| $\theta_{art}$ | | | | | | | | | | | | | | | | | | |
| $P_{sub}$ | SMALL | MED | BIG | SMALL | MED | BIG | SMALL | MED | BIG | SMALL | MED | BIG | SMALL | MED | BIG | SMALL | MED | BIG |
| Name | THDS | THDM | THDB | THIS | THIM | THIB | THWS | THWM | THWB | TLDS | TLDM | TLDB | TLIS | TLIM | TLIB | TLWS | TLWM | TLWB |
| Δ Mean InvGau | 6 | -4 | -9 | 12 | -2 | -6 | 21 | 4 | -1 | 31 | 25 | 22 | 102 | 44 | 32 | 60 | 35 | 18 |
| Gamma | -282 | -152 | -109 | -132 | -94 | -81 | 26 | -25 | -42 | -423 | -172 | -10 | -186 | -74 | 30 | -52 | 31 | 84 |
| LogN | 8 | -3 | -9 | 17 | 0 | -6 | 30 | 6 | 0 | 38 | 32 | 32 | 115 | 56 | 44 | 75 | 49 | 32 |
| Δ Median InvGau | -32 | 7 | 6 | -6 | 11 | -1 | 1 | -6 | -8 | -22 | -19 | -13 | 17 | -44 | -21 | -28 | -50 | -37 |
| Gamma | -15 | 17 | 8 | 12 | 20 | 2 | 17 | 2 | -4 | 18 | 10 | 8 | 59 | -13 | 1 | 6 | -26 | -20 |
| LogN | -28 | 10 | 7 | -1 | 14 | 0 | 6 | -3 | -6 | -13 | -11 | -6 | 27 | -35 | -14 | -18 | -43 | -31 |
| Fit InvGau | 0.44 | 0.32 | 0.33 | 0.68 | 0.22 | 0.19 | 1.20 | 0.31 | 0.30 | 0.51 | 0.92 | 1.10 | 1.78 | 1.80 | 1.65 | 2.63 | 2.40 | 2.10 |
| Gamma | 0.38 | 0.79 | 0.64 | 0.38 | 0.66 | 0.35 | 0.25 | 0.31 | 0.17 | 1.28 | 0.52 | 0.40 | 2.11 | 1.36 | 0.90 | 0.36 | 0.32 | 0.26 |
| LogN | 0.37 | 0.38 | 0.32 | 0.59 | 0.26 | 0.16 | 0.96 | 0.25 | 0.23 | 0.38 | 0.68 | 0.90 | 1.25 | 1.32 | 1.22 | 1.95 | 1.83 | 1.60 |

**SHALLOW (FLAT)**

| $D_{soil}$ Name | FHDS | FHDM | FHDB | FHIS | FHIM | FHIB | FHWS | FHWM | FHWB | FLDS | FLDM | FLDB | FLIS | FLIM | FLIB | FLWS | FLWM | FLWB |
|---|---|---|---|---|---|---|---|---|---|---|---|---|---|---|---|---|---|---|
| Δ Mean InvGau | -7 | -11 | -4 | -7 | -12 | -7 | 1 | -4 | -1 | 13 | 10 | 9 | 34 | 16 | 10 | 29 | 15 | 8 |
| Gamma | -156 | -113 | -67 | -98 | -89 | -54 | -23 | -45 | -29 | -195 | -56 | 11 | -87 | -17 | 40 | 1 | 35 | 57 |
| LogN | -5 | -11 | -4 | -5 | -11 | -7 | 4 | -3 | -1 | 19 | 15 | 15 | 45 | 26 | 19 | 42 | 26 | 18 |
| Δ Median InvGau | 10 | 3 | -4 | 10 | -2 | -2 | -5 | -10 | -2 | -33 | -18 | 6 | -41 | -27 | 0 | -32 | 3 | 1 |
| Gamma | 21 | 6 | -2 | 20 | 2 | 0 | 4 | -6 | 1 | -7 | 1 | 20 | -12 | -6 | 14 | -7 | 20 | 13 |
| LogN | 13 | 4 | -3 | 13 | -1 | 0 | -2 | -8 | 1 | -25 | -12 | 11 | -33 | -21 | 4 | -25 | 8 | 6 |
| Fit InvGau | 0.38 | 0.41 | 0.14 | 0.36 | 0.30 | 0.20 | 0.36 | 0.25 | 0.29 | 0.68 | 0.53 | 0.44 | 2.13 | 1.40 | 0.98 | 1.71 | 1.21 | 0.92 |
| Gamma | 0.85 | 0.77 | 0.14 | 0.92 | 0.54 | 0.38 | 0.47 | 0.35 | 0.13 | 0.73 | 0.73 | 0.44 | 2.51 | 1.61 | 0.98 | 1.02 | 0.81 | 0.64 |
| LogN | 0.43 | 0.40 | 0.14 | 0.38 | 0.27 | 0.20 | 0.28 | 0.24 | 0.26 | 0.52 | 0.52 | 0.39 | 1.69 | 1.14 | 0.74 | 1.24 | 0.89 | 0.65 |

Legend: small error — large error; small error — large error; good fit — bad fit

[revised manuscript text omitted]